# Generative AI Act II:
# Test Time Scaling Drives Cognition Engineering

## Abstract

The first generation of Large Language Models—what might be called "*Act I*" of generative AI (2020-2023)—achieved remarkable success through massive parameter and data scaling, yet exhibited fundamental limitations such as knowledge latency, shallow reasoning, and constrained cognitive processes. During this era, *prompt engineering* emerged as our primary interface with AI, enabling dialogue-level communication through natural language. We now witness the emergence of "***Act II***" (2024-present), where models are transitioning from knowledge-retrieval systems (in latent space) to thought-construction engines through test-time scaling techniques. In this paper, we clarify the conceptual foundations of ***cognition engineering*** and explain why this moment is critical for its development. We systematically break down these advanced approaches through comprehensive tutorials and optimized implementations, democratizing access to cognition engineering and enabling every practitioner to participate in AI's second act.

| Act I: Prompt Engineering (Knowledge-driven) | Act II: Cognition Engineering (Thought-driven) |
|---|---|
| **Pretraining era** 
 **2020-2023** | **Test-time scaling era** 
 **2024-present** |

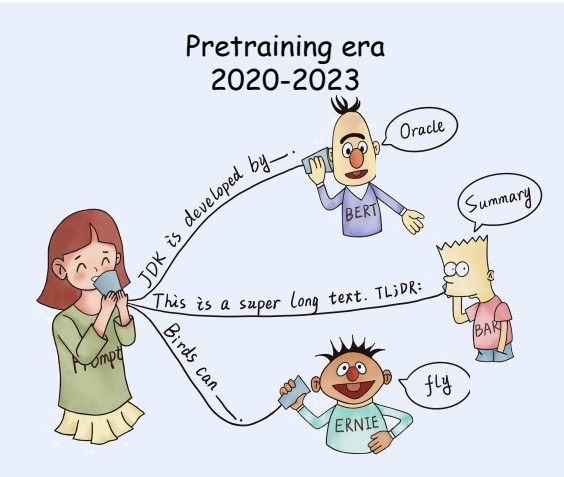

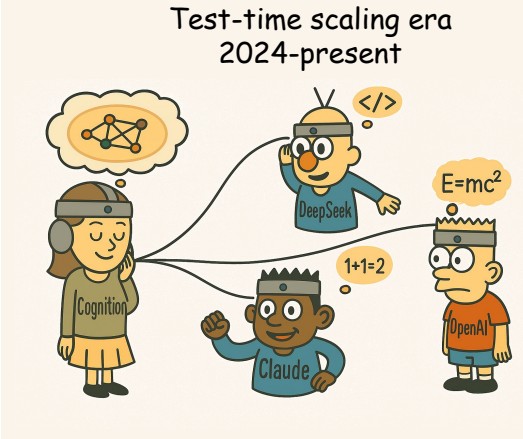

Act I: Prompt Engineering (Knowledge-driven)
1. examples are GPT-4, Llama 3
2. good at **knowledge memorization**, but lack deep thinking
3. require large, **human**-generated data
4. tech stack: *Pretraining -> SFT -> RLHF*

Act II: Cognition Engineering (Thought-driven)
1. examples are o1, R1
2. good at **deep thinking**, unlock many applications such as scientific discovery
3. thought-intensive, **AI**-generated data
4. tech stack: *Pretraining -> SFT -> RL*

## Contents

# 1 Introduction

In recent years, Large Language Models (LLMs) such as GPT (OpenAI, 2023), LLaMA (Meta, 2024; 2023), and Claude (Anthropic, 2024a) have emerged as powerful knowledge management tools through extensive pre-training and fine-tuning processes. These models, trained on vast corpora of human-generated text, have successfully organized and systematized accumulated human knowledge. Through a paradigm defined by scaling pre-training data, computation, and model parameters (Kaplan et al., 2020), these systems can engage in natural language conversations, retrieve information, generate content, and answer questions across diverse domains (Zhao et al., 2023b; Wang et al., 2024g; Zheng et al., 2023a). This first generation of LLMs—what we might call "Act I" of generative AI—introduced a fundamental shift in human-AI interaction. The cornerstone of this first act has been "*prompt engineering*"—the art of crafting inputs that guide models toward desired outputs (Liu et al., 2021; Sahoo et al., 2024). This innovation enabled humans to communicate with AI systems using natural language for the first time, dramatically lowering the barriers to human-machine interaction. Act I focused primarily on gathering and organizing existing knowledge through ever-larger models trained on increasingly vast datasets. However, despite their impressive capabilities, Act I models exhibit several significant limitations: (i) **knowledge latency**: These models primarily learn high-frequency information that has had time to accumulate in their training data, leaving them with limited understanding of emerging knowledge and concepts (Huang et al., 2025b). (ii) **shallow reasoning**: While capable of basic logical inferences, they struggle with problems requiring multi-step, deep reasoning processes (Zhang et al., 2024d; Mirzadeh et al., 2024; Kambhampati, 2024). (iii) **limited thought processes**: They fail to demonstrate human-like depth of thought, particularly when confronting novel or open-ended questions (Wu et al., 2024d). These constraints have kept Act I models primarily confined to knowledge retrieval and simple reasoning tasks, still considerably distant from achieving artificial general intelligence (AGI). Just as knowledge alone is insufficient for human intelligence development, merely amassing information has proven inadequate for AI systems to approach human-like intelligence—they must also develop the capacity for deep thinking and reasoning (Newell et al., 1959; 1972).

Recently, the AI field has witnessed a profound paradigm shift. A new technical approach centered on **"test-time scaling"** is redefining the boundaries of what LLMs can achieve (OpenAI, 2024; DeepSeek et al., 2025; Snell et al., 2024), inaugurating the second act of generative AI—***cognition engineering***. In this paper, we define cognition engineering as follows:

> *Cognition Engineering is the systematic and constructive development of AI thinking capabilities through test-time scaling paradigms that transcend traditional pretraining approaches. This methodology represents the deliberate cultivation of deep reasoning processes in LLMs through interventions at inference time and corresponding training strategies.*

At its core, cognition engineering sits at the intersection of two fundamental concepts: "***Cognition***" in this context refers not merely to knowledge acquisition but to deep cognition—the ability to perform complex reasoning, engage in deliberate thinking, connect disparate concepts, and generate novel insights. It encompasses the meta-cognitive processes (Metcalfe & Shimamura, 1994) that allow for understanding *not just "what" but "why" and "how"*—the very essence of human intellectual advancement. "***Engineering***" here signifies a constructive approach rather than a purely emergent one. It moves beyond the limitations of mere scaling toward a more intentional construction of cognitive capabilities through targeted interventions in both training methodologies (e.g., reinforcement learning) and inference optimization (e.g., extending inference-time computation).

Cognition engineering represents a comprehensive technological paradigm shift in LLM development. From the inference perspective, it transitions from crafting prompt templates that retrieve knowledge from LLMs to designing test-time scaling strategies that conduct deeper and more comprehensive searches through knowledge spaces. This evolution demands rigorous analysis of test-time scaling strategy components, characteristics, and efficiency, which underscores the necessity for a structured engineering approach. From the training perspective, cognition engineering redirects computational resources from knowledge-focused pre-training toward developing deep thinking abilities through techniques like reinforcement learning (RL).

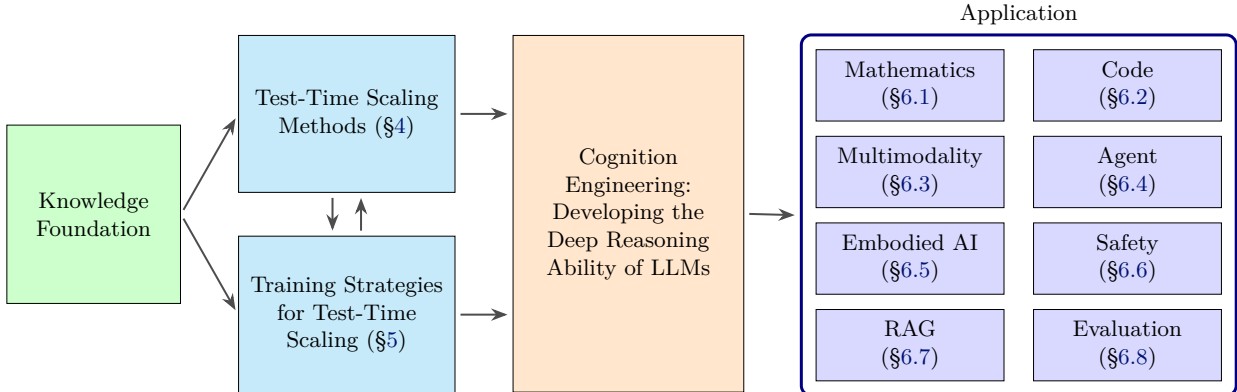

Figure 1: The roadmap of cognition engineering. RAG refers to Retrieval-Augmented Generation.

This paper will explore in depth the definition, technical foundations, and application prospects of cognition engineering. First, we will clarify the conceptual connotations of cognition engineering (§2) and explain why now is the critical moment for its development (§3). Next, we will analyze in detail the technical foundations of test-time scaling (§4) and various training strategies for it (§5). We will then examine the systemic changes cognition engineering brings to AI research and the applications that have already emerged (§6). Finally, we will discuss the infrastructure (§7) and identify several future directions for cognition engineering (§8). We provide comparison to existing works in §9. Through these discussions, we aim to outline the contours of generative AI's second act and provide researchers and practitioners with a framework for thinking in this new paradigm.

## 2   What – Cognition Engineering Definition

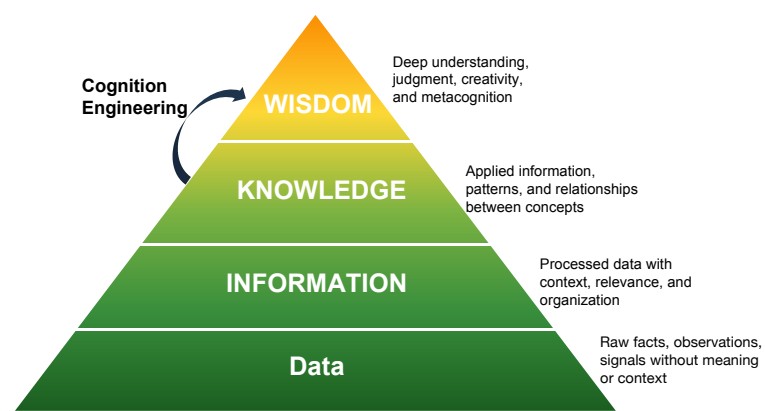

Figure 2: The DIKW pyramid and its relationship to cognition engineering paradigm.

To understand the essence of cognition engineering, we can use the Data-Information-Knowledge-Wisdom (DIKW) pyramid theory (Zeleny, 1987; Ackoff, 1989) as a conceptual framework. The DIKW theory, which portrays the cognitive process as a hierarchical transformation from raw data to contextualized information, then to applicable knowledge, and ultimately to profound wisdom, has been a fundamental theory in information science and knowledge management[1] (Rowley, 2007; Zins, 2007). In the theory, the knowledge level manifests as understanding and application of information, including mastery of rules, patterns, and

---

[1]While the framework has been criticized for its ambiguous divisions (Frické, 2009; Baskarada & Koronios, 2013), its distinction between the knowledge and wisdom levels provides a useful conceptual basis for articulating the goal of cognition engineering.

relationships; while the wisdom level embodies deep comprehension of knowledge, involving judgment, creativity, and metacognitive abilities. The first-generation LLMs achieved significant breakthroughs at the knowledge level. Cognition engineering represents the crucial step toward advancing to the wisdom level.

In psychology and cognitive science, cognition refers to the complex mental processes through which organisms acquire and process information, form knowledge, and apply it to problem-solving (Newell et al., 1972; Posner et al., 1993; Von Eckardt, 1995; Núñez et al., 2019). However, what cognition engineering pursues is not merely this basic cognitive capability, but the wisdom-level cognition described by DIKW—the ability to understand deep principles, engage in creative thinking, and demonstrate judgment. It aims to develop the deep thinking ability of LLMs to engage in multi-layered, complex reasoning exploring multiple pathways to resolution—alongside metacognitive capabilities that allow reflection on one's own thought processes. These cognitive abilities are the foundation of humanity's continuous advancement in scientific discovery and technological innovation.

## 3 Why & Why Now – Technical Foundation

### 3.1 The Necessity of Cognition Engineering

The rise of cognition engineering is not coincidental but a direct response to the "wisdom gap" encountered in the development of LLMs. Despite significant advances in knowledge retrieval, content generation, and basic reasoning, LLMs still exhibit notable shortcomings at the wisdom level:

**Limitations in Complex Reasoning**  Current models perform poorly on problems requiring multi-step deep reasoning (Zhang et al., 2024d; Mirzadeh et al., 2024; Kambhampati, 2024). Even the most advanced models struggle with reliable mathematical proofs, complex scientific problem-solving, or multidimensional analysis (Yang et al., 2024b; Rein et al., 2023). These tasks require models to decompose problems into sub-problems, explore multiple possible reasoning paths, and conduct deep logical analysis—capabilities beyond what scaling pre-training data alone can achieve.

**Challenges in Knowledge Updating and Creation**  Pre-trained models' knowledge is fixed at the end of training, unable to automatically adapt to new developments and changes. More importantly, they struggle to generate *truly original insights or discoveries—the essence of scientific discovery* is not merely understanding known facts but proposing new hypotheses, designing experimental methods, and drawing new conclusions from results. This knowledge creation ability requires going beyond simple knowledge retrieval and pattern recognition.

**Elevated Application Requirements**  As AI applications expand from simple tasks to complex decision-making, scientific research, and creative work, demands for AI systems' wisdom-level capabilities also increase (OpenAI, 2025b;a). Users are no longer satisfied with answers based on statistical patterns (*knowledge level*); they desire thoughtful analysis, multi-perspective considerations, and innovative insights (*wisdom level*) from AI.

### 3.2 Three Pillars

Cognition engineering emerges at this specific moment due to multiple technological breakthroughs reaching maturity simultaneously. These breakthroughs collectively create the necessary conditions enabling AI to progress from knowledge management to deep cognitive capabilities. The rise of cognition engineering stems from three key technological pillars:

#### 3.2.1 Knowledge Foundation

The first enabling foundation for cognition engineering is the fundamental transformation in how LLMs acquire knowledge. Modern foundation models have not only achieved exponential growth in training data volume (such as Llama 2's 2 trillion token training scale (Meta, 2023)) but more importantly, have experienced a qualitative transformation. Pre-training data has evolved from simple web-scraped text to carefully

curated knowledge corpora (Shao et al., 2024; Zhou et al., 2024b; Wang et al., 2023d; Yang et al., 2024a). These datasets now integrate scientific literature and technical documentation with mathematical textbooks and problem sets, multi-language programming code repositories, and structured knowledge from specialized domains, forming a much richer knowledge ecosystem than previously available. This comprehensive knowledge foundation is a necessary prerequisite for cognition engineering—without this extensive embedded knowledge, models would lack the raw materials required for deep thinking.

### 3.2.2 Test-time Scaling Foundation

The second critical pillar enabling cognition engineering is the fundamental reconceptualization of how computational resources are allocated during the inference phase—what we term "Test-Time Scaling." Traditional inference approaches were constrained by fixed output lengths and single-pass generation paradigms. Recently, a series of technical breakthroughs has significantly extended models' reasoning capabilities. Chain-of-Thought (CoT) prompting (Wei et al., 2022) methods encourage models to perform step-by-step reasoning like human problem-solving processes, clearly articulating intermediate steps. Tree search (Yao et al., 2023a; Hao et al., 2023; Feng et al., 2023) allow for systematic exploration of multiple reasoning pathways simultaneously, rather than being confined to a single line of thinking. Self-correction and verification techniques (DeepSeek et al., 2025; Kumar et al., 2024; Qu et al., 2024) further enhance these capabilities, enabling models to evaluate their own reasoning, identify potential errors, and refine their approaches—mimicking human metacognitive processes.

### 3.2.3 Self-Training Foundation

The third pillar of cognition engineering is advanced self-training methodologies. As demonstrated by DeepSeek-R1 (DeepSeek et al., 2025) and subsequent research (Gandhi et al., 2025; Yu et al., 2025a), training with RL using verifiable rewards enables models to master complex cognitive behaviors including reflection, backtracking, and verification when solving challenging problems. Through this process, models learn to dynamically allocate computational resources according to problem complexity, effectively internalizing test-time scaling techniques. Additionally, iterative self-training on reasoning trajectories generated through test-time scaling methods facilitates continuous improvement (Zelikman et al., 2022; Feng et al., 2023; Xiong et al., 2025), allowing AI systems to progressively enhance their problem-solving abilities.

### 3.3 Roadmap

We present the roadmap for cognition engineering in Figure 1. The core of cognition engineering is developing the deep reasoning ability of LLMs. The three pillars described above advance this goal significantly, and this area has seen rapid development recently. In §4, we delve into mainstream test-time scaling methods, which correspond to the second pillar. In §5, we discuss the training strategies for test-time scaling methods, covering the content of the self-training pillar. For readers interested in the knowledge foundation, we refer them to the survey (Zhou et al., 2025b) for further discussion. In §6, we discuss how to apply cognition engineering in different domains. Finally, we briefly discuss the infrastructure aspect in §7 and general future directions in §8.

## 4 How – Part I: Test-Time Scaling Methods

Given a query $q$ and a generator $g$, the test-time scaling method can be abstracted to a search strategy $M$ that guides the generator $g$ to find the optimal response (Welleck et al., 2024):

$$y \sim M(.|q, g, \phi) \tag{1}$$

where $\phi$ represents any additional inputs such as scoring functions $v$ (also known as value functions, reward models, or verifiers[2]) and hyperparameters of the strategy.

---

[2]These words share subtle differences in context and we choose the most appropriate term for each method.

**Scaling laws** For any test-time scaling method, there exist corresponding scaling dimensions $\lambda$ within $\phi$ that directly determine the computation cost during inference. The scaling laws of $M$ describe the relationship between $\lambda$ and performance.

**Scaling efficiency** Given a computation budget[3] $C$, we define an abstract function $f : C \times M \to \mathbb{R}$ that maps from the computation budget $C$ and the test-time scaling method $M$ to performance. The scaling efficiency measures this performance relative to the computation budget (Qu et al., 2025a):

$$\text{efficiency} = \frac{f(C, M)}{C} \tag{2}$$

The high-level strategies to improve efficiency of $M$ can be categorized into:[4] 1) *Optimizing individual test-time scaling methods:* This involves carefully selecting and tuning components within computation budget constraints, or leveraging additional training-time compute to optimize models specifically for test-time scaling; 2) *Combining multiple test-time scaling methods:* This includes simultaneously combining multiple methods or selecting appropriate test-time scaling methods according to different contexts.

In the following sections, we will investigate four primary test-time scaling methods: parallel sampling[5] (§4.1), tree search (§4.2), multi-turn correction (§4.3), and long CoT (§4.4). For each test-time scaling method, we will cover the construction method, the scaling laws, and how to improve the scaling efficiency from the individual optimization aspect. Furthermore, we compare these test-time scaling methods across multiple dimensions (§4.5) and discuss how to effectively combine them for enhanced performance (§4.6).

### 4.1  Parallel Sampling

### 4.1.1  Key Components

The parallel sampling algorithm samples a set of candidate responses $\mathcal{Y} = \{y_i\}_{i=1}^{N}$ independently from the generator for the same query, where N is the sampling number, and selects the targeted response or answer from them. This approach can be conceptualized as a global search within the knowledge space (Snell et al., 2024). The selection methods are as follows:

- **F1: Best-of-N (BoN).** This method utilizes a scoring function $v$ to evaluate each response and selects the one with the highest score:

$$y^* = \arg\max_{\tilde{y} \in \mathcal{Y}} v(\tilde{y}) \tag{3}$$

  The scoring function $v$ can be external tools that directly verify the effectiveness of the response, such as code interpreters (Li et al., 2022; Chen et al., 2023a) or math proof checkers (Brown et al., 2024). For tasks lacking verification tools, $v$ can be a specialized trained model. For instance, Cobbe et al. (2021) train the outcome reward model (ORM) to score the entire response, while Lightman et al. (2023); Uesato et al. (2022) train the process reward model (PRM) to score each step in the response and apply an aggregation function to determine the overall response score. Self-Certainty (Kang et al., 2025) eliminates the need for an additional reward model by leveraging the generator's inherent probability distribution for scoring.

---

[3]It can be measured by FLOPs, running time, token numbers, etc.

[4]In this section, we do not consider the model compression techniques like model quantization or inference acceleration from infrastructure aspects as they are orthogonal to the method design.

[5]Parallel sampling is considered a test-time scaling method as it uses the sampling number as a scaling dimension, increasing the inference cost to achieve better performance.

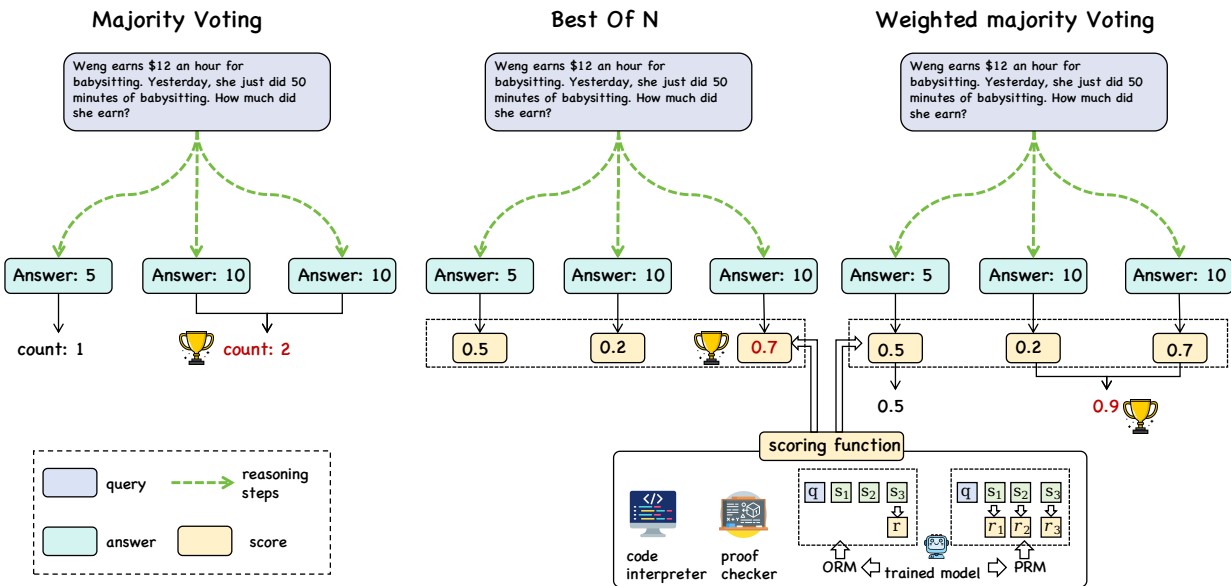

Figure 3: Illustration of parallel sampling selection methods: Best-of-N (F1), Majority voting (F2), and Combined strategy (F3). Here, $q$ denotes the query, $s_i$ is the $i$-th reasoning step, $r_i$ is the reward for step $i$, and $r$ is the total reward for the response. ORM and PRM are the outcome and process reward models, respectively.

- **F2: Majority voting.** Majority voting (or self-consistency (Wang et al., 2023c)) selects the most frequent answer from the candidates:

$$y^* = \arg\max_{\tilde{y} \in \mathcal{Y}} \sum_{\hat{y} \in \mathcal{Y}} g(\tilde{y}, \hat{y}) \tag{4}$$

$$g(\tilde{y}, \hat{y}) = \begin{cases} 1 & \text{if } \tilde{y} \text{ is equivalent to } \hat{y}, \\ 0 & \text{otherwise,} \end{cases} \tag{5}$$

where $g$ is an automatic grading function that first extracts the answers from the responses and checks for equivalence. While this method is lightweight, the requirement for easy answer equivalence comparison limits its applicability for open-ended tasks. Universal Self-Consistency (Chen et al., 2023b) employs LLMs themselves to select the most consistent answer among multiple candidates, though the limited context window size of models still presents challenges for large sampling numbers.

- **F3: Combining voting and scoring strategy.** The scoring strategy can help select targeted low-frequency responses but heavily depends on the reliability of the scoring function, whereas the voting strategy offers greater robustness but has a more fixed upper bound. This combined method leverages advantages from both approaches for more robust selection (Sun et al., 2024b). For example, weighted majority voting (Uesato et al., 2022; Liu et al., 2023d) re-ranks answer clusters according to the sum of the scores in each cluster and selects the answer cluster with the highest score:

$$y^* = \arg\max_{\tilde{y} \in \mathcal{Y}} \sum_{\hat{y} \in \mathcal{Y}} g(\tilde{y}, \hat{y}) v(\hat{y}) \tag{6}$$

Figure 3 illustrates these selection methods.

### 4.1.2 Scaling Laws

The main scaling dimension in parallel sampling is the sampling number N. We investigate the relationship between N and various performance metrics. Specifically, we focus on two types of metrics: Pass@N, which

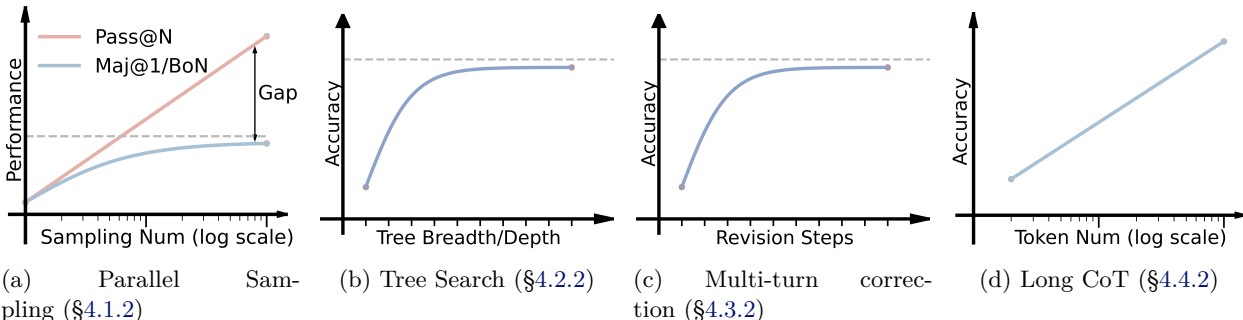

Figure 4: The relationship between scaling dimensions and performance for each test-time scaling method.

represents the probability of generating at least one correct response among N candidates, and metrics such as Maj@1 or BoN, which measure the practical performance of parallel sampling.

**The monotonic growth relationship between N and Pass@N** Brown et al. (2024) investigate the relationship between N and Pass@N across different models and tasks. The Pass@N grows steadily with sampling numbers. Moreover, the relationship between the two is often log-linear as demonstrated in Figure 4a, similar to the training time scaling law observed (Kaplan et al., 2020).

**Scaling Pass@N does not translate to real-world performance improvements** Although the continuous improvement of Pass@N with increased sampling numbers is promising, there remains a gap between this metric and true performance. This gap exists for several reasons. First, performance improvements can only be realized when appropriate tools exist to select the correct response from the sample set. However, perfect verifiers do not exist for most tasks. As Brown et al. (2024) observe, when using ArmoRM-Llama3-8B-v0.1 (Wang et al., 2024d) as the scoring model, a significant disparity emerges between Pass@N and practical metrics like Maj@1 or BoN (see Figure 4a). Second, the verifiers themselves can be hacked. Code may pass unit tests but fail with additional test cases (Stroebl et al., 2024), or mathematical solutions may reach correct answers through incorrect reasoning (Xia et al., 2024), leading to the false positive problems. Stroebl et al. (2024) observe that the false positive rate increases as the Pass@1 accuracy decreases in code tasks, concluding that this imposes an upper bound on the accuracy of resampling-based inference scaling, even with infinite computational resources. For practical application methods, such as majority voting or scoring methods, performance tends to saturate (Brown et al., 2024; Wu et al., 2024c; Li et al., 2024c) and may even degrade as the number of samples increases (Chen et al., 2024c) due to imperfect verifiers.

### 4.1.3 Improving Scaling Efficiency

The strategies to improve the scaling efficiency of parallel sampling are as follows:

**Query-aware sampling** Applying a fixed sampling number for all queries is not optimal, as difficult problems require more sampling while easier ones need fewer. This line of methods employs adaptive sampling numbers for different queries based on difficulty to improve sampling efficiency. Chen et al. (2024c) categorize queries into easy and difficult cases according to model uncertainty and apply different sampling numbers accordingly. Difficulty-Adaptive Self-Consistency (Wang et al., 2024f) prompts the model to rank query difficulty and distributes sampling numbers based on this ranking.

**Early stopping strategy** This method estimates the quality of responses during the sampling process and decides when to stop sampling early by utilizing prior knowledge or model estimation. It includes terminating sampling when observed answers are identical or fit predefined distributions within a small window size (Aggarwal et al., 2023; Li et al., 2024f; Wan et al., 2024), or training the generator itself to estimate the confidence of the response and stopping once a high-confidence response is observed (Huang et al., 2025a). Moreover, Speculative Rejection (Sun et al., 2024a) and ST-BoN (Wang et al., 2025e) propose

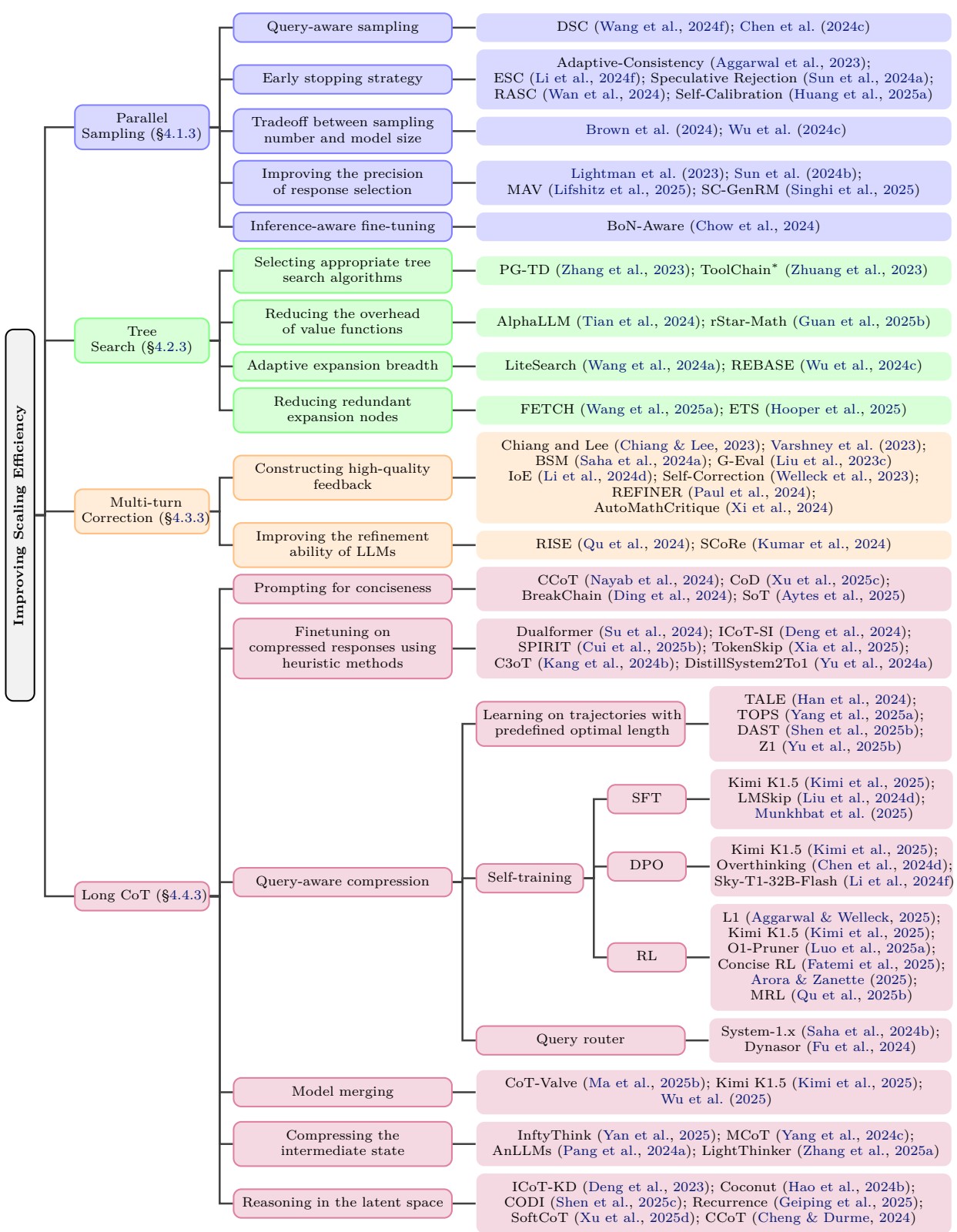

Figure 5: Overview of methods of improving scaling efficiency.

sampling responses in parallel and halting the decoding of responses with low reward model scores or self-estimation consistency scores to improve efficiency.

**Tradeoff between sampling number and model size**   Given a fixed inference computation budget, there exists a tradeoff between using larger models with fewer samples versus smaller models with more samples, considering the different computational costs across model sizes. Brown et al. (2024) observe that larger models perform better on code tasks while smaller models are more effective for mathematical tasks. Wu et al. (2024c) further find that for mathematical tasks, while smaller models are optimal, they also saturate earlier as the inference computation increases.

**Improving the precision of response selection**   Considering the importance of effective selection mechanisms, several works focus on improving their precision in response selection. Lightman et al. (2023) find that PRM is superior to ORM in BoN settings. Sun et al. (2024b) find that the performance of weighted majority voting is superior to majority voting or BoN when using large sampling numbers. MAV (Lifshitz et al., 2025) employs multiple verifiers to assess response quality and achieves better performance than a single verifier when the total computation budget for generator and verifiers is high.

**Inference-aware fine-tuning**   Chow et al. (2024) overcome the non-differentiable argmax operator within BoN sampling and develop BoN-Aware fine-tuning to directly optimize parallel sampling performance.

## 4.2  Tree Search

### 4.2.1  Key Components

The tree search method frames problems as searches over tree structures. Guided by a specific tree search algorithm, the generator searches in the search space $S$ and explore different problem-solving approaches, usually accompanied by value functions to assess node values in $S$. This framework enhances the model's deliberate planning ability. We detail each component below.

**Search space**   The search space defines the granularity of tree nodes, which significantly impacts search efficiency. It can be categorized as follows:

- **S1: Token.** Token-level search increases the optimality of candidate solutions but also incurs high computational costs due to its fine-grained nature. This approach is suitable for scenarios with low tolerance for even minor errors in individual tokens. PG-TD (Zhang et al., 2023) implements the Monte Carlo Tree Search (MCTS) algorithm for code tasks at the token level, as even minor changes in code may cause errors. PPO-MCTS (Liu et al., 2023b) employs token-level search methods to improve the helpfulness and harmlessness of responses.

- **S2: Step.** Step-level search balances search granularity and efficiency, making it the most common approach. The definition of "step" varies across different tasks. It can be sentences in solutions for reasoning problems (Yao et al., 2023a; Xie et al., 2023; Hao et al., 2023), actions in a simulated world (Gu et al., 2023; Zhuang et al., 2023), lines of code (Wang et al., 2024c), or proposed plans or hypotheses (Yao et al., 2023a; Wang et al., 2024e; 2023b).

- **S3: Solution.** Solution-level search[6] considers the expansion of tree nodes as updates to the whole response through actions like critique and revision (Zhang et al., 2024b;c). It overlaps with the multi-turn correction framework discussed later, and we also consider it as an ensemble of the two methods.

---

[6]It is noted in some works (Zeng et al., 2024) that they consider parallel sampling as a solution-level tree search. However, for parallel sampling, the solution tree nodes only involve the original solutions and do not involve node expansion behaviors, which are vital for tree search. Therefore, we do not merge it.

**Value function**   The value function estimates the value of candidate nodes for further pruning or exploitation. The popular methods to construct value functions are as follows:

- **E1: Self-evaluation.** This method directly instructs the generator to evaluate node values through well-crafted prompts. Tree-of-Thought (ToT) (Yao et al., 2023a) proposes to value each node independently or vote across nodes. Xie et al. (2023) design prompts in the form of multiple-choice questioning to better calibrate model predictions.

- **E2: Specialized trained models.** To reduce evaluation noise, this method utilizes specialized trained LLMs for evaluation (Gu et al., 2023; Kang et al., 2024a). This introduces process reward models (PRMs) for evaluating reasoning steps and token-level value functions for more fine-grained evaluation (Lee et al., 2024). For the PRM, it can be trained through the following approaches: 1) Human annotations: Lightman et al. (2023); Uesato et al. (2022) employ human labelers to label the correctness of each step. This method is costly and still cannot avoid noise in the training data; 2) Monte Carlo sampling: to achieve autonomous data annotation, this method employs Monte Carlo sampling that rolls out multiple completions from the current steps and estimates the rate leading to correct results (Wang et al., 2023a; 2024i; Havrilla et al., 2024; Luo et al., 2024). To improve sampling efficiency, OmegaPRM (Luo et al., 2024) utilizes binary search for error locating and integrates data collection into the search process; 3) From ORM to PRM: To avoid the high cost of training a PRM, a line of work (Lu et al., 2024; Yuan et al., 2024a) aims to derive a PRM from an ORM. AutoPSV (Lu et al., 2024) utilizes the ORM to automatically generate process annotations for each reasoning step by detecting its own confidence variations and thus uses the data for PRM training. Yuan et al. (2024a) theoretically demonstrate that a PRM can automatically derive from an ORM through simple reward parameterization. In the PRM utilization phase, Setlur et al. (2024b) demonstrate that process reward for a step should be advantages calculated by the difference of adjacent step values. For token-level value functions, these can directly come from the value functions in the post-training phase (Liu et al., 2023b) or training on data from Monte Carlo sampling (Lee et al., 2024).

- **E3: Likelihood of actions.** The likelihood-based approach utilizes the generator's probability of conducting a specific action (i.e., the tree node) to estimate the node value (Hao et al., 2023; Gao et al., 2024b).

- **E4: Self-consistency score.** The frequency of intermediate nodes can represent the model's confidence in them, thus being used for evaluation (Qi et al., 2024; Zhuang et al., 2023; Zhou et al., 2023). LATS (Zhou et al., 2023) combines the self-generated LLM score and the self-consistency score for node value. rStar (Qi et al., 2024) utilizes the self-consistency score as the reward for the terminal node.

- **E5: Roll out.** In algorithms like MCTS, the intermediate node value can be estimated by rollout and further updated based on backup from terminal states (Qi et al., 2024). The reward for terminal states can come from external tools like code interpreters or the aforementioned evaluation methods.

**Search Algorithm**   The search algorithm defines the operational rules for tree nodes. It can be instantiated as follows:

- **A1: Beam search.** For the beam search algorithm, it generates $k$ candidate nodes at each layer and selects the most promising $m$ nodes from them based on node values (Gu et al., 2023; Yao et al., 2023a; Xie et al., 2023).

- **A2: A\* algorithm.** For the A\* algorithm (Hart et al., 1968), it calculates the sum of the cumulative cost (i.e., the cost from the root node to the current node) and the future score (i.e., the cost of the path from the current node to the goal) in the search process and always selects the node with minimum value. ToolChain\* (Zhuang et al., 2023) mainly relies on the heuristic function to calculate the score, and Q\* (Wang et al., 2024b) utilizes the process reward model and roll out method for estimation.

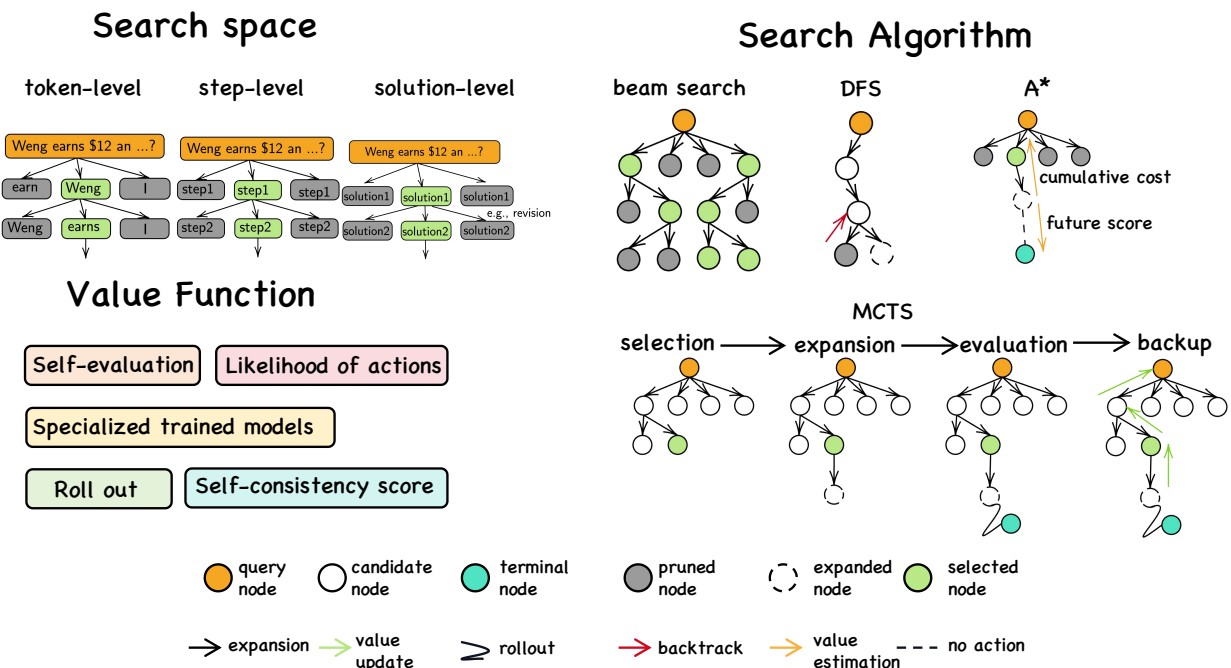

Figure 6: Illustration of key components of tree search.

- **A3: Depth-first search (DFS).** The DFS algorithm explores the most promising node first until the node is no longer promising or the final output is reached, then backtracks to the parent node to explore alternative thoughts (Yao et al., 2023a). Long (2023) implement the DFS algorithm in the sudoku puzzle, where a checker module utilizes the sudoku rules to check the validity of the partial solution and a controller controls the backtracking behaviors of LLMs.

- **A4: Monte carlo tree search (MCTS).** A line of work proposes using the advanced MCTS algorithm to improve the planning ability of LLMs (Zhang et al., 2023; Hao et al., 2023; Liu et al., 2023b), considering its success in AlphaGo (Silver et al., 2016). The key implementation of MCTS lies in four processes: selection, expansion, evaluation, and backup (see Figure 6 for the illustration). The selection phase traverses the tree from the root, iteratively selecting the most promising child nodes by balancing exploration and exploitation, until reaching an underexplored node. Widely used methods like Upper Confidence Bound applied on Trees Algorithm (UCT) (Kocsis & Szepesvári, 2006) and Predictor + UCT (Rosin, 2011) (PUCT) guide this choice, favoring nodes with high values while adjusting for visit frequency to prevent overconcentration on heavily explored nodes. The expansion operation grows the tree by adding one or more new child nodes to the underexplored node based on possible actions from the current node's state. The evaluation process evaluates a new child node by methods described in previous value function paragraph, such as rollouts to a terminal state or direct evaluation with LLMs. In backup, the evaluation result is propagated upward along the selected path, updating the values and visit counts of all nodes traversed.

Table 1 presents an organization of works on tree search based on the established taxonomy. Additionally, Table 9 presents more works applying tree search across various domains.

### 4.2.2 Scaling Laws

Empirical results show that performance can be further enhanced by scaling the breadth and depth of tree search. This includes increasing the number of rollouts in the MCTS algorithm (Zhang et al., 2023; Liu et al., 2023b; Zhang et al., 2024b; Qi et al., 2024), the beam size in beam search (Yao et al., 2023a; Xie et al., 2023), and the step limitations in the A* algorithm (Zhuang et al., 2023). Snell et al. (2024) analyze the scaling

Table 1: An organization of works on tree search. This table includes inference-only approaches, while work combining training strategies is discussed in §5.3. Under **Value Function**, **E1** denotes self-evaluation, **E2** denotes specialized trained models, **E3** denotes likelihood of actions, **E4** denotes self-consistency score, **E5** denotes roll out.

| Work | Application | Search Space | Value Function | Search Algorithm |
|------|-------------|--------------|----------------|------------------|
| Pangu (Gu et al., 2023) | Knowledge Base QA | Step | E2 | Beam Search |
| PG-TD (Zhang et al., 2023) | Code | Token | E5 | MCTS |
| ToT (Yao et al., 2023a) | Game of 24, Writing, Crosswords | Step | E1 | BFS, DFS |
| GuidedDecoding (Xie et al., 2023) | Math | Step | E1 | Beam Search |
| RAP (Hao et al., 2023) | Reasoning | Step | E1, E3, E4 | MCTS |
| PPO-MCTS (Liu et al., 2023b) | Alignment | Token | E2 | MCTS |
| GoT (Besta et al., 2024) | Reasoning | Step | E1 | BFS, DFS |
| LATS (Zhou et al., 2023) | Programming, Reasoning | Step | E1, E4, E5 | MCTS |
| ToolChain* (Zhuang et al., 2023) | Tool-use, Reasoning | Step | E1, E3, E4 | A* |
| MindStar (Kang et al., 2024a) | Math | Step | E2 | BFS |
| Q* (Wang et al., 2024b) | Math, Code | Step | E2, E5 | A* |
| LiteSearch (Wang et al., 2024a) | Math | Step | E2 | BFS |
| MCTSr (Zhang et al., 2024b) | Math | Solution | E1 | MCTS |
| REBASE (Wu et al., 2024c) | Math | Step | E2 | BFS |
| SearchAgent (Koh et al., 2024) | Web agents | Step | E1 | A* |
| rStar (Qi et al., 2024) | Math | Step | E4, E5 | MCTS |
| PLANSEARCH (Wang et al., 2024c) | Code | Step | - | Beam Search |
| RethinkMCTS (Li et al., 2024e) | Code | Step | E1, E5 | MCTS |
| SC-MCTS* (Gao et al., 2024b) | Blocksworld | Step | E1, E3 | MCTS |
| LLaMA-Berry (Zhang et al., 2024c) | Math | Solution | E2 | MCTS |
| ETS (Hooper et al., 2025) | Math | Step | E2 | BFS |

behavior and finds that performance will eventually saturate. This may be due to the model struggling to produce diverse nodes as the number of sampled candidates increases. Additionally, Kang et al. (2024a) find that increasing the model size of the PRM employed in the search algorithm can enhance performance. This highlights the importance of improving the reliability of value functions through extra training time or test-time compute.

### 4.2.3 Improving Scaling Efficiency

The strategies to improve the scaling efficiency of tree search are as follows:

**Selecting appropriate tree search algorithms** The characteristics of different tree search algorithms make them suitable for different tasks. For code generation tasks, PG-TD (Zhang et al., 2023) compares MCTS using public test cases for terminal state evaluation against simple beam search, finding that MCTS achieves significantly better performance given the same computation time. ToolChain* (Zhuang et al., 2023) demonstrates that the A* algorithm is more time-efficient than MCTS and other alternatives in API call tasks.

**Reducing the overhead of value functions** For algorithms like MCTS, reliable value functions traditionally rely on multiple rollouts, which incur significant computational costs. ALPHALLM (Tian et al., 2024) employs a smaller language model as a fast rollout policy to reduce computational overhead. rStar-Math (Guan et al., 2025b) introduces a two-phase approach: first estimating node values through rollouts, then using this data to train a separate value function that replaces rollouts in subsequent iterations.

**Adaptive expansion breadth** Traditional beam search algorithms fix the expansion breadth of each node to a constant value, which may not optimally balance exploration and exploitation. LiteSearch (Wang et al., 2024a) allocates expansion breadth based on node value and depth, encouraging exploration on high-value nodes and at the beginning of the search process. This approach helps achieve higher token efficiency than beam search and DFS. REBASE (Wu et al., 2024c) takes a similar strategy by defining trajectory collection requirements and dynamically allocating expansion breadth at each depth based on node value and remaining collection needs, resulting in higher efficiency compared to traditional MCTS algorithms.

Table 2: An organization of works on multi-turn correction. **Fine-tuning** indicates whether the method requires additional training. ✓ in the **Self-feedback** and **Self-refinement** columns represents that it shares the same parameters with the initial generator but is prompted with different roles.

| Work | Feedback | | Refinement | | Fine-tuning |
|------|----------|--|------------|--|-------------|
| | Self-feedback | External | Self-refinement | External | |
| Self-Correction (Welleck et al., 2023) | ✗ | ✗ | ✗ | Trained LM | ✓ |
| Self-refine (Madaan et al., 2023) | ✓ | ✗ | ✓ | ✗ | ✗ |
| Reflexion (Shinn et al., 2023) | ✓ | Game Envs; Interpreter; Oracle | ✓ | ✗ | ✗ |
| RCI (Kim et al., 2023) | ✓ | Oracle | ✓ | ✗ | ✗ |
| Self-Debug (Chen et al., 2023c) | ✓ | Interpreter | ✓ | ✗ | ✗ |
| Baldur (First et al., 2023) | ✗ | Proof checker | ✗ | Trained LM | ✓ |
| REFINER (Paul et al., 2024) | ✗ | Trained LM | ✗ | Trained LM | ✓ |
| LLM-Debate (Du et al., 2023) | ✓ | ✗ | ✓ | ✗ | ✗ |
| MAD (Liang et al., 2023) | ✓ | ✗ | ✓ | ✗ | ✗ |
| CRITIC (Gou et al., 2023) | ✓ | Search engine; Interpreter | ✓ | ✗ | ✗ |
| CoVe (Dhuliawala et al., 2023) | ✓ | ✗ | ✓ | ✗ | ✗ |
| RISE (Qu et al., 2024) | ✗ | ✗ | ✓ | ✗ | ✓ |
| IHR (Qiu et al., 2023) | ✗ | Interpreter | ✓ | ✗ | ✗ |
| SCoRe (Kumar et al., 2024) | ✗ | ✗ | ✓ | ✗ | ✓ |
| AutoMathCritique (Xi et al., 2024) | ✗ | Trained LM | ✗ | Trained LM | ✓ |
| DARS (Li et al., 2025d) | ✗ | Trained LM | ✗ | Trained LM | ✓ |

**Reducing redundant expansion nodes** In the node expansion process, there may exist nodes with semantically equivalent content, leading to unnecessary exploration costs. FETCH (Wang et al., 2025a) and ETS (Hooper et al., 2025) merge semantically similar nodes using agglomerative clustering of text embeddings obtained from a fine-tuned model, achieving higher token efficiency.

### 4.3 Multi-turn Correction

#### 4.3.1 Key Components

Multi-turn correction aims to improve response quality through iterative revision. It consists of an initial generator $g_0$ that proposes the initial response, a feedback model $f$ that generates feedback for the latest response, and a refinement model $g$ that revises the response given the interaction history (Welleck et al., 2024):

$$y^0 \sim g_0(y|x) \tag{7}$$

$$z^t \sim f(z|x, y^{(<t)}, z^{(<t)}) \tag{8}$$

$$y^t \sim g(y|x, y^{(<t)}, z^{(\leq t)}) \tag{9}$$

where $x$ represents the query, $y^t$ represents the response at timestep $t$, and $z^t$ represents the feedback at timestep $t$. The system outputs the final response when a stopping condition is met. The feedback generation stage can be omitted, resulting in direct refinement of the initial response (Welleck et al., 2023; Kamoi et al., 2024).

Multi-turn correction imitates human reflection and refinement cognitive processes. The core design of it lies in constructing reliable feedback signals and refinement models to improve response quality. Feedback sources can be categorized as follows (Pan et al., 2023):

- **F1: Self-feedback.** The initial generator $g_0$ and the feedback model can share a single language model, resulting in self-feedback. For example, Self-Debug (Chen et al., 2023c) instructs $g_0$ to explain code line by line and generate execution traces as feedback signals. Self-Refine (Madaan et al., 2023) incentivizes $g_0$ to generate feedback using reflective prompts. Moreover, $g_0$ can be prompted with different roles to encourage divergent thinking (Du et al., 2023; Liang et al., 2023; Khan et al., 2024), a technique known as "multi-agent debate."

- **F2: External feedback.** The feedback can come from external sources to $g_0$, including: 1) external tools: such as code interpreters (Chen et al., 2023c; Gou et al., 2023; Shinn et al., 2023), proof

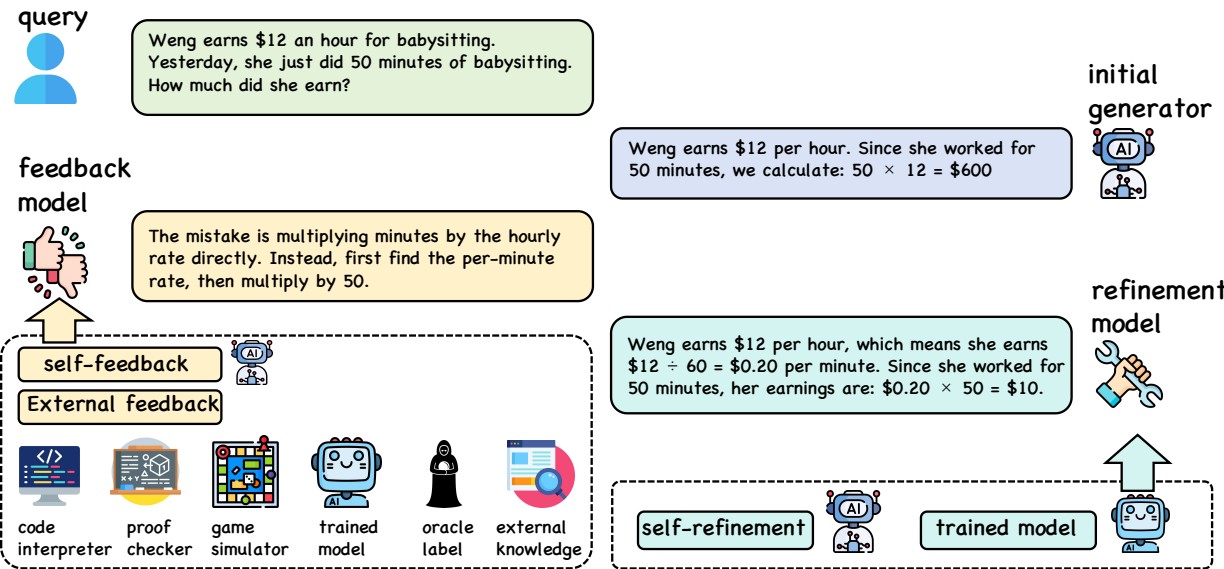

Figure 7: Illustration of key components of multi-turn correction.

checkers (First et al., 2023), game simulators (Shinn et al., 2023); 2) external knowledge (Gou et al., 2023; Zhao et al., 2023a); 3) oracle labels: such as ground truth answers to math problems (Shinn et al., 2023), though these are not guaranteed to be available in real-world applications (Huang et al., 2023b); 4) specialized trained models (Paul et al., 2024; Xi et al., 2024).

The refinement model can also be instantiated similarly to the feedback models, including self-refining the response (Madaan et al., 2023; Shinn et al., 2023), or using a specialized trained model (Welleck et al., 2023). Specifically, Self-Refine (Madaan et al., 2023) instantiates the initial generator, feedback model, and refinement model with the same language model. Table 2 presents an organization of works on multi-turn correction based on the established taxonomy. Furthermore, Table 9 showcases more studies that implement multi-turn correction techniques across diverse application domains.

Research demonstrates that with reliable external feedback, multi-turn correction significantly enhances model performance across diverse tasks (Kamoi et al., 2024). However, this approach has faced criticism because high-quality external feedback is often unavailable in real-world scenarios (Huang et al., 2023b). In the *intrinsic self-correction* setting, where a model critiques and revises its own responses without external feedback, empirical studies indicate that LLMs generally struggle to generate reliable critiques and revisions, particularly in planning (Valmeekam et al., 2023; Stechly et al., 2023) and reasoning (Huang et al., 2023b; Tyen et al., 2023) tasks, leading to little or no performance gains. It has also been observed that self-biases can amplify during the self-correction process (Xu et al., 2024c).

### 4.3.2 Scaling Laws

In tasks with reliable external feedback or correctors, performance can be further improved by scaling the number of revision steps until it finally saturates (Welleck et al., 2023; Madaan et al., 2023; Du et al., 2023; Qiu et al., 2023). In intrinsic self-correction settings where the model lacks critique ability, increasing the revision steps can harm performance (Welleck et al., 2023; Huang et al., 2023b). This limitation can be addressed through additional training to improve self-correction ability (Qu et al., 2024; Snell et al., 2024). Snell et al. (2024) observe that after fine-tuning the model to improve its correction ability, performance grows steadily as revision steps increase and eventually saturates, even beyond the revision number used during training. This highlights that additional investment in training compute before deploying multi-turn correction can expand the ceiling of scaling test time.

### 4.3.3 Improving Scaling Efficiency

As discussed above, the effectiveness of multi-turn correction is constrained by the reliability of feedback and the model's refinement capabilities. To enhance scaling efficiency, efforts should concentrate on developing high-quality feedback mechanisms or improving the model's refinement abilities.

**Constructing high-quality feedback** Reliable feedback is often limited to specific task types. Several strategies can improve feedback quality for broader applications: 1) Using reference-free LLM-based evaluation metrics with human-written evaluation criteria (Chiang & Lee, 2023; Liu et al., 2023c); 2) Employing task-specific decomposition (Saha et al., 2024a) to break down complex verification into manageable subtasks; 3) Leveraging confidence estimation through generation probabilities (Varshney et al., 2023) or prompting techniques (Li et al., 2024d); 4) Fine-tuning models specifically for feedback (Welleck et al., 2023; Paul et al., 2024; Xi et al., 2024).

**Improving the refinement ability of LLMs** Instead of focusing solely on constructing high-quality feedback, this line of work directly improves the refinement ability of models through additional training-time compute without the feedback models. RISE (Qu et al., 2024) generates synthetic multi-turn correction data by concatenating the incorrect response before the final correct response and fine-tunes the model on these examples to improve its refinement ability. SCoRe (Kumar et al., 2024) further identifies the behavior collapse issues in the Supervised Fine-tuning (SFT) method and proposes multi-turn RL training with carefully designed rewards for different turns.

## 4.4 Long CoT

### 4.4.1 Key Components

CoT prompting (Wei et al., 2022; Nye et al., 2021) instructs models to generate human-readable explanations of how problems are solved. This approach can help improve models' representational complexity (Merrill & Sabharwal, 2023; Nowak et al., 2024) and significantly enhances their performance in reasoning tasks (Wei et al., 2022). Current models like ChatGPT or Llama 3.1 default to CoT when presented with reasoning problems (Sprague et al., 2024). Despite its widespread application, the reasoning process in CoT is usually shallow and linear, revealing limitations in complex cognitive capabilities (Kambhampati, 2024; Chen et al., 2025e). Recently, models like OpenAI o1 (OpenAI, 2024) or Deepseek R1 (DeepSeek et al., 2025) have advanced the traditional CoT into long CoT, which incorporates more sophisticated thinking patterns and extended responses. The cognitive patterns present in long CoT but typically less observed in traditional CoT are as follows:

- **Reflection:** The model develops metacognitive abilities (Metcalfe & Shimamura, 1994) to assess the correctness and rationality of its own responses. For example, the model may pause its reasoning by outputting "wait" when it detects potential issues.

- **Backtracking:** When the model detects an error in its response, it can return to previous steps and revise them. This capability is vital for long-horizon planning problems, such as sudoku and code-breaking. In these problems, the model must find optimal solutions among multiple possibilities, and since the initial solution is not guaranteed to be correct, the model needs to employ trial and error.

- **Verification:** The model learns to recheck both individual steps and complete solutions, which enhances the robustness of its problem-solving approach.

- **Divergent thinking:** When the model recognizes that the current solution cannot solve the problem or leads to an obviously wrong answer, it can employ divergent thinking to explore alternative solutions, often signaled by transitional phrases like "alternatively."

- **Internal thinking:** The model can generate human-like thinking processes beyond explicit problem-solving steps. This enables more fine-grained reasoning before generating each subsequent step, thereby improving its overall performance (Wu et al., 2024a).

### 4.4.2 Scaling Laws

Early work demonstrates that extending reasoning steps significantly enhances LLMs' reasoning capabilities (Jin et al., 2024). In the context of long CoT models, recent studies have identified a positive correlation between token count and model performance. Although not explicitly describing token control methodologies, OpenAI (2024) and DeepSeek et al. (2025) discover that performance increases with token count following a log-linear relationship. More transparent research by Hou et al. (2025) and Muennighoff et al. (2025) applies response post-processing or decoding techniques to regulate token count, revealing a positive association between response length and performance. Specifically, Hou et al. (2025) truncate responses to varying lengths from the beginning and suggest using a summarization model to extract final answers. Muennighoff et al. (2025) develop a budget-forcing technique to control token count through the addition or suppression of end-of-thinking token delimiters.

Despite these studies providing substantial evidence for the positive correlation between token number and model performance, debate on the effectiveness of extensive response length remains. These debates primarily stem from observations that shorter response lengths yield higher accuracy than longer responses (Zeng et al., 2025b; Ballon et al., 2025). This phenomenon may be explained by models generating more tokens for more challenging problems where failure risks are higher, or by approaches chosen in longer responses being more convoluted than those in shorter responses, thus increasing the likelihood of failure (Fatemi et al., 2025).

### 4.4.3 Improving Scaling Efficiency

Although long CoT endows models with deep thinking abilities, it can lead to overthinking problems. For instance, models might generate hundreds of tokens for simple questions like "2+3=5" (Chen et al., 2024d), where the correct answer is reached early but followed by unnecessary reasoning. Furthermore, CoT-based methods operate in the language space, allocating similar computational resources to each token regardless of its importance. This uniform allocation is suboptimal since some tokens, like those maintaining text coherence, require minimal planning, while others crucial to the reasoning process demand more intensive processing (Hao et al., 2024b). We detail techniques to resolve these issues below.[7]

**Prompting for conciseness**  This approach directly instructs models to limit response tokens to a specific number (Nayab et al., 2024; Xu et al., 2025c) or capture only essential information (Ding et al., 2024; Aytes et al., 2025) through prompting. Although straightforward to implement, its effectiveness is limited to simple tasks, and LLMs cannot strictly adhere to token number restrictions (Muennighoff et al., 2025; Aggarwal & Welleck, 2025).

**Finetuning on compressed responses using heuristic methods**  This approach first compresses CoT responses using heuristic methods and then finetunes on them. The heuristic compression techniques include directly removing intermediate steps (Su et al., 2024; Deng et al., 2024), assessing token importance in CoT through perplexity (Cui et al., 2025b) or a specifically trained model (Xia et al., 2025) to retain only the most relevant tokens, and leveraging advanced models like GPT-4 to reconstruct CoT sequences while preserving essential information and eliminating redundancy (Kang et al., 2024b). The effectiveness of this method heavily depends on the design of the heuristic compression techniques, limiting its generalizability across tasks. For example, Conditioned Compressed Chain-of-Thought (C3oT) (Kang et al., 2024b) finds that training directly on GPT-4-compressed data significantly degrades task performance, necessitating the inclusion of original uncompressed data during training.

**Query-aware compression**  The compression limitation of response length varies based on query type (Lee et al., 2025; Arora & Zanette, 2025), as difficult problems require more tokens while easier ones require fewer. This method aims to approach the limitation in a query-aware way, which helps improve token efficiency while maintaining or improving the models' adaptivity in computational resource allocation. The methods are as follows:

---

[7]It is noted that some of the work focuses on traditional CoT instead of long CoT, but we include them considering their easy generalization.

- **Learning on trajectories with predefined optimal length:** This approach first determines optimal length explicitly and trains on them. The reference for the optimal length can be based on task types. For example, in the SFT phase of DeepSeek-R1 (DeepSeek et al., 2025), for reasoning tasks they collect long CoT responses for training, while for non-reasoning tasks they collect CoT responses for certain tasks and even no-CoT responses for simpler queries. These approaches help the model learn to switch reasoning modes based on the query type. Beyond this, other estimations for the optimal length can be based on search (Han et al., 2024; Yang et al., 2025a), prompting (Han et al., 2024), or query difficulty estimated by sampling (Shen et al., 2025b). These selected optimal-length trajectories can be used for further SFT or Direct Preference Optimization (DPO) training.

- **Self-training:** Instead of predefining the optimal length, this method first rolls out trajectories from the generator and incentivizes the model to achieve fewer tokens while maintaining accuracy through self-training, which can be considered an on-policy optimal-length estimation. The training methods can be: 1) SFT: This approach generates multiple responses for each question and selects the shorter correct ones for SFT (Kimi et al., 2025; Munkhbat et al., 2025; Liu et al., 2024d); 2) DPO: This method uses the long-CoT model to generate multiple response samples, selecting the shorter correct solution as the positive sample while treating longer responses as negative samples. These positive-negative pairs form the pairwise preference data used for preference learning. For preference data construction, Chen et al. (2024d) find that choosing responses including two solving attempts that reach the correct answer as positive examples performs best. Sky-T1-32B-Flash (Li et al., 2024f) employs multiple preference data construction methods to avoid accuracy drops while reducing reasoning length. For the training algorithm, Chen et al. (2024d) empirically demonstrate that SimPO (Meng et al., 2024) performs better than DPO. 3) RL: This approach adds a length penalty in the reward function to reduce response length (Aggarwal & Welleck, 2025; Kimi et al., 2025; Luo et al., 2025a; Arora & Zanette, 2025) or designs dense reward in the intermediate steps (Qu et al., 2025b). For example, L1 (Aggarwal & Welleck, 2025) adds a length control factor in the RL reward function to train the model to adhere to the length given in the prompt or not exceed the maximum length. Furthermore, MRL (Qu et al., 2025b) measures progress at each intermediate generation episode through on-policy rollouts and develops corresponding SFT and RL methods for maximizing dense rewards based on the progress. While there is no explicit length-relevant factor in the algorithm, it helps the model balance exploration and exploitation in the content of CoT and improve token efficiency.

- **Query router:** This method classifies queries as difficult or easy and handles them differently by applying different types of models (Saha et al., 2024b) or different computation budgets of the same model (Fu et al., 2024). For example, System-1.x (Saha et al., 2024b) trains a controller that decomposes a planning problem into sub-goals and classifies them as easy or hard to be solved by either the System-1 planner or the System-2 planner.

**Model merging**  This method combines a long-CoT model with a short-CoT model to create a new model without additional training. CoT-Valve (Ma et al., 2025b) manipulates the weights between the parameters of the two models to achieve varying lengths.

**Compressing the intermediate state**  In the response generation process, the storage overhead of the KV cache increases linearly with the context length for the Transformer architecture. This line of work aims to compress intermediate steps into a shorter form and reason starting from it, continuing the compressing and generation process in the decoding phase. It helps to reduce the number of tokens stored in the context window, thereby lowering memory overhead and computational costs. This includes compressing intermediate steps into a summary (Yan et al., 2025), a subquestion (Yang et al., 2024c), or a special token (Pang et al., 2024a; Zhang et al., 2025a) through specific training and corresponding inference strategies.

**Reasoning in the latent space**  Switching reasoning from the language space to other spaces like latent space may overcome the restrictions of language and improve token efficiency. This can be achieved by finetuning existing models to possess this capability (Deng et al., 2023; Hao et al., 2024b; Shen et al., 2025c)

Table 3: Comparisons of test-time scaling methods. Gray color represents the model is optional or can share the same parameters with others. The description of these features is for the standard version.

| Method | Required Model | Controllability | Adaptivity | Training-free | Compatibility |
|---|---|---|---|---|---|
| Parallel Sampling | Generator
Scoring function | Coarse-grained | Not supported | ✓ | Full |
| Tree Search | Generator
Value function | Coarse-grained | Partial supported | ✓ | Full |
| Multi-turn Correction | Initial generator
Feedback model
Refinement model | Coarse-grained | Partial supported | ✓ | Full |
| Long CoT | Long-CoT model | Not supported | Supported | ✗ | Full |

or developing new language model architectures capable of implicitly reasoning in latent space (Geiping et al., 2025).

### 4.5 Comparisons of Test-Time Scaling Methods

For different test-time scaling methods, we summarize their characteristics in Table 3. Specifically, we focus on the following aspects:

**Performance**    *What is the optimal test-time scaling method given the same computation budget?* Establishing an absolute ranking of test-time scaling methods is challenging due to the various implementations within each approach and the difficulty in ensuring fair comparisons. For performance ceiling, long CoT methods consistently outperform other test-time scaling approaches that are based on traditional LLMs, particularly for olympic-level problems (OpenAI, 2024; DeepSeek et al., 2025). Moreover, different test-time scaling methods exhibit distinct advantages for problems of varying difficulty and under different computational constraints. For instance, Snell et al. (2024) empirically demonstrate that beam-search excels on complex questions when operating under limited computation budgets, whereas BoN sampling achieves superior performance on simpler questions when greater computational resources are available. These complementary strengths create opportunities for ensemble methods, which will be discussed in subsequent sections.

**Cognitive behaviors**    *Which test-time scaling method exhibits the most human-like cognitive behaviors?* Long CoT exhibits the most cognitive behaviors compared to others, including reflection, backtracking, divergent thinking, etc. More importantly, it unifies these cognitive behaviors in the generation process, enabling greater flexibility. Methods like tree search and multi-turn correction rely on external tree search algorithms or predefined multi-turn correction frameworks to endow the model with planning or reflection cognitive capability, limiting their adaptation to specific problems.

**Adaptivity**    *Can the test-time scaling method allocate different computational resources to different queries?* The degree of adaptivity of a test-time scaling method depends on its stopping condition. In parallel sampling approaches, the standard implementation assigns identical sampling numbers across all queries, resulting in a lack of adaptivity. For tree search and multi-turn correction approaches, different cases exist. One variant of methods stops once reaching predefined hyperparameters (e.g., correction numbers, tree depth) or the answers (Yao et al., 2023a; Kang et al., 2024a; Snell et al., 2024), thus providing no additional adaptivity from the framework. Another line of methods incorporates verifiers in the stopping condition, such as requiring the quality score of outputs to exceed a given threshold (Wang et al., 2024a; Welleck et al., 2023), which introduces adaptivity based on the reliability of these verifiers. For example, LiteSearch (Wang et al., 2024a) observes that tree search algorithms allocate larger computational resources for harder problems where stopping conditions include verifier values. For long CoT methods, the stopping condition is implicit and inherent in the generation process. Recent studies observe that the long CoT

model generates longer responses to more challenging problems (Zeng et al., 2025b). From the perspective of generalization, long CoT is the most promising approach for differentially allocating computational resources.

**Controllability**  *Given a computation budget, can it operate within the specified constraints?* For test-time scaling methods with externally controllable scaling dimensions (e.g., sampling numbers, tree depth, revision steps), coarse-grained controllability can be achieved by mapping the computation budget to specific quantities of these hyperparameters according to empirical estimation (Welleck et al., 2024). For long CoT, although directly truncating the response to a specific number ensures not exceeding the computation budget, the resulting incomplete response significantly harms performance, making it impractical to consider standard long CoT as a method with controllability. To address this limitation, S1 (Muennighoff et al., 2025) achieves control of response length through the implementation of end-of-thinking token delimiters, while L1 (Aggarwal & Welleck, 2025) develops a RL algorithm to achieve precise control over token number with higher token efficiency compared to S1.

**Simplicity**  *Are the components of the test-time scaling method straightforward to implement?* Methods excluding long CoT usually require additional roles such as evaluators to guide the search process and multiple processes to derive the final solutions. Considering the extra cost to deploy high-quality evaluators for most tasks, this may hinder their practical application. In contrast, the long CoT method eliminates the need for multiple components and is straightforward to implement.

**Training-free**  *Does the test-time scaling method require additional training?* Methods excluding long CoT can be operated with the traditional LLM directly, while the long CoT ability needs to be elicited with additional training. It is notable that additional training can help improve scaling efficiency across methods, such as enhancing the self-correction ability of models (Kumar et al., 2024; Qu et al., 2024) or applying inference-aware fine-tuning to improve computation utilization (Chow et al., 2024; Yu et al., 2025c).

**Compatibility**  *Can this method be integrated with other test-time scaling methods?* As will be discussed in §4.6, all methods can be compatible with each other. Among them, parallel sampling is most easily compatible with others considering the ease of implementing multiple sampling.

Overall, the long CoT test-time scaling method outperforms others with its simplicity, adaptivity, higher ceiling performance, and more complex cognitive behaviors, but it requires additional training to elicit. Moreover, the compatibility and advantages of these test-time scaling methods make it beneficial to comprehensively utilize them together to achieve better performance instead of focusing on a single method.

### 4.6  Ensemble of Test-Time Scaling Methods

Ensemble methods aim to comprehensively utilize multiple test-time scaling approaches rather than allocating computational resources to a single method, potentially achieving superior performance compared to individual approaches. These include simultaneously combining multiple methods or selecting appropriate test-time scaling methods according to different contexts. Figure 8 presents an organization of works on ensemble methods.

**Combining parallel sampling with other methods**  The simplicity of parallel sampling facilitates compatibility with other test-time scaling methods:

- **Tree search.** Instead of searching along a single tree, parallel sampling can enhance tree search diversity by expanding the initial set of beams into multiple independent subtrees that are searched independently (Beeching et al., 2024; Bi et al., 2024). Empirical results demonstrate improved tree search performance, especially at large computation budgets. Moreover, tree search algorithms can also help accelerate parallel sampling. For example, TreeBON (Qiu et al., 2024) reduces the computational overhead of BoN by using tree search to prune low-quality responses at an early stage.

- **Multi-turn correction.** Parallel sampling functions as a global search by generating responses independently in parallel, whereas multi-turn correction operates as a local search on the initial

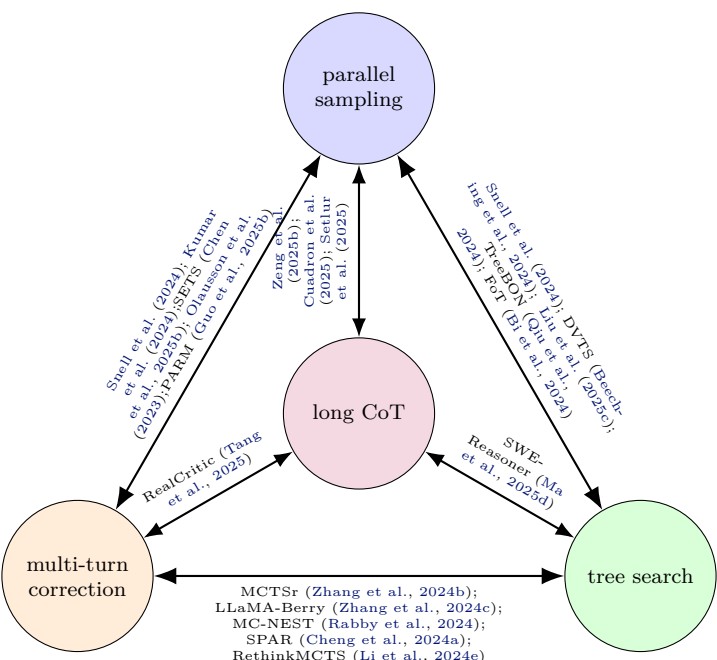

Figure 8: Ensemble of Test-Time Scaling Methods (§4.6). Solid lines represent combination work between two connected methods.

response (Snell et al., 2024). This complementarity indicates that combining the two methods can yield better performance. Kumar et al. (2024); Chen et al. (2025b) show that allocating a portion of the computation budget to self-correction of the initial response rather than solely increasing the sampling number can achieve higher token efficiency. Olausson et al. (2023) demonstrate that in the mixed scheme, allocating more of the sampling budget to generating a diverse set of initial candidates is more optimal than carrying out extensive correction.

- **Long CoT.** Combining long CoT with parallel sampling methods such as majority voting is straightforward. Recent research has optimized majority voting strategies by considering the overthinking phenomenon in long CoT (Zeng et al., 2025b; Cuadron et al., 2025). Specifically, high overthinking correlates with decreased performance in math tasks or agentic environments. Therefore, integrating metrics that measure the degree of overthinking with voting strategies can outperform both majority voting and single high-computation-cost response generation (Zeng et al., 2025b; Cuadron et al., 2025). Additionally, Setlur et al. (2025) compare the performance of scaling long-CoT length by budget-forcing (Muennighoff et al., 2025) against applying the computation for parallel sampling with shorter responses, finding the latter to be more compute-optimal.

**Combining tree search with multi-turn correction**  This line of work incorporates critique and revision into the tree search algorithm by treating the revision behavior as the action to update the response, at either the solution level (Zhang et al., 2024b;c; Rabby et al., 2024; Cheng et al., 2024a) or the step level (Li et al., 2024e). This approach enriches the expansion behavior of tree search and helps achieve better performance than simply revising responses sequentially (Li et al., 2024e; Cheng et al., 2024a).

**Combining long CoT with tree search or multi-turn correction**  The content of long CoT implicitly contains branch search processes or self-correction (Xiang et al., 2025). Thus, it can be viewed as a method that internalizes these two approaches. For the combination with multi-turn correction, Tang et al. (2025) show that o1-mini benefits from self-correction while traditional LLMs perform worse, demonstrating that long CoT models possess strong intrinsic self-correction ability. For tree search, future research should analyze how to define the search space within the long thinking processes.

**Adaptive selection of test-time scaling methods** Empirical analysis of different test-time scaling methods' performance relative to various factors can help derive optimal test-time scaling methods based on adaptive selection. Snell et al. (2024) find that multi-turn correction methods are better suited for simpler queries, while a certain ratio of parallel sampling and multi-turn correction is appropriate for difficult queries. Moreover, they determine that beam search is more effective for harder questions whereas BoN is more effective for easier questions. These findings guide optimal test-time scaling strategies based on query difficulty classifiers. Liu et al. (2025c) analyze the relationship between model size and test-time scaling methods to derive an optimal scaling strategy.

# 5 How – Part II: Training Strategies for Test-Time Scaling

As discussed in §4.5, long CoT demonstrates higher ceiling performance and more complex cognitive behaviors compared to other test-time scaling strategies, though it requires additional training. In this section, we examine methods to elicit the model's long CoT capabilities through two primary approaches: RL (§5.1) and SFT (§5.2). Additionally, we discuss how to effectively combine test-time scaling techniques with iterative training methodologies to achieve self-improvement (§5.3).

## 5.1 Scaling Reinforcement Learning

Recent research demonstrates that training LLMs through online RL with rule-based rewards in tasks like mathematics and code can significantly enhance their reasoning abilities (DeepSeek et al., 2025; Kimi et al., 2025). During the training process, models autonomously learn to master long-CoT test-time scaling methods to solve challenging problems and demonstrate cognitive behaviors including self-reflection and self-correction. This phenomenon has been described as the RL scaling phenomenon[8] or the "Aha moment" (DeepSeek et al., 2025; Liu et al., 2025g). We systematically summarize recent works in Table 4. Additionally, Table 5 presents recipes to address common challenges in RL scaling training based on recent studies. In the following sections, we detail the design considerations for each component.

### 5.1.1 Training Algorithm

**REINFORCE** The REINFORCE (Sutton et al., 1999) algorithm is a foundational policy gradient method in RL that directly optimizes the expected return of a policy through gradient ascent. The algorithm optimizes the policy model $\pi_\theta$ by minimizing the loss:

$$\mathcal{L}_{\text{REINFORCE}}(\theta) = -\mathbb{E}_{\tau \sim \pi_\theta} \left[ \sum_{t=1}^{T} G_t \nabla_\theta \log \pi_\theta(a_t|s_t) \right] \tag{10}$$

where $G_t$ is the discounted cumulative reward from time step $t$. Despite its simplicity, REINFORCE suffers from high variance in gradient estimates.

**Proximal Policy Optimization (PPO)** For the PPO algorithm (Schulman et al., 2017), it optimizes the policy model by minimizing the loss:

$$\mathcal{L}_{\text{PPO}}(\theta) = -\mathbb{E}_{q \sim P(Q), o \sim \pi_{\theta_{\text{old}}}(O|q)} \frac{1}{|o|} \sum_{t=1}^{|o|} \min \left( \frac{\pi_\theta(o_t|q, o_{<t})}{\pi_{\theta_{\text{old}}}(o_t|q, o_{<t})} A_t, \text{clip}(\theta) A_t \right) \tag{11}$$

$$\text{clip}(\theta) = \text{clip} \left( \frac{\pi_\theta(o_t|q, o_{<t})}{\pi_{\theta_{\text{old}}}(o_t|q, o_{<t})}, 1 - \varepsilon, 1 + \varepsilon \right) \tag{12}$$

where $\pi_\theta$ and $\pi_{\theta_{\text{old}}}$ are the current and old policy models, and $q, o$ are the sampled questions and outputs. The clip($\theta$) function constrains policy updates to ensure stable training. $A_t$ is the advantage computed by applying GAE (Schulman et al., 2016) based on the rewards $\{r_{\geq t}\}$ and a learned value function $V_\psi$. The KL

---

[8]In the paper, we use "RL scaling" to describe the line of work.

Table 4: Summary of recent works on RL scaling. For training algorithms, 'PMD' denotes policy mirror descent method. REINFORCE* denotes REINFORCE-style method. For reward types, 📏 and 🤖 represent rule-based and model-based rewards respectively, while ◎ and 📈 represent outcome and process rewards respectively. '#D' indicates the query dataset size. 'MS' denotes the multi-stage training strategy, including long CoT cold start (LCS), iterative lengthening strategy (ILS), and curriculum sampling strategy (CSS). In accuracy (Acc.) and length (Len.) figures, for works presenting multiple figures, we show the common pattern. "Cog." indicates whether the response contains words indicating cognitive behaviors like "wait."

| Work | Algorithm (§5.1.1) | Reward (§5.1.2) | Series (§5.1.3) | Size (§5.1.3) | #D (§5.1.4) | MS (§5.1.5) | Acc. | Len. | Cog. |
|---|---|---|---|---|---|---|---|---|---|
| Eurus-2-7B-PRIME (Cui et al., 2025a) | REINFORCE* | 📏 🤖 ◎ 📈 | Qwen | 7B | 150K | ✗ | | - | - |
| Deepseek-R1-Zero (DeepSeek et al., 2025) | GRPO | 📏 ◎ | Deepseek | 671B | - | ✗ | | | ✓ |
| Kimi k1.5 (Kimi et al., 2025) | PMD | 📏 ◎ | Kimi | - | - | LCS CSS | | | - |
| SimpleRL-Zero (Zeng et al., 2025a) | PPO | 📏 ◎ | Qwen | 7B | 8K | ✗ | | | ✓ |
| SimpleRL (Zeng et al., 2025a) | PPO | 📏 ◎ | Qwen | 7B | 8K | LCS | | | - |
| STILL-3-ZERO-32B (Chen et al., 2025g) | GRPO | 📏 ◎ | Qwen | 32B | 90K | ILS | | | ✓ |
| Sea AI Lab (Liu et al., 2025f) | PPO | 📏 ◎ | Qwen | 1.5B | 8K | ✗ | | | ✓ |
| DeepScaleR-1.5B-Preview (Luo et al., 2025c) | GRPO | 📏 ◎ | Qwen | 1.5B | 40K | ILS | | 8K 16K 24K | - |
| T1 (Hou et al., 2025) | REINFORCE* | 📏 ◎ | Qwen | 14B | 30K | LCS | | | ✓ |
| DAPO (Yu et al., 2025a) | GRPO | 📏 ◎ | Qwen | 32B | 17K | ✗ | | | ✓ |
| LIMR (Li et al., 2025f) | GRPO | 📏 ◎ | Qwen | 7B | 1.4K | ✗ | | | - |
| Open-Reasoner-Zero (Hu et al., 2025) | PPO | 📏 ◎ | Qwen | 7B 32B | 57K | ✗ | | | ✓ |
| Logic-RL (Xie et al., 2025) | REINFORCE* | 📏 ◎ | Qwen | 7B | 5K | ✗ | | | ✓ |

Table 5: Recipes to resolve common problems in RL scaling training based on recent studies.

| Problem to Solve | Method Overview | Related Studies |
|---|---|---|
| **TRAINING ALGORITHM** | | |
| **Token inefficiency and overthinking in long-form reasoning** | **Dr.GRPO (Doctor GRPO):** Addresses optimization bias in GRPO by removing response-length normalization and reward standardization, implementing an unbiased policy gradient estimation. | Liu et al. (2025g) |
| **Instability with varying response lengths in long-form reasoning** | **DAPO (Decouple Clip and Dynamic Sampling Policy Optimization):** Implements token-level policy gradient calculation, allowing longer sequences to appropriately influence the gradient updates regardless of individual response lengths. | Yu et al. (2025a) |
| **Limited policy exploration due to rigid constraints** | **GPG (Group Policy Gradient):** Simplifies the policy gradient approach by removing reference models and policy constraints while maintaining stability through group-level reward normalization. | Chu et al. (2025b) |
| **Repetitive or narrow reasoning patterns** | **Auxiliary entropy bonus:** Incorporates an additive entropy term into the RL loss function to encourage token diversity and prevent deterministic response patterns. | Hou et al. (2025) |
| **Limitations of fixed reference models** | **On-policy KL normalization:** Combines KL normalization with Exponential Moving Average (EMA) updates to the reference model. | Hou et al. (2025) |
| **Value model misalignment with strong prior policies** | **Value-Pretraining Alignment:** Implements a dedicated pretraining phase for the value model to ensure alignment with strong prior policies before RL begins. | Yuan et al. (2025); Yue et al. (2025b) |
| **Conflicting variance-bias requirements between value and policy optimization** | **Decoupled-GAE (Generalized Advantage Estimation):** Separates the GAE parameter for value function and policy optimization, allowing unbiased value estimation while maintaining variance reduction benefits for policy updates. | Yuan et al. (2025); Yue et al. (2025b) |
| **Limited exploration in constrained policy optimization** | **KL Divergence Removal:** Eliminates the KL penalty term that constrains policy divergence from the reference model, allowing the reasoning policy to explore more freely. | Hu et al. (2025); Yu et al. (2025a) |
| **Premature deterministic behavior in RL systems** | **Clip-Higher Strategy:** Decouples lower and higher clipping ranges in PPO to specifically promote exploration of low-probability tokens while maintaining stability. | Yu et al. (2025a) |
| **Ineffective gradient signals in late-stage training** | **Dynamic Sampling:** Implements an adaptive sampling approach that filters out prompts with accuracy values of exactly 0 or 1 to ensure effective gradient signals. | Yu et al. (2025a); Bae et al. (2025) |
| **Noisy reward signals from length-truncated samples** | **Overlong Filtering:** Masks the loss contribution of truncated samples that exceed maximum length to prevent inappropriate penalization of otherwise sound reasoning. | Yu et al. (2025a) |
| **Inconsistent advantage estimation across variable-length sequences** | **Length-Adaptive GAE:** Dynamically adjusts the $\lambda$ parameter in GAE based on sequence length, ensuring balanced TD-error influence for both short and long outputs. | Yue et al. (2025b) |
| **REWARD DESIGN** | | |
| **Uncontrolled CoT length in reasoning tasks** | **Cosine Length Reward:** Applies a cosine-based reward shaping that prioritizes shorter, correct CoTs while penalizing short, incorrect ones. | Yeo et al. (2025) |
| **Language mixing issues in multilingual environments** | **Language Consistency Incentive:** Calculates rewards based on the proportion of target language words in the CoT to mitigate language mixing issues. | DeepSeek et al. (2025) |
| **Model overthinking and verbosity** | **Overthinking Length Penalty:** Implements a weighted reward mechanism that penalizes excessive response length while preserving correctness to combat model overthinking. | Kimi et al. (2025); Yu et al. (2025a) |
| **TRAINING DATA** | | |
| **Resource-constrained RL training environments** | **High-impact Sample Selection:** Prioritizes training samples based on learning impact measurement. | Li et al. (2025f) |
| **Training with noisy web-extracted data** | **Noise Reduction Filtering:** Employs filtering mechanisms to remove noisy web-extracted data. | Yeo et al. (2025) |
| **MULTI-STAGE TRAINING** | | |
| **Poor readability and reasoning in direct RL approaches** | **Cold-start Progression:** Implements a phased training approach beginning with high-quality CoT data fine-tuning before transitioning to large-scale reinforcement learning. | DeepSeek et al. (2025); Hou et al. (2025); Luo et al. (2025c); |
| **Inefficient training with problems of varied difficulty** | **Strategic Sampling:** Combines curriculum-based progression from simple to complex problems with prioritization of difficult cases where model performance is weakest. | Kimi et al. (2025) |
| **Inefficient use of context in long-form reasoning** | **Progressive Context Scaling:** Implements a multi-stage training approach that gradually increases context window size as model performance begins to plateau at each level. | Luo et al. (2025c) |
| **Performance gaps on challenging reasoning problems** | **Targeted Annealing:** Implements a final training phase on specifically mined challenging problems with a linearly decaying learning rate to refine reasoning capabilities. | Hu et al. (2025) |

penalty can be added to the reward function:

$$r_t = r_\varphi(q, o_{\leq t}) - \beta \log \frac{\pi_\theta(o_t|q, o_{<t})}{\pi_{\text{ref}}(o_t|q, o_{<t})} \tag{13}$$

where $r_\varphi$ is the reward model, $\pi_{\text{ref}}$ is the reference model (initial SFT model), and $\beta$ is the coefficient of the KL penalty.

**Group Relative Policy Optimization (GRPO)** The GRPO algorithm (Shao et al., 2024) directly uses the average reward of multiple parallel sampled responses as the baseline, eliminating the need for additional value function approximation as in PPO. Specifically, for each question $q$, GRPO samples a group of outputs $\{o_1, o_2, \cdots, o_G\}$ from the old policy $\pi_{\theta_{old}}$ and then optimizes the policy model $\pi_\theta$ by minimizing the loss:

$$\mathcal{L}_{\text{GRPO}}(\theta) = -\mathbb{E}_{q \sim P(Q), \{o_i\}_{i=1}^G \sim \pi_{\theta_{old}}(O|q)}$$

$$\frac{1}{G} \sum_{i=1}^{G} \frac{1}{|o_i|} \sum_{t=1}^{|o_i|} \left[ \min \left( \frac{\pi_\theta(o_{i,t}|q, o_{i,<t})}{\pi_{\theta_{old}}(o_{i,t}|q, o_{i,<t})} \hat{A_{i,t}}, \text{clip}(\theta) \hat{A_{i,t}} \right) - \beta \mathbb{D}_{\text{KL}}[\pi_\theta || \pi_{\text{ref}}] \right] \tag{14}$$

$$\text{clip}(\theta) = \text{clip} \left( \frac{\pi_\theta(o_{i,t}|q, o_{i,<t})}{\pi_{\theta_{old}}(o_{i,t}|q, o_{i,<t})}, 1 - \varepsilon, 1 + \varepsilon \right) \tag{15}$$

$$\mathbb{D}_{\text{KL}}[\pi_\theta || \pi_{\text{ref}}] = \frac{\pi_{\text{ref}}(o_{i,t}|q, o_{i,<t})}{\pi_\theta(o_{i,t}|q, o_{i,<t})} - \log \frac{\pi_{\text{ref}}(o_{i,t}|q, o_{i,<t})}{\pi_\theta(o_{i,t}|q, o_{i,<t})} - 1 \tag{16}$$

where $\varepsilon$ and $\beta$ are hyper-parameters, and $\hat{A_{i,t}}$ is the advantage computed using a group of rewards corresponding to the outputs within each group.

**REINFORCE++** REINFORCE++ (Hu, 2025) is a variant of the classical REINFORCE algorithm that integrates key optimization techniques from PPO while eliminating the need for a critic network. The algorithm incorporates several enhancements to address the limitations of REINFORCE as follows:

- It implements a token-level KL divergence penalty to prevent the policy from deviating too far from the initial model.

- It adopts PPO's clipping mechanism to constrain policy updates and maintain stability during training.

- It introduces mini-batch updates for improved training efficiency and better convergence rates.

- It employs comprehensive reward normalization and clipping to stabilize training by mitigating outliers and constraining reward values within predefined bounds.

- It implements advantage normalization using z-score normalization to ensure stable gradients and prevent divergence during training.

**Comparisons with different algorithms** We summarize the characteristics of different training algorithms in Table 6. Regarding computational cost, PPO shows predominant computational cost with four models to be loaded, among which the policy model and the critic model need to perform both inference and training. GRPO and REINFORCE++ eliminate the need for a critic model and achieve higher training stability than REINFORCE (Hu, 2025). For specific performance comparisons, Hou et al. (2024) find that the performance of PPO and GRPO is similar in Reinforcement Learning from Human Feedback (RLHF) settings, while Xie et al. (2025) observe that the performance of PPO and REINFORCE++ is superior to GRPO in rule-based reward settings for synthetic logic puzzles. More rigorous and large-scale studies should be conducted to comprehensively evaluate the performance of these algorithms.

Table 6: Comparisons of different training algorithms. For the computational overhead, 'Update & Inference' indicates the component's parameters are updated and it is also used for inference. In contrast, 'Inference only' means the component is exclusively used for inference with its parameters frozen.

| Algorithm | Computational Overhead | | | |
|---|---|---|---|---|
| | **Policy** | **Reward** | **Critic** | **Reference** |
| REINFORCE | Update & Inference | Inference only | — | — |
| PPO | Update & Inference | Inference only | Update & Inference | Inference only |
| GRPO | Update & Inference | Inference only | — | Inference only |
| REINFORCE++ | Update & Inference | Inference only | — | Inference only |

### 5.1.2 Reward Function

The reward types can be categorized according to their source and granularity as follows:

- Model-based reward: In traditional RLHF (Ouyang et al., 2022) settings, an explicit reward model is learned from human preference data and guides the optimization process in RL training. The explicit reward model can be omitted by directly training on human preference data, resulting in an implicit reward model (Rafailov et al., 2023).

- Rule-based reward: The term "rule-based" represents rewards that are well-defined and can be determined by explicit rules, sometimes also termed verifiable rewards. For example, for math problems with ground truth answers or code tasks with unit tests, response correctness can be easily verified and thus used to construct the reward. This can be further extended to include response format or language consistency. Even when verification is automated using a specialized model to check answer equivalence (Chen et al., 2024b; Kimi et al., 2025), we still attribute it to rule-based reward as long as the model's performance closely matches ideal rule verification.

- Outcome reward: In general settings, the rule-based reward or model-based reward is only given to the last token of the response, termed "outcome reward."

- Process reward: In multi-step reasoning tasks, the outcome reward may not be sufficient to supervise the policy model and help avoid logic errors in the solutions (Shao et al., 2024; Lightman et al., 2023). This necessitates more fine-grained rewards for each step, termed "process reward," which are typically calculated in a model-based way. We detail the construction of process reward models in §4.2.1. Besides constructing process reward models, recent work also explores other ways to help achieve more accurate credit assignment. For example, Kazemnejad et al. (2024) replace the value networks in the PPO algorithm with unbiased Monte Carlo-based estimates. Hwang et al. (2024) and Setlur et al. (2024a) introduce MC-based methods to detect key errors in reasoning chains for use as ad-hoc mechanisms in DPO.

Figure 9 presents a comparison of different reward types. We detail the discussion below.

**Rule-based reward vs. model-based reward: The model-based reward can be applied to general tasks but also easily leads to reward hacking problems.** This pipeline of constructing preference data to learn a reward model to proxy human preference can be applied to general tasks, leading to its widespread adoption. However, it has been observed that the reward is an imperfect proxy in the training process. There are two prevailing explanations for this phenomenon (Rafailov et al., 2024): 1) OOD Robustness: the reward function is continuously queried using unseen model samples which are potentially out-of-distribution, and 2) Reward Mis-specification: learned reward functions may exhibit spurious correlations that cause them to prefer unintended behaviors. These issues lead to reward overoptimization problems where, during the training process, while the proxy reward score monotonically increases, the golden reward score will saturate and then decrease (Gao et al., 2023). Although this issue can be alleviated by improving the reward model's capability through increased scale or training data (Ouyang et al., 2022; Hou et al., 2024)

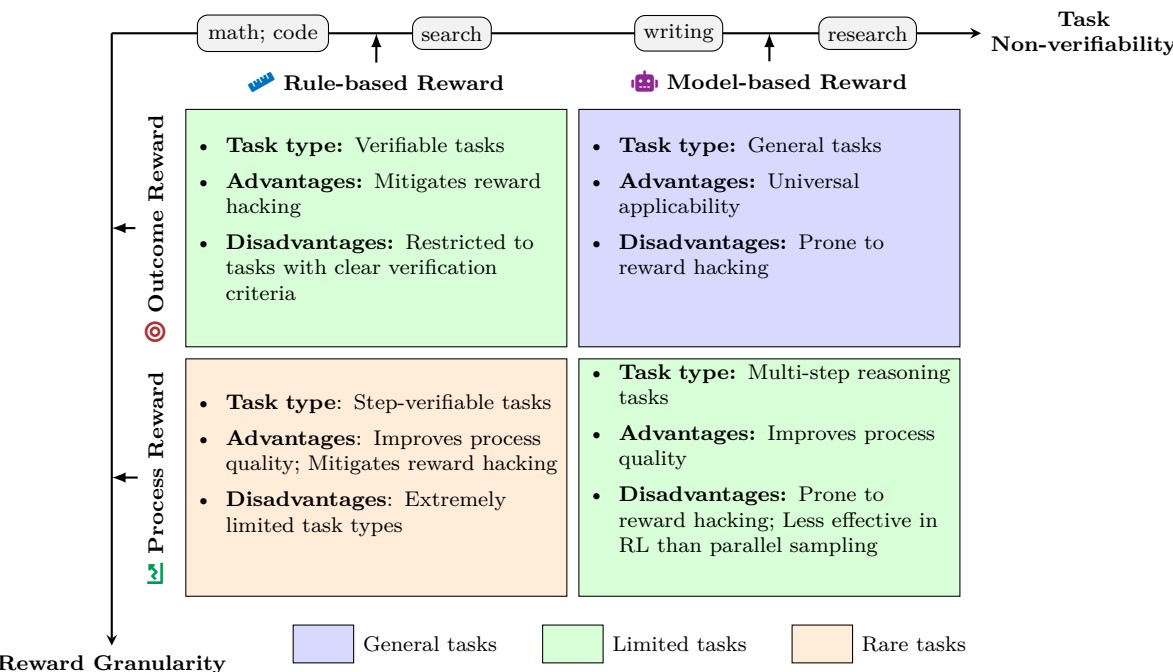

Figure 9: Comparisons of different reward types. Colors indicate applicable task scope.

or iteratively retraining the reward model to improve its supervision of the policy model (Shao et al., 2024), the phenomenon still exists and hinders the success of large-scale RL (DeepSeek et al., 2025).

**Outcome reward vs. process reward: The fine-grained process reward may help improve the RL performance, but also introduces reward hacking problems.** Empirical results show that process rewards can help improve RL performance compared to using only outcome rewards (Cui et al., 2025a; Shao et al., 2024). However, it still faces several challenges: 1) the construction of high-quality training data for process reward models requires significant labor; 2) an imperfect process reward model can be easily hacked. For example, Gao et al. (2024a) find that repeating correct but unnecessary reasoning steps can lead to high rewards from process reward model. Although these issues can be addressed through reward refinement, it complicates the RL pipeline; 3) process rewards show less significant improvements in RL training than in parallel sampling settings. In parallel sampling settings, empirical results show that process reward models significantly outperform outcome reward models (Lightman et al., 2023; Wang et al., 2023a) in response selection. However, the gain is not as pronounced in RL settings (Gao et al., 2024b; Cui et al., 2025a; Shao et al., 2024).

**Optimization for rule-based reward** Rule-based rewards for eliciting long CoT reasoning primarily consist of correctness rewards and format rewards for specific tags. While this approach has proven sufficient for RL scaling, it can lead to potential content misalignment problems due to its narrow focus on accuracy. Two main issues arise from this approach. First, it may result in poor readability and inconsistent language use. Deepseek-R1 (DeepSeek et al., 2025) addresses these challenges by initially fine-tuning their model on thousands of carefully selected long CoT examples. Additionally, it introduces a language consistency reward during RL training to mitigate language misalignment issues. Second, this approach may lead to excessive response length, potentially causing overthinking problems. To address this, Kimi k1.5 (Kimi et al., 2025) implements length penalties in the later training stages, while T1 (Hou et al., 2025) penalizes responses that either exceed the context window size or contain repetitive n-grams.

The success of RL in verifiable tasks demonstrates the importance of robust reward signals. As more research into RL scaling strengthens its theoretical and empirical foundation to facilitate implementation, it decouples the RL training process into two distinct steps: first defining verifiable rewards and then conducting RL

training, as partially implemented in OpenAI's Reinforcement Fine-Tuning Service.[9] Search-R1 (Jin et al., 2025) utilizes a simple outcome reward function that verifies the correctness of final answers to conduct RL training and successfully endows LLMs with the ability to autonomously generate search queries during step-by-step reasoning with real-time retrieval, showcasing the power of RL beyond math and code. For future work in fields like open scientific questions, constructing reliable reward signals remains an open challenge and offers significant potential for innovation.

### 5.1.3 Policy Model Selection

The policy model is a prerequisite for successful RL training. The selection criteria can be based on the following aspects:

**Model Family** As shown in Table 4, most RL scaling work utilizes Qwen2.5 as the base model. Recent studies demonstrate that Qwen2.5 exhibits cognitive behaviors such as verification and correction in its problem-solving process before applying RL (Gandhi et al., 2025; Liu et al., 2025g;f), although the model cannot effectively use them. This indicates that the model's pretrained knowledge already contains these thinking patterns. Gandhi et al. (2025) investigate this phenomenon based on the observation that Qwen-2.5-3B exhibits substantial gains while Llama-3.2-3B quickly plateaus under identical RL training conditions for the game of Countdown. When Llama is primed with synthetic reasoning traces containing these behaviors or pretrained on cognitive behavioral augmentation data, it shows substantial improvements during RL, matching Qwen's performance trajectory. This highlights the importance of pretraining on corpus containing the cognitive behaviors before conducting RL. Moreover, Yue et al. (2025a) find that RL training does not increase the Pass@K score at large K values, indicating that most reasoning abilities manifested in RL-trained models are already possessed by base models. This underscores the importance of base model selection. Further studies should be conducted to analyze the relationship between base model capability and RL training in this new context.

**Model Size** While traditional RLHF settings show that larger models gain fewer benefits from RL optimization (Gao et al., 2023; Hou et al., 2024), RL scaling settings demonstrate that larger models achieve higher token efficiency and thus better performance (Kimi et al., 2025). The limited success in reproducing DeepSeek-R1-Zero's (671B) scaling behavior in 7B or smaller models for challenging tasks without long CoT cold start further suggests that model size significantly impacts scaling behavior.

### 5.1.4 Training Data Construction

The quality and quantity of training data significantly affect the efficiency and upper bound of RL.

**Data Quality** Eliminating easy queries that require no further training helps save the unnecessary computation cost of RL as a post-training technique, where query difficulty can be estimated by sampling multiple times from the policy model to calculate the success rate for correct answers (Kimi et al., 2025; Chen et al., 2025g). Similarly, it is also beneficial to remove problems for which the current model lacks the fundamental capability to solve (Chen et al., 2025g). From the training perspective, queries that the model consistently answers correctly or incorrectly introduce the gradient-decreasing problem. DAPO (Yu et al., 2025a) proposes a dynamic sampling strategy that over-samples and filters out prompts with accuracies of 1 and 0, observing significant performance gains, which can be considered an online difficulty control method.

**Data Quantity** In traditional RLHF settings, scaling the prompt quantity does not lead to significant performance improvements (Hou et al., 2024). However, this conclusion does not hold for RL scaling scenarios. Open-Reasoner-Zero (Hu et al., 2025) investigates the performance discrepancy between a 7.5K MATH (Hendrycks et al., 2021) training set and their curated 57K prompt set, finding that the larger set leads to continuous scaling in both accuracy and response length, while the smaller set plateaus. Similarly, DeepSeek-R1-Zero observes continuous performance improvements using their large-scale curated dataset (DeepSeek et al., 2025).

---

[9] https://openai.com/form/rft-research-program/

### 5.1.5 Multi-stage Training

Training efficiency can be enhanced by employing the following multi-stage training strategy:

**Long CoT Cold Start**  Fine-tuning on long CoT data before RL training can facilitate subsequent RL improvements (Yeo et al., 2025) and mitigate early instability issues during RL training (DeepSeek et al., 2025). Additionally, enhancing the quality of long CoT significantly amplifies RL gains (Yeo et al., 2025). Furthermore, Li (2025) demonstrates improved performance by incorporating sparse updates and adaptive termination mechanisms into the SFT loss function, which helps preserve response diversity after training.

**Iterative Lengthening Strategy**  DeepScaleR-1.5B-Preview (Luo et al., 2025c) initially restricts the context window size to 8K, during which the model generates shorter responses while training rewards increase. Upon reaching a critical point where model responses begin to lengthen, the context window size is expanded to 16K and subsequently to 24K (see the 'DeepScaleR-1.5B-Preview' row in Table 4). This strategy guides controlled response length expansion while reducing computational costs.

**Curriculum Sampling Strategy**  When allocating a restricted computation budget in the initial training phase to very challenging problems, this often yields few correct samples, resulting in lower training efficiency. To address this limitation, the curriculum sampling strategy begins with training on simpler tasks before progressively advancing to more complex ones. Kimi K1.5 (Kimi et al., 2025) reports enhanced performance by implementing this curriculum sampling strategy, leveraging their training dataset that naturally incorporates grade and difficulty labels. Similarly, logic-RL (Xie et al., 2025) examines the utility of this approach but finds that improvements are not substantial in logic puzzles tasks, concluding that it is necessary to balance the complexity of staged training against potential performance gains.

## 5.2 Supervised Fine-tuning

Recent work demonstrates that long CoT test-time scaling behavior can be elicited through simple SFT on similar data (Muennighoff et al., 2025; Ye et al., 2025). This approach is promising given its simpler training process and higher data efficiency compared to RL-based methods. Table 7 presents an organization of long CoT resources. We detail the core design considerations of SFT-based methods as follows:

**Training data source**  The data sources can be categorized based on synthesized data or distillation from existing long CoT models. The trajectory synthesis method includes directly collecting trajectories from tree or graph search processes (Lehnert et al., 2024; Gandhi et al., 2024; Ye et al., 2024) in tasks like path finding or formal logic problems. However, the limited tasks and cognitive behaviors hinder its wider application. Another line of work first solves problems using test-time scaling methods such as tree search processes and multi-turn correction, then translates the search history into thorough exploration trajectories (Zhao et al., 2024; Ma et al., 2025a). For example, Journey Learning (Qin et al., 2024) proposes first guiding LLMs to solve problems using tree search, then using another LLM to translate the backtracking or evaluation steps into natural language to form the trajectory. However, the lack of logical coherence and diversity in these trajectories limits their performance. Xi et al. (2024) transfer multi-turn correction processes into self-talk data, but this approach struggles to generalize to challenging problems due to LLMs' limitations in critique. In contrast to the complexity of synthesis methods, the distillation method directly extracts trajectories from open-source long CoT models, such as Deepseek R1 or QwQ (Ye et al., 2025; Muennighoff et al., 2025; Li et al., 2025c). While this method is cost-effective and performs well compared to synthesis methods, the nature of distillation makes it difficult for student models to surpass their teacher models (Huang et al., 2024).

**Training data quality**  The quality of long-CoT data significantly determines its effectiveness in eliciting model reasoning ability. It comprises both query quality and response quality. For queries, they should be challenging to the base model and cover diverse domains. LIMO (Ye et al., 2025) applies a multi-stage filtration process and retains queries that are challenging even for state-of-the-art reasoning models. S1 (Muennighoff et al., 2025) maintains difficult queries based on model performance and reasoning trace

Table 7: An organization of long CoT resource. The icon 🤗 links to the Hugging Face dataset, ⚙ links to the ModelScope dataset, and ⚫ links to the GitHub repository.

| Work | Application | Type | Source | Quantity | Modality | Link |
|---|---|---|---|---|---|---|
| O1 Journey–Part 1 (Qin et al., 2024) | Math | Synthesize | GPT-4o | 0.3K | Text | ⚫🤗 |
| Marco-o1 (Zhao et al., 2024) | Reasoning | Synthesize | Qwen2-7B-Instruct | 10K | Text | ⚫ |
| STILL-2 (Min et al., 2024) | Math, Code, Science, Puzzle | Distillation | DeepSeek-R1-Lite-Preview QwQ-32B-preview | 5K | Text | ⚫🤗 |
| RedStar-math (Xu et al., 2025b) | Math | Distillation | QwQ-32B-preview | 4K | Text | 🤗 |
| RedStar-code (Xu et al., 2025b) | Code | Distillation | QwQ-32B-preview | 16K | Text | 🤗 |
| RedStar-multimodal (Xu et al., 2025b) | Math | Distillation | QwQ-32B-preview | 12K | Vision Text | 🤗 |
| S1K (Muennighoff et al., 2025) | Math, Science, Code | Distillation | Gemini Flash Thinking | 1K | Text | ⚫🤗 |
| S1K-1.1 (Muennighoff et al., 2025) | Math, Science, Code | Distillation | DeepSeek R1 | 1K | Text | ⚫🤗 |
| LIMO (Ye et al., 2025) | Math | Distillation | DeepSeek R1 DeepSeekR1-Distill-Qwen-32B | 0.8K | Text | ⚫🤗 |
| OpenThoughts-114k (Team, 2025a) | Math, Code, Science, Puzzle | Distillation | DeepSeek R1 | 114K | Text | ⚫🤗 |
| OpenR1-Math-220k (Face, 2025) | Math | Distillation | DeepSeek R1 | 220K | Text | ⚫🤗 |
| OpenThoughts2-1M (Team, 2025a) | Math, Code, Science, Puzzle | Distillation | DeepSeek R1 | 1M | Text | ⚫🤗 |
| CodeForces-CoTs (Team, 2025a) | Code | Distillation | DeepSeek R1 | 47K | Text | ⚫🤗 |
| Sky-T1-17k (Li et al., 2025c) | Math, Code, Science, Puzzle | Distillation | QwQ-32B-Preview | 17K | Text | ⚫🤗 |
| $S^2R$ (Ma et al., 2025a) | Math | Synthesize | Qwen2.5-Math-7B | 3K | Text | ⚫🤗 |
| R1-Onevision (Yang et al., 2025b) | Science, Math, General | Distillation | DeepSeek R1 | 155K | Vision Text | ⚫🤗 |
| OpenO1-SFT (Team, 2024b) | Math, Code | Synthesize | - | 77K | Text | ⚫🤗 |
| Medical-o1 (Chen et al., 2024b) | Medical | Distillation | Deepseek R1 | 25K | Text | ⚫🤗 |
| O1 Journey–Part 3 (Huang et al., 2025d) | Medical | Distillation | o1-preview | 0.5K | Text | ⚫🤗 |
| SCP-116K (Lu et al., 2025) | Math, Science | Distillation | Deepseek R1 | 116K | Text | ⚫🤗 |
| open-r1-multimodal (EvolvingLMMs, 2025) | Math | Distillation | GPT-4o | 8K | Vision Text | ⚫🤗 |
| Vision-R1-cold (Huang et al., 2025c) | Science, Math, General | Distillation | Deepseek R1 | 200K | Vision Text | ⚫🤗 |
| MMMU-Reasoning-Distill-Validation (ModelScope, 2024) | Science, Math, General | Distillation | Deepseek R1 | 0.8K | Vision Text | ⚙ |
| Clevr-CoGenT (Chen et al., 2025c) | Vision Counting | Distillation | Deepseek R1 | 37.8K | Vision Text | ⚫🤗 |
| VL-Thinking (Chen et al., 2025a) | Science, Math, General | Distillation | Deepseek R1 | 158K | Vision Text | ⚫🤗 |
| Video-R1 (Feng et al., 2025a) | Video | Distillation | Qwen2.5-VL-72B | 158K | Vision Text | ⚫🤗 |
| Embodied-Reasoner (Zhang et al., 2025e) | Embodied AI | Synthesize | GPT-4o | 9K | Vision Text | ⚫🤗 |
| OpenCodeReasoning (Ahmad et al., 2025) | Code | Distillation | DeepSeek R1 | 736K | Text | 🤗 |
| SafeChain (Jiang et al., 2025a) | Safety | Distillation | Deepseek R1 | 40K | Text | ⚫🤗 |
| KodCode (Xu et al., 2025e) | Code | Distillation | DeepSeek R1 | 2.8K | Text | ⚫🤗 |

length while covering diverse subjects. For responses, they can be post-filtered through answer checkers and code interpreters. For example, OpenThoughts-114k (Team, 2025a) with verifiers for filtering demonstrates higher performance than OpenThoughts-Unverified-173k, and the precision of verifiers affects the performance. Regarding response content, Li et al. (2025c) find that the global reasoning structure matters more than local content details through perturbation experiments.

**Training data quantity** Empirical results show that scaling data quantity does not bring expected performance improvements relative to computational cost. S1 (Muennighoff et al., 2025) compares the curated 1k dataset with the 59k-full dataset and finds that performance gains are limited. Similarly, the modest performance gap between OpenThoughts-114k (Team, 2025a) and carefully curated 1k datasets further supports this observation. LIMO (Ye et al., 2025) attributes this phenomenon to the Less-Is-More Reasoning Hypothesis, which suggests that the training data primarily serves to elicit sophisticated reasoning capabilities inherent in the model rather than to teach new knowledge.

**Training methods** Whether the self-correction and backtracking ability can be learned through parameter-efficient fine-tuning such as LoRA (Hu et al., 2022) is still under exploration. Li et al. (2025c) compare the performance of LoRA fine-tuning with full parameter fine-tuning and find that the performance is close. This observation contradicts the conclusion of Ye et al. (2024) that the self-correction pattern cannot be learned with LoRA fine-tuning, though the latter only conduct experiments on the synthesized dataset using GPT2-small. More studies should be conducted to verify the effectiveness of LoRA fine-tuning.

**Base models** The performance gain of different base models from long CoT fine-tuning varies significantly (Li et al., 2025c). Li et al. (2025h) find small models ($\leq$ 3B parameters) do not consistently benefit from long CoT reasoning and instead perform better when fine-tuned on shorter reasoning chains. They attribute this to the limited domain knowledge of small models and demonstrate that models with more domain knowledge perform better than those without. Future work should quantitatively analyze the relationship between performance and characteristics of base models.

Although the SFT-based method is easier to implement and more cost-effective compared to RL-based methods, it has several potential limitations. First, the success of the SFT method in eliciting long CoT reasoning ability largely depends on existing open-source long CoT models trained through RL, highlighting the reliance on the teacher models. This characteristic suggests that SFT and RL methods should be combined for higher data efficiency. For example, in the training process of Deepseek-R1, they employ multi-stage training methods with interleaved RL and SFT training, fully utilizing the advantages of both approaches. Second, the SFT-based method is often criticized for memorizing fixed patterns rather than achieving true generalization (Chu et al., 2025a; Mirzadeh et al., 2024; Zhang et al., 2024d). Although empirical results show this criticism may not always hold true as models after small-data SFT can still improve their performance in other subjects and domains (Ye et al., 2025), future studies should carefully analyze the relationship between SFT training steps and generalization ability.

## 5.3 Iterative Self-reinforced Learning

The trajectories generated from test-time scaling methods can be utilized to optimize the policy model through offline methods such as SFT or DPO, thereby achieving self-improvement. This framework functions as a self-reinforcing cycle where data is first generated, then leveraged for learning, after which the original policy model is replaced by its enhanced iteration. We term this training paradigm **iterative self-reinforced learning (ISRL)**. An organization of relevant works is presented in Table 8. The core steps of the algorithm are detailed as follows:

**Sampling** First, responses are sampled from the policy model using a controlled sampling function, which can be implemented with the aforementioned test-time scaling methods, including parallel sampling (Zelikman et al., 2022; Gulcehre et al., 2023; Dong et al., 2023), tree search (Feng et al., 2023; Tian et al., 2024; Zhang et al., 2024a), and multi-turn correction (Xiong et al., 2025; Jain et al., 2025). To enhance the diversity and quality of candidate responses, STaR (Zelikman et al., 2022) provides the correct answer as a hint to guide the generation of rationales when the model fails to solve the problem independently. ReST-MCTS* (Zhang et al., 2024a) empirically demonstrates that step-level tree search outperforms parallel sampling, as the step-level search improves the quality of intermediate reasoning steps. Additionally, other research explores methods to increase query diversity, such as using few-shot prompting to synthesize new problems (Haluptzok et al., 2023; Yuan et al., 2024b).

**Scoring** The sampled responses can be scored using the following methods: 1) for tasks like math and code, the generated solution can be verified against ground truth answers (Zelikman et al., 2022) or validated with unit tests (Huang et al., 2023a); 2) for general tasks, an off-the-shelf reward model can be utilized to score the responses (Dong et al., 2023), or the policy model itself can serve as a judge (Yuan et al., 2024b); 3) for tree search algorithms, accompanying scores from the sampling process help select the correct solution or construct preference pairs (Guan et al., 2025b; Zhang et al., 2024f; Xie et al., 2024b); 4) majority voting. The majority voting strategy can help determine the correct answer when the ground truth is unavailable (Huang et al., 2023a).

Table 8: An organization of works on iterative self-reinforced learning. **IT** denotes whether iterative training is involved. Under **Sampling**, **Query** denotes whether new queries are synthesized, **Response** denotes the sampling method. For the **Scoring** column, **GT** denotes ground truth, **CI** denotes code interpreter, **MV** denotes majority voting, **RM** denotes model-based reward model or LLM-as-a-judge. Under the **Selection & Update**, **Algorithm** represents the training algorithm for the policy model, **Model** represents the training model of each turn (**Orig.** denotes the original model, **Curr.** denotes the current model in the iteration). **Data** represents the source of training data (**Curr.** represents data from the current turn, **Orig.** represents the original data, **Prev.** represents data from all previous turns). **RM** represents whether the reward model gets updated in the process.

| Work | IT | Sampling | | Scoring | Selection & Update | | | |
| | | Query | Response | | Algorithm | Model | Data | RM |
|---|---|---|---|---|---|---|---|---|
| STaR (Zelikman et al., 2022) | ✓ | ✗ | Parallel | GT | SFT | Orig. | Curr. | ✗ |
| P3 (Haluptzok et al., 2023) | ✓ | ✓ | Parallel | CI | SFT | Curr. | Curr. | ✗ |
| LMSI (Huang et al., 2023a) | ✗ | ✓ | Parallel | MV | SFT | Curr. | Curr. | ✗ |
| RAFT (Dong et al., 2023) | ✓ | ✗ | Parallel | RM | SFT | Curr. | Curr. | ✗ |
| RFT (Yuan et al., 2023) | ✗ | ✗ | Parallel | GT | SFT | Orig. | Curr.; Orig. | ✗ |
| ReST (Gulcehre et al., 2023) | ✓ | ✗ | Parallel | RM | SFT | Curr. | Curr.; Prev. | ✗ |
| ReST$^{EM}$ (Singh et al., 2023) | ✓ | ✗ | Parallel | GT | SFT | Orig. | Curr. | ✗ |
| Self-Rewarding (Yuan et al., 2024b) | ✓ | ✓ | Parallel | RM | DPO | Curr. | Curr. | ✓ |
| V-STaR (Hosseini et al., 2024) | ✓ | ✗ | Parallel | GT | SFT | Orig. | Curr.; Prev. | ✓ |
| IRPO (Pang et al., 2024b) | ✓ | ✗ | Parallel | GT | DPO | Curr. | Curr. | ✗ |
| Qwen2.5-MATH (Yang et al., 2024a) | ✓ | ✗ | Parallel | GT; RM | SFT | - | - | ✓ |
| Process-SelfRewarding (Zhang et al., 2025d) | ✓ | ✗ | Parallel | RM | DPO | Curr. | Curr. | ✓ |
| TS-LLM (Feng et al., 2023) | ✓ | ✗ | MCTS | GT | SFT | Curr. | Curr. | ✓ |
| ALPHALLM (Tian et al., 2024) | ✓ | ✗ | MCTS | RM | SFT | Curr. | Curr. | ✓ |
| AlphaMath (Chen et al., 2024a) | ✓ | ✗ | MCTS | GT | SFT | Curr. | Curr. | ✓ |
| ReST-MCTS* (Zhang et al., 2024a) | ✓ | ✗ | MCTS | GT | SFT | Curr. | Curr. | ✓ |
| MCTS-IPL (Xie et al., 2024b) | ✓ | ✗ | MCTS | GT; RM | DPO | Curr. | Curr. | ✗ |
| CPL (Wang et al., 2024e) | ✓ | ✗ | MCTS | GT; RM | SFT; Step-APO | Curr. | Curr. | ✓ |
| SRA-MCTS (Xu et al., 2024a) | ✗ | ✗ | MCTS | RM | SFT | Curr. | Curr. | ✓ |
| rStar-Math (Guan et al., 2025b) | ✓ | ✗ | MCTS | GT | SFT | Curr. | Curr. | ✓ |
| Xiong et al. (2025) | ✗ | ✗ | Multi-turn correction | GT | SFT | Curr. | Curr. | ✗ |
| μCODE (Jain et al., 2025) | ✓ | ✗ | Multi-turn correction | RM | SFT | Curr. | Curr. | ✓ |
| SPaR (Cheng et al., 2024a) | ✓ | ✗ | Ensemble | RM | DPO | Curr. | Curr. | ✓ |
| SWE-Reasoner (Ma et al., 2025d) | ✗ | ✗ | Long CoT | GT | SFT | Curr. | Curr. | ✗ |

**Selection and Update** The response pool accompanied by scores is further selected and utilized to update the policy model and optionally the reward model. The policy model can be updated using SFT (Zelikman et al., 2022; Gulcehre et al., 2023; Singh et al., 2023) or DPO (Yuan et al., 2024b; Pang et al., 2024b). The reward model can also be updated, such as updating the process reward model on labels generated by rollouts in the tree search process (Feng et al., 2023; Zhang et al., 2024a) or training an outcome reward model on generated positive and negative samples (Hosseini et al., 2024; Yang et al., 2024a). For example, V-star (Hosseini et al., 2024) trains an ORM to utilize negative samples generated in the sampling process and uses it for reranking during inference time. During the update process, the policy model for the next iteration of data generation can be fine-tuned from either the initial model or the model in the current iteration. The training data can originate from various sources: the current turn, the initial dataset, or accumulated data from all previous turns.

Although ISRL is promising considering its data efficiency empowered by offline methods and the advantage of not requiring expert demonstrations, empirical results show that the rate of improvement tends to plateau or even decline slightly after few iterations (Wu et al., 2024b). This phenomenon contrasts with RL scaling methods where model performance improves monotonically. One major distinction between the two algorithms is that ISRL is closer to off-policy sampling. As demonstrated by Tajwar et al. (2024), a higher degree of on-policy sampling leads to better performance. This is further supported by the observation of Shao et al. (2024) that the performance of online rejection sampling fine-tuning (RFT) is comparable to RFT in the early stage of training but gains a significant advantage in the later stage. Moreover, for algo-

Table 9: Application of test-time scaling in different domains (Part 1). This table covers tree search and multi-turn correction. For the parallel sampling and long CoT columns, please see Table 10. Green color represents combining test-time scaling with iterative training, i.e., the iterative self-reinforced learning in the paper.

| Application | Tree Search | Multi-turn Correction |
|---|---|---|
| **Mathematics** | GPT-f (Polu & Sutskever, 2020) 
 HTPS (Lample et al., 2022) 
 BFS-Prover (Xin et al., 2025) 
 ToT (Yao et al., 2023a) 
 MindStar (Kang et al., 2024a) 
 RAP (Hao et al., 2023) 
 Q* (Wang et al., 2024b) 
 Self-Evaluation Guided (Xie et al., 2023) 
 TS-LLM (Feng et al., 2023) 
 LiteSearch (Wang et al., 2024a) 
 ALPHALLM (Tian et al., 2024) 
 AlphaMath (Chen et al., 2024a) 
 MCTSr (Zhang et al., 2024b) 
 ReST-MCTS* (Zhang et al., 2024a) 
 REBASE (Wu et al., 2024c) 
 rStar-Math (Guan et al., 2025b) 
 LLaMA-Berry (Zhang et al., 2024c) | RISE (Qu et al., 2024) 
 SCoRe (Kumar et al., 2024) 
 AutoMathCritique (Xi et al., 2024) |
| **Code** | PG-TD (Zhang et al., 2023) 
 o1-Coder (Zhang et al., 2024g) 
 Q* (Wang et al., 2024b) 
 SWE-Reasoner (Ma et al., 2025d) 
 PLANSEARCH (Wang et al., 2024c) 
 RethinkMCTS (Li et al., 2024e) 
 SRA-MCTS (Xu et al., 2024a) | Reflexion (Shinn et al., 2023) 
 Self-Debug (Chen et al., 2023c) 
 CRITIC (Gou et al., 2023) 
 IHR (Qiu et al., 2023) 
 Self-Repair (Olausson et al., 2023) 
 STOP (Zelikman et al., 2024b) |
| **Multimodality** | VisVM (Xiyao et al., 2024) 
 LLaVA-CoT (Xu et al., 2024b) 
 Mulberry (Yao et al., 2024a) 
 LlamaV-o1 (Thawakar et al., 2025) 
 Video-T1 (Liu et al., 2025a) | $R^3V$ (Cheng et al., 2024b) 
 Insight-V (Dong et al., 2024) 
 PARM (Guo et al., 2025b) 
 GoT (Fang et al., 2025) |
| **Agent** | Agent Q (Putta et al., 2024) 
 ToT (Yao et al., 2023a) 
 SearchAgent (Koh et al., 2024) | Reflexion (Shinn et al., 2023) 
 Agent Q (Putta et al., 2024) 
 Agent-Eval-Refine (Pan et al., 2024b) |
| **Embodied AI** | - | Inner Monologue (Huang et al., 2022) 
 REFLECT (Liu et al., 2023e) 
 KnowNo (Ren et al., 2023) |
| **Safety** | STAIR (Zhang et al., 2025f) 
 C-MCTS (Parthasarathy et al., 2023) 
 HaluSearch (Cheng et al., 2025) 
 InferenceGuard (Ji et al., 2025a) 
 ARGS (Khanov et al., 2024) | MART (Ge et al., 2024) 
 Combat Adv. Attacks (Chern et al., 2024b) 
 Improve Factuality (Du et al., 2023) 
 Multi-expert Prompting (Long et al., 2024) 
 DebateGPT (Subramaniam et al., 2024) |
| **RAG** | AirRAG (Feng et al., 2025b) 
 CoRAG (Wang et al., 2025c) | - |
| **Evaluation** | MCTS-Judge (Wang et al., 2025f) | ChatEval (Chan et al., 2023) 
 ScaleEval (Chern et al., 2024a) |

rithms that employ SFT for policy updates, they do not utilize negative gradients for pushing down certain responses. Empirical results show that including negative gradients in policy updates leads to significantly better performance compared to using positive gradients alone, especially in reasoning tasks (Kimi et al., 2025).

# 6 How's Progress – Application So Far

In this section, we examine the systemic changes in AI research that cognition engineering driven by test-time scaling brings and the applications that have already emerged.

Table 10: Application of test-time scaling in different domains (Part 2). This is a continuation of Table 9. This table covers parallel sampling and long CoT. *Italics* represents traditional CoT work, Yellow color represents using RL tech, Purple color represents using SFT tech, Green color represents combining test-time scaling with iterative training.

| Application | Parallel Sampling | Long CoT |
|---|---|---|
| Mathematics | Self-Consistency (Wang et al., 2023c) ORM (Cobbe et al., 2021) PRM800K (Lightman et al., 2023) Math-shepherd (Wang et al., 2023a) STaR (Zelikman et al., 2022) V-STaR (Hosseini et al., 2024) IRPO (Pang et al., 2024b) | Openai o1 (Team, 2024b); O1 Journey–Part1 (Qin et al., 2024) O1 Journey-Part2 (Huang et al., 2024); STILL-2 (Min et al., 2024) T1 (Hou et al., 2025); Deepseek-R1 (DeepSeek et al., 2025) Kimi k1.5 (Kimi et al., 2025); SimpleRL (Zeng et al., 2025a) S1 (Muennighoff et al., 2025); LIMO (Ye et al., 2025) Demystifying (Yeo et al., 2025); LIMR (Li et al., 2025f) DeepScaleR (Luo et al., 2025c); QwQ (Team, 2025b) DAPO (Yu et al., 2025a); *Eurus-2-7B-PRIME* (Cui et al., 2025a) STILL-3 (Chen et al., 2025g); Open-Reasoner-Zero (Hu et al., 2025) VAPO (Yue et al., 2025b); Open-RS (Dang & Ngo, 2025) |
| Code | MBR-EXEC (Shi et al., 2022) CodeT (Chen et al., 2023a) S* (Li et al., 2025b) AlphaCode (Li et al., 2022) AlphaCode2 (Team, 2024a) | Deepseek-R1 (DeepSeek et al., 2025); Kimi k1.5 (Kimi et al., 2025) SWE-RL (Wei et al., 2025); SWE-Gym (Pan et al., 2024a) OpenAI o1 (OpenAI, 2024); QwQ-preview (Team, 2024c) QwQ (Team, 2025b); SWE-Reasoner (Ma et al., 2025d) OpenCodeReasoning (Ahmad et al., 2025); ToRL (Li et al., 2025g) DeepCoder (Luo et al., 2025b); Seed-Thinking-v1.5 (Seed, 2025) |
| Multimodality | URSA (Luo et al., 2025d) PARM (Guo et al., 2025b) | *MAmmoTH-VL* (Guo et al., 2024); Virgo (Du et al., 2025) QVQ-72B-Preview (Qwen, 2024); R1V (Chen et al., 2025c) Open-R1-Multimodal (EvolvingLMMs, 2025) R1-Multimodal-Journey (Meng et al., 2025a) LMM-R1 (Peng et al., 2025); VLM-R1 (Shen et al., 2025a) R1-Video (Wang & Peng, 2025); R1-Onevision (Yang et al., 2025b) MM-Eureka (Meng et al., 2025b); Vision-R1 (Huang et al., 2025c) VisualThinker-R1-Zero (Zhou et al., 2025a); Visual-RFT (Shen et al., 2025a); Seg-Zero (Liu et al., 2025e) vsGRPO (Liao et al., 2025); Video-R1 (Feng et al., 2025a) *MVoT* (Li et al., 2025a); Kimi-VL (Kimi Team, 2025) Kimi k1.5 (Kimi et al., 2025); O3/O4-mini (OpenAI, 2025a;b) MAYE (Ma et al., 2025c) |
| Agent | - | *ReAct* (Yao et al., 2023b); Deep Research (OpenAI, 2025b) SWE-RL (Wei et al., 2025); *Operator* (OpenAI, 2025a) *UI-TRARS* (Qin et al., 2025); *PC Agent* (He et al., 2024) DeepResearcher (Zheng et al., 2025); *SWEET-RL* (Zhou et al., 2025c) Claude 3.7 Sonnet (Anthropic, 2025) |
| Embodied AI | - | *Embodied-CoT* (Michał et al., 2024); *CoA* (Li et al., 2024b) *SpatialCoT* (Liu et al., 2025d); *RAD* (Clark et al., 2025) Cosmos-Reason1 (Azzolini et al., 2025); *Gemini Robotics* (Gemini Robotics et al., 2025) CoT-VLA (Zhao et al., 2025); Embodied-Reasoner (Zhang et al., 2025e) |
| Safety | SelfCheckGPT (Manakul et al., 2023) SRG (Wang et al., 2025b) | Deliberate Alignment (Guan et al., 2024); Chain-of-Verification (Dhuliawala et al., 2023); SafeChain (Jiang et al., 2025a); MoTE (Liu et al., 2024e) |
| RAG | CoRAG (Wang et al., 2025c) | *IterDRAG* (Yue et al., 2024); *Plan*RAG* (Verma et al., 2024) DeepRAG (Guan et al., 2025a); Search-o1 (Li et al., 2025e) AirRAG (Feng et al., 2025b); Auto-RAG (Yu et al., 2024b) CoRAG (Wang et al., 2025c); Search-R1 (Jin et al., 2025) R1-Searcher (Song et al., 2025); ReSearch (Chen et al., 2025d) DeepRetrieval (Jiang et al., 2025b) |
| Evaluation | CCE (Zhang et al., 2025c) SPCT (Liu et al., 2025h) | *FActScore* (Min et al., 2023); *FacTool* (Chern et al., 2023) *RefChecker* (Hu et al., 2024b); *RAGChecker* (Ru et al., 2024) *Agent-as-a-Judge* (Zhuge et al., 2024); *EvalPlanner* (Saha et al., 2025) Kim et al. (2025) |

## 6.1 Mathematics

Mathematical reasoning is crucial for resolving complex problems and making informed decisions (Hendrycks et al., 2021; Xia et al., 2024). Research in AI for mathematics (AI4Math) has developed along two complementary paths: natural language reasoning focusing on questions with verifiable answers, and formal language reasoning utilizing formal systems like Lean (De Moura et al., 2015) and Isabelle (Nipkow et al., 2002) for automatic formal theorem proving. For questions with verifiable answers, the easy verification characteristic makes it reliable to construct feedback signals for search and learning, facilitating the wide application of test-time scaling methods to enhance reasoning abilities. These include parallel sampling (Cobbe et al., 2021; Wang et al., 2023c), tree search (Feng et al., 2023; Chen et al., 2024a; Hao et al., 2023), multi-turn correction (Kumar et al., 2024; Qu et al., 2024), and long CoT (DeepSeek et al., 2025; OpenAI, 2024). Notably,

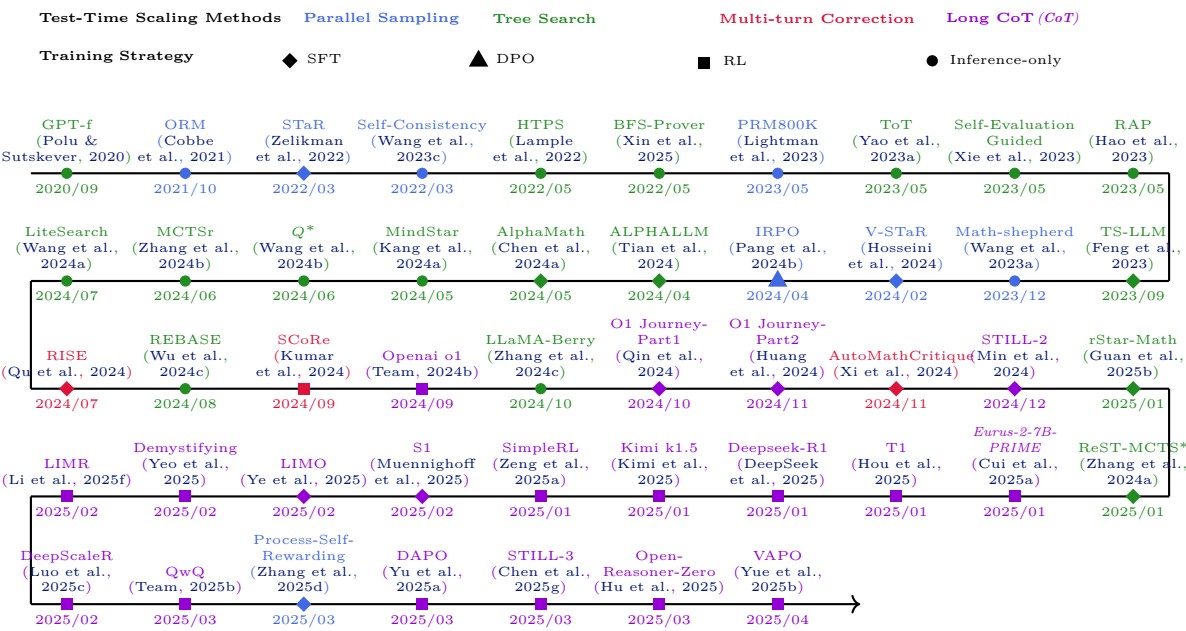

Figure 10: Works of applying test-time scaling methods in the math field.

powered by long CoT, DeepSeek-R1 achieves a score of 79.8 on the American Invitational Mathematics Examination (AIME), significantly outperforming traditional models without long CoT and approaching competitive human performance. For formal language reasoning, formal systems make the reasoning process verifiable and provide signals for tree search (Polu & Sutskever, 2020; Lample et al., 2022; Xin et al., 2024; 2025) or multi-turn correction (First et al., 2023). Breakthrough systems like AlphaProof (AlphaProof & teams, 2024) and AlphaGeometry (Trinh et al., 2024) demonstrate that combining neural networks with formal methods and proof checkers can achieve unprecedented mathematical reasoning abilities.

Despite these successes, there still exists room for improvement. In natural language reasoning, while training data accumulation is substantial, the difficulty in strictly verifying reasoning process correctness means solutions generated by LLMs may contain logical errors or lack rigor in intermediate steps (Lightman et al., 2023; Xia et al., 2024). For formal language reasoning, while it ensures reasoning process verifiability, the lack of training data compared to natural language limits its development. Future work can focus on unifying the advantages of formal and natural languages for more robust model development. Moreover, although LLMs with strong reasoning and cognition abilities have made progress on exam problems and competition tasks, applications to more advanced domains such as mathematical research remain relatively unexplored (Yang et al., 2024b). This necessitates not merely enhanced model capabilities but also novel evaluation frameworks to assess these competencies.

---

**Future Direction for Mathematics**

- Unify the advantages of formal and natural languages to develop more robust reasoning models that combine the verifiability of formal systems with the rich training data available in natural language.

- Expand applications of LLMs with strong reasoning capabilities beyond exam problems toward more advanced domains such as mathematical research, developing novel evaluation frameworks to assess these higher-level competencies.

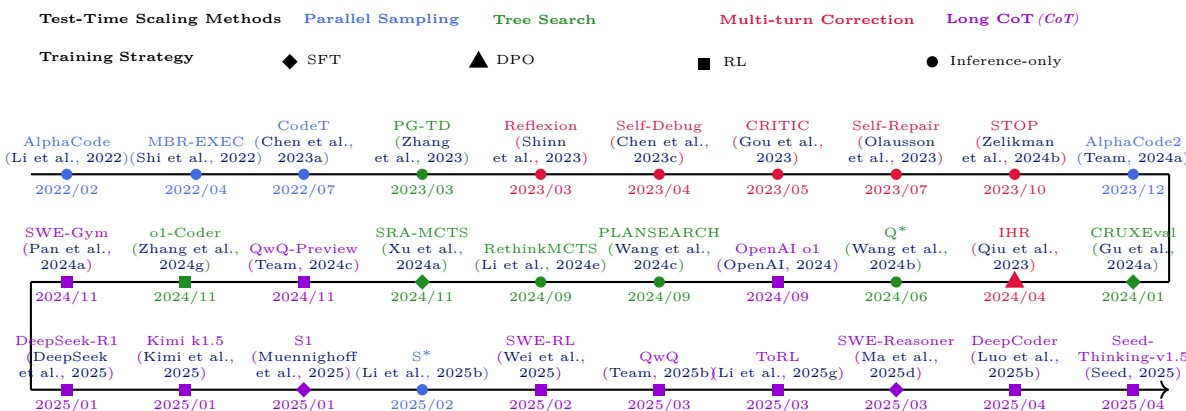

Figure 11: Works of applying test-time scaling methods in the code field.

## 6.2 Code

The swift emergence of coding capabilities in language models—exemplified by Codex (Chen et al., 2021) and AlphaCode (Li et al., 2022; Team, 2024a), has transformed software development. Several studies (Fu & Khot, 2022; Ma et al., 2023; Shao et al., 2024) even suggest that code enhances model intelligence. Moreover, coding is now a core feature of general-purpose foundation models (DeepSeek-AI, 2024), highlighting its critical role in modern model development.

Previously, approaches in code synthesis and code generation have demonstrated how execution verification enables both scalable and verifiable training and test-time signals, laying the foundation for subsequent advancements (Le et al., 2022; Chen et al., 2023a; Zhu et al., 2024). Additionally, directly prompting LLMs to self-reflect, debug, and generate tests for coding tasks represents another direction of test-time scaling (Shinn et al., 2023; Chen et al., 2023c), which enhances model performance in specific downstream scenarios while also providing feedback for refining training strategies (Gu et al., 2024a). These innovations are believed to be instrumental in developing stronger reasoning models, such as the o1-series, which achieve state-of-the-art performance on elite programming benchmarks like SWE-bench (Jimenez et al., 2024) and even human-competitive platforms such as Codeforces.[10] This is further exemplified by o1 and o3 model variants earning gold medals at the 2024 International Olympiad in Informatics (IOI), where RL on challenging programming tasks is combined with cognitively aligned human guidance (El-Kishky et al., 2025).

Still, critical challenges remain. First, while code executability facilitates verification, naive execution poses security risks, necessitating robust sandboxing solutions (Hui et al., 2024; Liu et al., 2024c). This also requires the development of infrastructure for the practical deployment of these safety mechanisms. Second, frequent reflection behaviors—often referred to as "overthinking"—can degrade performance in certain tasks, as observed in coding agentic benchmarks like Aider.[11] Third, the reliability of execution-based feedback remains an open question. While DeepSeek-R1 (DeepSeek et al., 2025) relies on execution results as feedback signals to achieve superior performance, code that passes unit tests can still fail with additional tests, leading to false positives (Stroebl et al., 2024), highlighting a fundamental limitation in execution-based evaluation. Furthermore, further investigation is also needed to align models with real-world coding tasks, as motivated by SWE-Arena (team swe arena, 2025) and Copilot-Arena (Chi et al., 2025). These challenges may even intersect with broader issues such as multi-modal understanding and agentic capabilities which is further discussed in the next few sections.

---

[10]https://codeforces.com
[11]https://aider.chat/

**Future Direction for Coding**

- Go beyond competition-level programming to encompass real-world software development tasks and advance toward automatic code review, debugging, and repo-level optimization.

- Expand to support more programming languages and continuously learn newly released libraries to achieve expert-level proficiency.

## 6.3 Multimodality

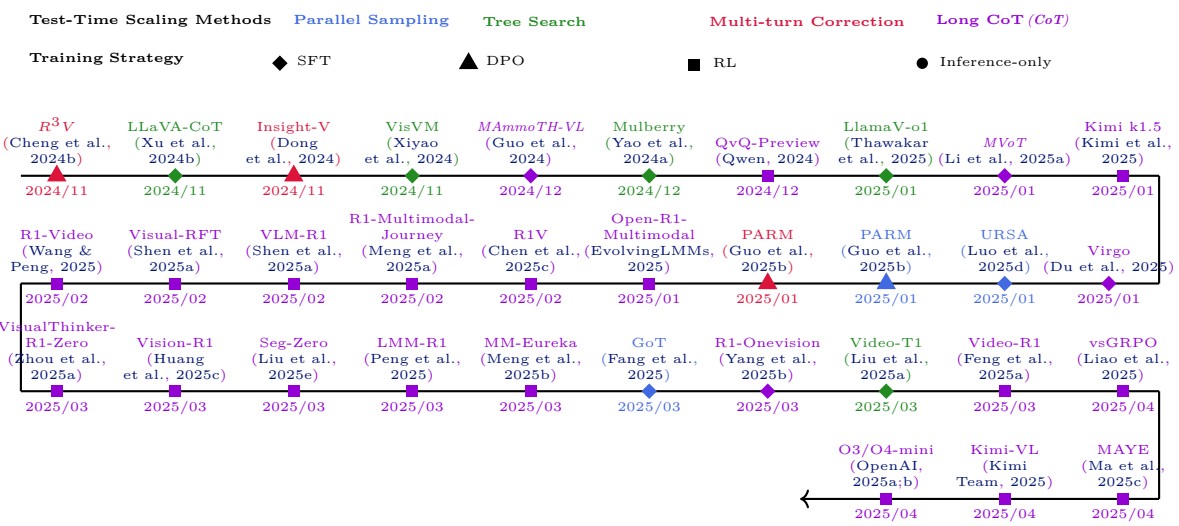

Figure 12: Works of applying test-time scaling methods in the multi-modal field.

**Test-Time Scaling for Multimodal Understanding**   Vision-Language Models (VLMs) (Zhang et al., 2024e) have demonstrated significant potential in tasks involving multimodal understanding and generating textual outputs. Test-time scaling techniques for VLMs can be directly adapted from LLMs since the task outputs are textual.

DeepSeek R1 represents a significant milestone, being the first to fully demonstrate the effectiveness of RL training in scaling test time for LLMs. It effectively divides research on test-time scaling in VLMs into two distinct phases: pre-R1 and post-R1. Before R1, research efforts relied on long CoT distillation (Guo et al., 2024; Du et al., 2025; Xu et al., 2024b), tree search (Xu et al., 2024b; Thawakar et al., 2025; Yao et al., 2024a; Xiyao et al., 2024), multi-turn correction (Cheng et al., 2024b; Dong et al., 2024), and parallel sampling (Luo et al., 2025d). For example, LLaVA-CoT (Xu et al., 2024b) and LlamaV-O1 (Thawakar et al., 2025) enhance reasoning ability by decomposing the reasoning process into explicit sequential steps, such as summarizing the problem, reviewing image content, and performing reasoning. Similarly, Mulberry (Yao et al., 2024a) and LLaVA-CoT (Xu et al., 2024b) employ search algorithms like beam search and MCTS to expand the search space during reasoning. MAmmoTH-VL (Guo et al., 2024) and Virgo (Du et al., 2025) scale the output length of smaller models by distilling reasoning chains from larger VLMs, resulting in significant improvements across various visual reasoning tasks. As part of R1's concurrent work, QVQ-72B-Preview (Qwen, 2024) and K1.5 (Kimi et al., 2025) have demonstrated the immense potential of RL in VLMs. After R1, the research community has increasingly explored RL-based training as a strategy to elicit VLMs' test-time scaling ability. Recent efforts primarily fall into two directions: 1) enhancing the depth of multimodal reasoning, by pushing the limits of VLMs' visual problem-solving capabilities—such as in open-r1-multimodal (EvolvingLMMs, 2025), MM-Eureka (Meng et al., 2025b), LMM-R1 (Peng et al., 2025), Vision-R1 (Huang et al., 2025c), VisualThinker-R1-Zero (Zhou et al., 2025a) and so on; and 2) expanding the breadth of multi-modal tasks, demonstrating the effectiveness of RL training across diverse vision-centric

domains, including visual counting (Chen et al., 2025c), detection (Liu et al., 2025i; Shen et al., 2025a), segmentation (Liu et al., 2025e), etc.

Despite promising results, long-CoT-based test-time scaling on VLMs still faces several challenges. First, unlike LLMs, input instructions for VLMs involve multiple modalities including both visual and textual data. Many studies that replicate LLM test-time scaling methods directly on VLMs fail to incorporate rethinking and reflection on visual inputs in the VLM's responses, overlooking the unique characteristics of multimodal understanding tasks. Future research should aim to better integrate test-time scaling with processing different modality inputs to address hallucinations from non-textual inputs (Liu et al., 2024b), while leveraging the synergies of multimodal inputs to enhance the effectiveness of test-time scaling. Second, conducting training directly on base models remains challenging. Unlike LLMs, VLMs lack strong base models because their training focuses mainly on modality alignment through image-caption pairs and instruction tuning (Liu et al., 2023a; 2024a; Li et al., 2024a), without extensive pre-training on general multimodal corpora, which demands substantial computational and data resources.

**Test-Time Scaling for Multimodal Generation**   Recent breakthroughs such as Gemini's image-text interleaved generation and GPT-4o's image editing capabilities have reignited broad interest in multimodal generation. Unlike multimodal understanding tasks, where the output remains textual and test-time scaling techniques can be readily borrowed from LLMs, multimodal generation involves producing non-textual outputs (e.g., images, videos, or audio), thus requiring new approaches to unlock the full potential of test-time scaling in this domain.

Several recent works have explored this frontier. Li et al. (2025a) enhance MLLM spatial reasoning via the Multimodal Visualization-of-Thought (MVoT) framework, which adopts a long CoT strategy to scale reasoning at test time. PARM (Guo et al., 2025b) and MINT (Wang et al., 2025d) promote multimodal generation through multi-turn correction, aiming to improve image quality by iteratively reflecting on and revising the associated text prompts. GoT (Fang et al., 2025) unleashes the reasoning capability of MLLMs for visual generation and editing by introducing the Generation Chain-of-Thought framework, which adopts a long CoT strategy for test-time scaling. Expanding beyond images, Video-T1 (Liu et al., 2025a) extends input modalities to video generation, utilizing a tree search method at inference time to enhance video generation quality.

Looking ahead, applying test-time scaling to multimodal generation represents a highly promising but underexplored direction. Open questions include identifying the most effective model architecture (Chameleon, 2024; Zhou et al., 2024a; Chen et al., 2025f), designing pretraining and instruction-tuning strategies tailored to non-textual outputs, and improving the computational efficiency of such systems. Moreover, if RL is to be extended to multimodal generation, key challenges emerge in defining suitable reward signals and constructing RL infrastructure for non-text modalities.

> **Future Direction for Multimodality**
>
> - Focus more on applying test-time scaling to vision-centric tasks such as classification and detection, as well as non-text outputs like images and videos.
>
> - Integrate more modalities (e.g., audio) and develop stronger native multimodal foundation models to unlock test-time scaling potential.

## 6.4   Agent

LLM Agent is an autonomous system that leverages LLMs as its cognitive core to automatically perform complex tasks through action execution in dynamic environments (Sumers et al., 2023). Building on prior efforts, the objective of agents is evolving from specific pre-defined workflows to handling more open-ended tasks in complex environments that need decision-making at scale, such as software engineering (Jimenez et al., 2024), deep research (OpenAI, 2025b), and computer use (Anthropic, 2024b; OpenAI, 2025a). Such tasks often require long-horizon planning to be accomplished with multi-step interaction with the environment, leading to substantial test-time computation. For instance, OpenAI DeepResearch (OpenAI, 2025b)

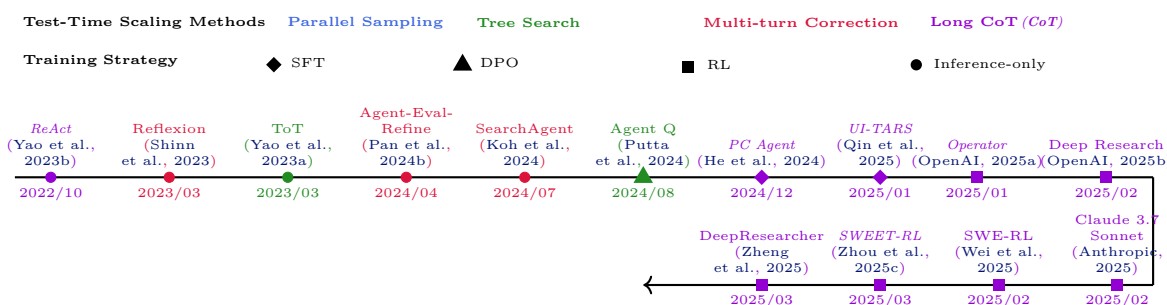

Figure 13: Works of applying test-time scaling methods in the agent field.

demands 5–30 minutes for a single research task, while CUA (OpenAI, 2025a) may require hundreds of steps to complete a computer use task and exhibit clear test-time scaling behavior.

A prerequisite for effective multi-step task execution is the enhancement of decision-making in each step. This necessitates models with advanced reasoning capabilities to perform verification, backtracking, and reflection based on historical trajectories and current environment observations, thereby aligning actions with long-term goals. Numerous approaches have adopted test-time scaling strategies to optimize per-step decision quality. For example, ReAct (Yao et al., 2023b) introduces CoT reasoning during action selection. Reflexion (Shinn et al., 2023) further advances this paradigm by incorporating explicit feedback signals from prior steps to enable self-correction. Recently, Deep Research has highlighted the potential of scaling the reasoning process in single-step decision making. Powered by an optimized version of the OpenAI o3, it achieves research-analyst-level proficiency in synthesizing online sources into comprehensive reports, and attains 26.6% accuracy on the challenging Humanity's Last Exam benchmark (Phan et al., 2025), significantly surpassing previous SOTA models.

Regarding training strategies, many methods introduce historical trajectories into SFT training samples to help models learn to handle multi-step histories, and incorporate thinking processes at each step to achieve CoT capabilities (He et al., 2024). Additionally, UI-TARS (Qin et al., 2025) addresses the limitation of SFT methods that only utilize the corrected steps through DPO methods, thereby enhancing the agent's error correction and post-reflection abilities. Although the specific implementation of OpenAI Deep Research remains unclear, recent work applies RL for end-to-end training of deep research agents (Zheng et al., 2025).

Despite some production-ready implementations such as GitHub Copilot, most agentic systems remain confined to proof-of-concept demonstrations rather than robust, large-scale deployment. Key barriers include insufficient general model capabilities, the predominance of prompt engineering over specialized agentic training, and context window constraints for long trajectories—particularly for visual observations. Additionally, unlike code or math tasks, many agentic tasks lack well-defined external verifiers, making it challenging to provide reliable rewards in RL frameworks. Moreover, the involvement of long CoT in reasoning also introduces the "Reasoning-Action Dilemma," which requires models to carefully balance active engagement with the environment against the need for internal reasoning, highlighting the importance of developing reasoning models that remain effectively grounded in environmental context when applied to agentic tasks (Cuadron et al., 2025).

> **Future Direction for Agents**
>
> - Develop robust execution environments that embrace diverse tool use, and craft scalable evaluation frameworks, which pave the way for RL in agent training together.
>
> - Further explore action scaling as a new scaling dimension—expand agents' competence by growing the number of interaction steps with the environment.

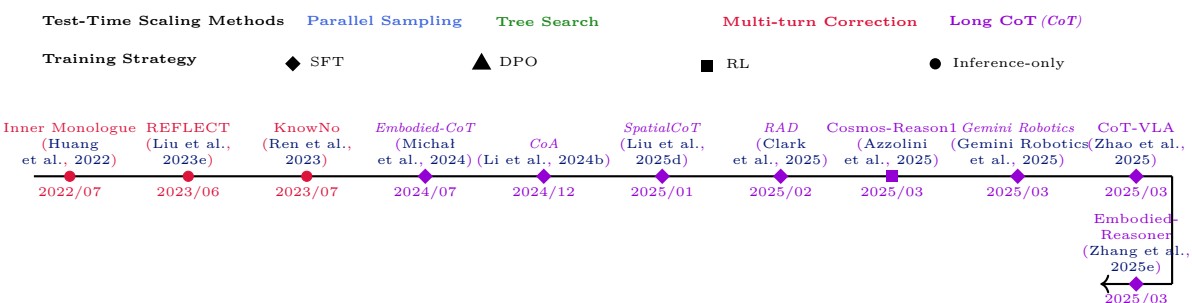

Figure 14: Works of applying test-time scaling methods in the Embodied AI field.

## 6.5 Embodied AI

Embodied AI is essential to advancing AGI, as it establishes the foundational link between cognitive representation and interaction with the physical world. By enabling robots to engage with their surrounding environments and manipulate objects, embodied AI allows for the execution of real-world and complex tasks, necessitating sophisticated levels of cognition and reasoning abilities. Cognition engineering further supports the enhancement of these cognitive and reasoning abilities within embodied AI systems. Embodied AI systems typically adopt a hierarchical architecture comprising two distinct yet interconnected operational phases: high-level planning and low-level control policy execution (Ahn et al., 2022).

First, high-level planning is the process of creating a sequence of sub-tasks that a robot can follow to achieve a specific goal within its environment. This involves decision-making based on both the robot's current state and the predicted outcomes of its actions, aiming to optimize efficiency, safety, and goal attainment within often dynamic and complex surroundings. This process frequently requires advanced cognitive abilities for analytical thinking and reasoning. Huang et al. (2022) pioneer the use of inner thoughts as a feedback mechanism to enhance reasoning capabilities, enabling multi-turn self-correction through internal monologue. Building on this idea, Liu et al. (2023e) leverage LLMs to generate explicit failure explanations, which are then utilized to refine reasoning processes, thereby achieving substantial improvements in planning and problem-solving performance. Ren et al. (2023) propose a method to quantify the uncertainty of LLM-based planners and trigger assistance requests when the uncertainty surpasses a predefined threshold, which enhances reasoning ability through uncertainty alignment. Inspired by o1 (OpenAI, 2024), Liu et al. (2025d) employ bi-directional spatial coordinate alignment and chain-of-thought spatial grounding to facilitate the generation of long thoughts, thereby improving planning performance. Furthermore, Chai et al. (2024) propose a novel approach based on Q-learning, which enables the model to make optimal decisions. Inspired by Deepseek-R1 (DeepSeek et al., 2025), Azzolini et al. (2025) introduce a novel VLM designed from the ground up to enhance reasoning capabilities in physical environments. The model incorporates an innovative hybrid architecture that combines Mamba, MLP, and Transformer components. This approach is complemented by comprehensive vision pre-training, specialized physical AI SFT, and physical AI RL. The resulting system effectively processes video input paired with linguistic instructions, generating long reasoning thoughts before predicting appropriate subsequent actions. Meanwhile, Zhang et al. (2025e) propose a dataset comprising 9.3K synthesized instances to enhance embodied reasoning capabilities, utilizing GPT-4o to generate Observation-Thought-Action trajectories that serve as extended reasoning chains for long-horizon action prediction tasks.

Second, low-level control policy aims to translate tasks into executable actions, such as those performed by a 7-DOF robotic arm in joint space. Currently, the most prevalent approach leverages large models, particularly through vision-language-action (VLA) frameworks. These models fine-tune pretrained vision-language models on trajectory data to generate actionable sequences. Importantly, reasoning plays a key role in this process. For instance, if the task involves placing apples on one plate and bananas on another, the model must first distinguish between apples and bananas, rather than relying on simple "muscle memory" from prior learning. Embodied CoT (Michał et al., 2024) proposes constructing CoT that bridges perceptual information and task objectives by incorporating external knowledge from other models or algorithms. This

approach has been demonstrated to substantially improve performance. In contrast, Zhao et al. (2025) introduce a visual CoT framework that integrates explicit visual reasoning processes into VLA models. Their approach generates future image frames in an autoregressive manner, establishing them as visual objectives before producing concise action sequences designed to achieve these predetermined goals. Zhang et al. (2024h) introduce a novel preference alignment approach for robotic policy learning, allowing VLA models to learn not only from successful trajectories but also from failure trajectories. Additionally, Li et al. (2024b) integrate diverse robot affordance information to enhance the model's generalization in reasoning during testing and leverage the generated reasoning to foster long-horizon reasoning capabilities. However, SFT-based approaches rely on high-quality expert datasets, which are both expensive and challenging to obtain in the robotics domain. Moreover, these datasets may not fully align VLA models with real-world physical environments due to distribution shift issues. To address this limitation, Guo et al. (2025a) propose a novel method for low-level action generation that alternates between online RL and supervised learning stages, significantly enhancing the generalization capability of VLA models. Additionally, Clark et al. (2025) harness extensive human video data to augment reasoning capabilities. This approach employs Gemini to synthesize reasoning steps from human demonstrations. Subsequently, the method leverages limited robot data to train the model in mapping abstract reasoning to low-level actions, while the action-free datasets serve to enhance the model's overall reasoning proficiency. Moreover, Gemini Robotics et al. (2025) train Gemini Robotics-ER based on Gemini 2.0, a VLM that demonstrates enhanced embodied reasoning capabilities. They subsequently extend this work to develop Gemini Robotics, a VLA model that integrates robot action data. This integration enables high-frequency dexterous control, robust generalization, and rapid adaptation across diverse robotic tasks and embodiments.

While numerous studies have successfully elicited long CoT reasoning to enhance performance, there remains significant room for improvement in embodied AI. First, although pure RL has proven effective in LLMs for fostering self-reflection through self-exploration (DeepSeek et al., 2025), a viable approach for embodied AI has yet to be established. Second, existing frameworks typically decouple high-level planning from low-level policy. However, a more effective paradigm would integrate planning and execution into a unified process, enabling continuous optimization through internal reasoning and iterative feedback.

---

**Future Direction for Embodied AI**

- Develop unified frameworks that seamlessly integrate high-level planning and low-level policy execution into a continuous process, enabling agents to optimize performance through internal reasoning loops and real-time iterative feedback during physical environment interactions.

- Establish fine-grained evaluation frameworks that systematically isolate and assess embodied agents' effectiveness across thinking, planning, and execution processes, providing deeper insights into how individual components contribute to overall performance in physical-world interactions.

---

## 6.6 Safety

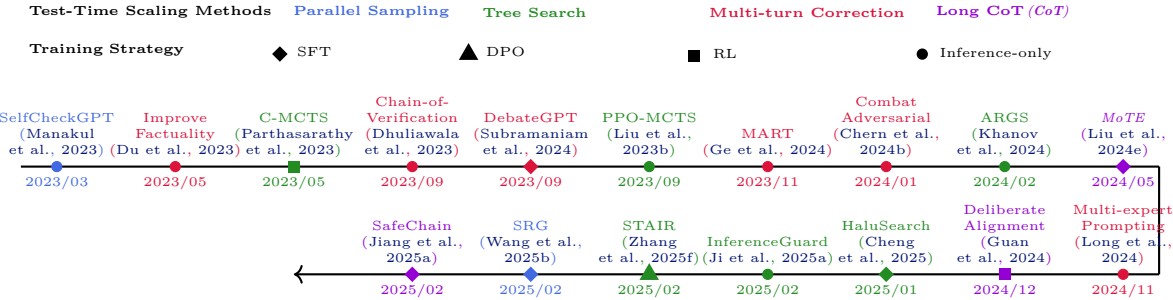

Figure 15: Works of applying test-time scaling methods in the safety field.

The advancement of AI systems with the ability to conduct complex, long-horizon reasoning as a result of test-time scaling carries significant implications for AI safety (Bengio, 2023; Park et al., 2024). The consequences are two-fold: AI systems that evolve to successfully solve highly complex problems can help address and identify emerging safety concerns themselves, potentially detecting and mitigating risks that would otherwise remain unnoticed by humans, which allows for more thorough exploration of edge cases and vulnerabilities (OpenAI, 2024). However, these same AI systems may also explore decisions that exceed human cognitive limits, potentially pursuing strategies misaligned with human values that could cause catastrophic consequences due to unmanageable, prolonged exchanges (Hendrycks et al., 2023). Below, we address how scaling test-time thinking has helped with identifying and mitigating safety risks (such as hallucination, jailbreaks, adversarial attacks, and more) in LLMs.

Recent research has increasingly explored parallel sampling as a mechanism for improving safety reasoning in LLMs, particularly in settings where retraining or internal access is impractical. The core insight is that sampling multiple responses to the same input enables more reliable estimation of factuality, uncertainty, and safety alignment. SelfCheckGPT (Manakul et al., 2023) introduces a zero-resource hallucination detection method that uses sampling to detect factual inconsistencies across generations, under the assumption that reliable knowledge yields consistent responses. Similarly, Lin et al. (2024) formalize this principle through the lens of semantic dispersion, showing that uncertainty estimates based on the diversity of sampled responses can reliably predict the trustworthiness of a model's output, even under black-box access. While these methods focus on sampling in their frameworks, SRG (Wang et al., 2025b) proposes injecting explicit reasoning steps guided by predefined safety policies into the models. Conducting evaluation with BoN sampling, they demonstrate improvements in generalization performance against OOD attacks. These approaches exemplify a powerful paradigm of test-time scaling: the ability to amplify alignment, robustness, and factual reliability by operating over multiple outputs during inference, rather than modifying the underlying model weights. Parallel sampling thus serves as a latent knowledge elicitation strategy that decouples safety improvements from expensive and infrastructure-heavy training processes, suggesting a scalable path forward for enhancing trustworthiness in real-world LLM deployments.

Apart from parallel sampling strategies, tree search-based methods offer a structured approach to test-time safety by enabling deliberate exploration of alternative outputs or reasoning paths. Techniques such as InferenceGuard (Ji et al., 2025a) and ARGS (Khanov et al., 2024) frame generation as a reward-constrained search process, steering outputs toward safety-aligned completions using constrained Markov decision process within the LLM's latent space or learned reward models. Other methods like HaluSearch (Cheng et al., 2025) improve factual reliability and reasoning robustness by explicitly searching over intermediate thought steps and scoring them using self-evaluation or heuristics (deciding when models should "think slower" versus fallback to fast generation). Planning-focused approaches such as C-MCTS (Parthasarathy et al., 2023) and STAIR (Zhang et al., 2025f) integrate safety critics or introspective mechanisms into MCTS to avoid unsafe action sequences during decision making. These methods enhance test-time alignment by enabling backtracking and surfacing interpretable intermediate decision steps—key advantages for maintaining safety as models scale.

Building on this foundation, recent research has also explored how increasing interaction steps during inference—through multi-turn correction or multi-agent collaboration—can further improve safety and robustness. Several multi-agent interaction frameworks (Du et al., 2023; Ge et al., 2024; Chern et al., 2024b; Long et al., 2024) have proven effective in reducing harmful outputs by enabling models to critique and refine responses collaboratively to identify and mitigate potential safety risks, such as reducing hallucination, toxicity, and adversarial attacks. Further, encouraging divergent thinking through structured debate (Liang et al., 2023) enhances reliability by promoting nuanced reasoning and cross-validation of outputs. Additionally, recent work leverages long CoT reasoning to enhance safety in LLMs by extending the depth of model deliberation at test time. SafeChain (Jiang et al., 2025a) introduces a training dataset of long-form safety-aligned reasoning and shows that LLMs fine-tuned on these trajectories can maintain high reasoning ability while improving refusal rates and reduces harmful content. Deliberative Alignment (Guan et al., 2024) trains models to explicitly consult and reason over safety policies via multi-step CoT before responding, demonstrating models that "think before speaking" become safer as reasoning depth increases—linking CoT length directly to improved alignment under test-time scaling. Similarly, Chain-of-Verification (Dhuliawala et al.,

2023) enhances factual safety by prompting the model to generate and answer verification questions about its own output, turning the reasoning process into a multi-phase, self-auditing chain—which becomes more reliable as models grow and can handle longer CoT. Lastly, MoTE (Liu et al., 2024e) decomposes safety reasoning into specialized CoT stages (question analysis, guidance, answer, and checking), assigning expert modules to each; this modular approach benefits from larger models and longer reasoning chains, enabling more scalable and interpretable self-alignment.

These studies collectively suggest that strategically increasing interaction and reasoning steps at inference time offers a scalable path to safer and more robust model behavior that doesn't require retraining.

---

**Future Direction for Safety**

- Investigate how test-time scaling methods can be integrated with existing alignment strategies, such as RLHF and process-based supervision, to ensure they complement each other and understand how they can collectively enhance model safety.

- Investigate the extent to which test-time scaling methods improve a model's generalization to real-world safety challenges, including deception, adversarial attacks, and other out-of-distribution threats.

- Conduct rigorous testing to determine the resilience of long-CoT models against various jailbreaks and attacks. This includes analyzing whether increased test-time computation inadvertently introduces new vulnerabilities or mitigates existing ones.

---

## 6.7 Retrieval-Augmented Generation

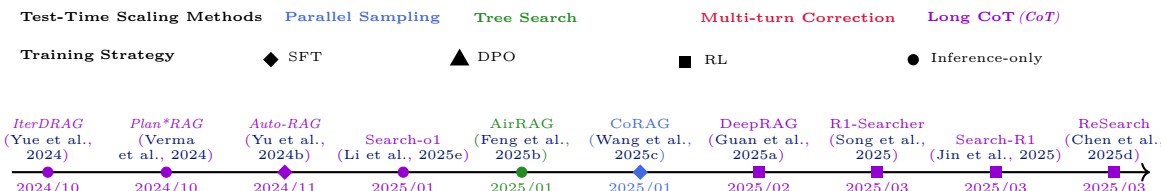

Figure 16: Works of applying test-time scaling methods in the RAG field.

Retrieval-Augmented Generation (RAG) systems enhance LLMs by incorporating external knowledge sources, producing responses that are more factual and contextually grounded. Despite their effectiveness, these systems often struggle when confronted with complex queries requiring multi-hop reasoning across multiple documents. Test-time scaling has emerged as a promising approach to strengthen the reasoning capabilities of RAG systems by strategically allocating additional computational resources during inference.

Yue et al. (2024) introduce IterDRAG, which methodically decomposes complex questions into sequential sub-queries and conducts iterative search and reasoning processes to construct comprehensive answers. Their research demonstrates a near-linear relationship between RAG performance and effective context length, establishing a clear test-time scaling law for RAG systems. The effectiveness of this agentic workflow has been further validated in several follow-up studies (Verma et al., 2024; Guan et al., 2025a; Li et al., 2025e; Feng et al., 2025b; Yu et al., 2024b; Wang et al., 2025c). Beyond prompt-based agents, researchers have developed approaches to fine-tune LLMs for end-to-end interleaved reasoning and search capabilities. One approach for training data collection involves synthesizing reasoning and search trajectories through rejection sampling, followed by model training via SFT (Wang et al., 2025c; Yu et al., 2024b) or preference fine-tuning (Guan et al., 2025a). More recently, RL has been applied for end-to-end training of RAG, with works such as R1-Searcher (Song et al., 2025), Search-R1 (Jin et al., 2025), and ReSearch (Chen et al., 2025d). These approaches enable models to learn more efficient search strategies through trial and error rather than imitating human-designed search patterns. However, a significant limitation is that these works primarily

focus on open-domain question answering tasks and rely on rule-based rewards designed for short, factual answers, which may not generalize well to more complex reasoning tasks requiring detailed explanations.

For future directions, an important aspect lies in developing more sophisticated reward functions that can evaluate long-form generation rather than just short factual answers for RL training. Moreover, current benchmarks do not separately assess the benefits of internal reasoning and external search in RAG systems, highlighting the importance of developing specialized evaluation frameworks for RAG in this new era.

> **Future Direction for RAG**
>
> - Develop more sophisticated reward functions for RL that can effectively evaluate long-form generation rather than focusing solely on short factual answers.
>
> - Create specialized evaluation frameworks that isolate and measure the distinct contributions of internal reasoning versus external search in RAG systems.

## 6.8 Evaluation

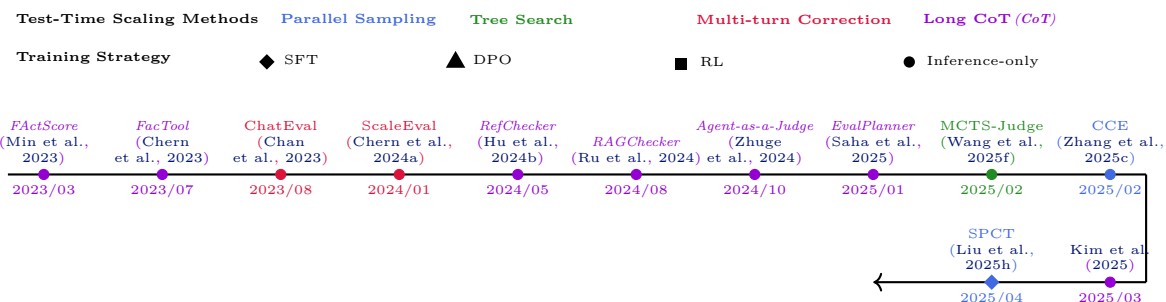

Figure 17: Works of applying test-time scaling methods in the evaluation field.

The LLM-as-a-Judge paradigm (Zheng et al., 2023a; Gu et al., 2024b) has transformed the evaluation of language model outputs, shifting away from rule-based metrics like BLEU and ROUGE toward more human-like assessment of generated content. Recent research demonstrates that allocating additional computational resources during inference significantly enhances evaluation quality through several approaches. These include fine-grained evaluation that decomposes the LLM's response and examines it step by step (Chern et al., 2023; Min et al., 2023; Hu et al., 2024b; Ru et al., 2024), structuring the CoT in LLM-as-a-Judge into distinct planning and execution phases (Saha et al., 2025), employing multi-agent systems to provide intermediate feedback (Zhuge et al., 2024), and utilizing parallel sampled crowd responses for more reliable pairwise comparisons (Zhang et al., 2025c). Furthermore, MCTS-Judge (Wang et al., 2025f) applies MCTS to systematically explore different evaluation perspectives for code assessment, demonstrating the scalability of this method where increasing search depth and rollouts consistently improves accuracy. Kim et al. (2025) also observe that generating more reasoning tokens leads to better performance of the evaluators for long-CoT models.

As AI development progresses toward more realistic real-world tasks such as software development which comprises multiple subtasks, current evaluation frameworks increasingly focus on complex agents and workflows (Jimenez et al., 2024; Xie et al., 2024a). Future research directions should emphasize enhancing the reliability of evaluation for these complex and long-horizon tasks, which require strategies to balance the benefits of test-time scaling and evaluation speed.

> **Future Direction for Evaluation**
>
> - Enhance the reliability of evaluation for complex and long-horizon tasks.
>
> - Integrate evaluation frameworks with reward design in RL training to fully unleash the potential of RL.

Table 11: Comparison between OpenRLHF and veRL frameworks.

| Framework | Supported Algorithm | Hybrid Engine | Training Backend | Inference Engine | Multi-Modality |
|-----------|---------------------|---------------|------------------|------------------|----------------|
| **OpenRLHF** | PPO, GRPO, RLOO, REINFORCE++ | Supported | Deepspeed ZERO | vLLM, HF Transformers | Supported |
| **veRL** | PPO, GRPO, RLOO, REINFORCE++, DAPO, PRIME | Supported | Megatron-LM, FSDP | vLLM, SGLang, HF Transformers | Supported |

## 7 Infrastructure

Infrastructure is crucial for the large-scale deployment of the previously mentioned technologies. This section briefly discusses the infrastructure for RL and MCTS, two important and active fields, to help readers grasp current developments and future directions. Besides these, the acceleration of long text generation is also important. We refer interested readers to the survey (Liu et al., 2025b) for a comprehensive review.

### 7.1 RL

Taking the PPO algorithm as the example, a typical RL workflow consists of two primary steps:

- **Rollout:** Prepared prompts are fed into the policy model, which generates responses (i.e., rollout). This process requires the inference backend of the policy model. Although conventional training libraries support inference, they often exhibit slow decoding speeds when generating new sequences. To address this limitation, a dedicated inference backend (e.g., vLLM (Kwon et al., 2023) or SGLang (Zheng et al., 2023b)) is typically deployed to accelerate rollout for the policy model.

- **Model Update:** Given the generated sequences, policy log probabilities, value estimates, reference log probabilities, and rewards are computed using the policy, critic, reference model, and reward function, respectively. Additional values, such as KL divergence loss, return, and advantage, are then derived from these computations. These computed values are then used to calculate the respective losses for the policy and critic models, which are then applied to update their parameters. The training backend for model updates typically employs frameworks such as DeepSpeed (Rasley et al., 2020) or Fully Sharded Data Parallel (FSDP) (Zhao et al., 2023c).

Popular open-source frameworks for RL training include DeepSpeed-Chat, NeMo-Aligner (Shen et al., 2024), OpenRLHF (Hu et al., 2024a), and veRL (Sheng et al., 2024). Among these, OpenRLHF and veRL are actively maintained frameworks that support RL training, with their key features summarized in Table 11. Due to the complex workflow involving multiple models described above, these RL frameworks implement different strategies for resource allocation and process scheduling. OpenRLHF, for instance, can allocate dedicated resources to each backend, ensuring operations within a given module remain confined to its designated resource pool. In contrast, veRL primarily employs a shared-resource approach, dynamically reclaiming resources from inactive modules when another requires them. OpenRLHF offers a more concise code implementation, making it more accessible for beginners to understand the algorithm, while veRL provides a relatively centralized programming interface with enhanced programmability.

Although these open-source frameworks facilitate RL training, large-scale RL training can still present significant challenges. For example, Yeo et al. (2025) encountered substantial difficulties when attempting to scale the policy model to 32B parameters, ultimately determining that the required number of GPUs was prohibitively large. They observed low hardware utilization during the training process, an issue that is particularly exacerbated in long CoT scenarios due to the higher variance in CoT length, which leads to stragglers during inference. This highlights the ongoing need for optimization of RL frameworks specifically for these scenarios.

## 7.2 MCTS

For the infrastructure of MCTS, several of the aforementioned works have released their code to provide a foundation for applying MCTS in LLMs (Hao et al., 2024a; 2023; Chen et al., 2024a; Feng et al., 2023). Although these code repositories facilitate subsequent research, they often lack optimized acceleration strategies that consider hardware and software optimization, which limits large-scale MCTS deployment. We outline the key acceleration strategies as follows:

**Speculative Decoding**  Speculative Decoding employs a small draft model to generate tokens sequentially, with a larger target model validating these tokens, which is widely implemented to accelerate the rollout speed (Gao et al., 2024b; Wang et al., 2024h). SEED (Wang et al., 2024h) implements scheduled speculative decoding, which efficiently manages both runtime speed and GPU memory usage simultaneously. The framework leverages a rounds-scheduled strategy that manages the execution flow using a First-Come-First-Serve queue to control verification of the target model without conflicts. SC-MCTS (Gao et al., 2024b) utilizes speculative decoding to speed up MCTS reasoning by an average of 52% as a "free lunch."

**KV Cache Management**  LLM inference is typically memory bandwidth-bound (Hooper et al., 2024). In tree search, each unique trajectory requires a separate KV cache state, creating a significant memory bottleneck. DEFT (Yao et al., 2024b) introduces efficient kernel implementations that compute attention with tree-structured KV sharing. Hydragen (Juravsky et al., 2024) and vLLM (Kwon et al., 2023) offer support for shared prefix workloads, effectively eliminating KV cache duplication. SGLang (Zheng et al., 2023b) implements Radix Attention, which stores and dynamically references reused KV cache segments. Additionally, ETS (Hooper et al., 2025) employs a linear programming cost model that encourages KV cache sharing by penalizing node retention while incorporating semantic coverage parameters to maintain diversity among retained trajectories.

**Parallel Processing**  The acceleration of tree expansion and simulation phases can be achieved through parallel processing techniques. However, the frequent switches among paths complicates the parallelism. Ding et al. (2025) develop a flexible and adaptive parallelism system for arbitrary paths by implementing fine-grained cache management and alignment during the generation phase. The system adjusts the number of parallel paths processed based on real-time GPU memory availability, optimizing resource utilization.

## 8 Future Directions

Building on the three pillars, including the knowledge foundation, test-time scaling methods, and corresponding self-training strategies, cognition engineering has significantly advanced the deep reasoning abilities of LLMs. In this section, we identify critical future directions for each of these aspects to substantially accelerate the development of cognition engineering.

**Pretraining on latent thought**  For the first pillar, the knowledge foundation, current pretraining data primarily consists of human-written texts but lacks the latent thought processes behind them. The success of RL scaling on models pretrained with data containing cognitive behaviors demonstrates the potential of including human thinking processes (Gandhi et al., 2025; Liu et al., 2025g;f). Recent works have shown the benefits of incorporating hidden thinking processes beyond explicit text (Zelikman et al., 2024a; Jiang et al., 2024; Ruan et al., 2025), though these are still limited to small-scale experiments. Future research should focus on acquiring large-scale data rich in cognitive behaviors through techniques such as inferring latent thoughts by utilizing existing reasoning models and examining how pretraining on such data could benefit test-time scaling methods.

**New architecture**  For the second pillar, test-time scaling, one key scaling dimension is the response length. Transformer-based architectures face fundamental limitations due to their linear memory scaling and memory-bound nature when generating long contexts. While the methods described in the improving scaling efficiency section can alleviate this problem, a more fundamental solution requires exploring new architectures. Promising alternatives include state space models like Mamba (Gu & Dao, 2023; Dao &

Gu, 2024), which offers linear-time complexity for sequence modeling, linear transformers that reduce the quadratic attention bottleneck (Katharopoulos et al., 2020), and even language diffusion models (Nie et al., 2025). This architectural transformation requires comprehensive system engineering efforts across multiple dimensions: developing robust theoretical frameworks for new architectures, building infrastructure support for efficient training and inference, and creating large-scale pretrained foundation models based on these architectures. The integration of these architectures with cognition engineering technology could significantly enhance both cognitive capabilities and computational efficiency.

**RL scaling**   For the third pillar, self-training, RL scaling is a core component due to the high performance demonstrated in recent works. While we have examined the common design principles of RL scaling based on the latest research, the field remains in its early development stage, holding significant potential to unlock the cognitive abilities of AI. Given the complex components of RL and the numerous hyperparameters requiring tuning, future research should adopt a more rigorous approach when drawing conclusions that account for all these elements (Jordan et al., 2024; Hochlehnert et al., 2025). Moreover, reproducible open-source work is currently limited to small models and datasets, which constrains the scope of possible conclusions. Large-scale experiments require both infrastructure improvements and algorithm optimizations to become accessible to researchers with modest computational resources. Furthermore, current RL scaling primarily focuses on verifiable tasks such as mathematics and code. Expanding to broader domains necessitates deeper investigation into reward hacking phenomena and establishing clearer relationships between reward reliability and RL scaling. This advancement demands not only empirical investigation but also theoretical analysis.

## 9   Comparison to Existing Work

Our paper combines aspects of both a comprehensive survey and a position paper. From the survey perspective (§4-§7), we summarize and compare existing works in Table 12. Our survey differs from previous works in three key aspects. First, we organize the content from a test-time scaling perspective, which differs from works focused on either system-2 reasoning (Li et al., 2025i) or long CoT (Chen et al., 2025e; Ji et al., 2025b). Building on this focus, we offer detailed discussions of scaling laws, scaling efficiency for four primary test-time scaling methods, comparisons between these methods, and their ensemble applications. While Welleck et al. (2024) discuss test-time scaling from the meta-generation perspective, their coverage of these critical aspects for utilizing test-time scaling methods remains limited and they do not include the long-CoT technique. Concurrent to our work, Zhang et al. (2025b) focus on test-time scaling but offer insufficient discussion of these vital topics. Other concurrent works such as Sui et al. (2025); Qu et al. (2025a) focus on scaling efficiency but restrict their analysis to partial test-time scaling methods, primarily long CoT. Second, we provide detailed, multi-faceted discussions of RL for long-CoT techniques and SFT for long-CoT based on recent research, offering more concrete and specific insights than previous works' vague discussions (Tie et al., 2025; Besta et al., 2025; Xu et al., 2025a). Third, we cover a broader range of applications for test-time scaling and provide thorough discussion and future directions for each.

From the position paper perspective, we propose the concept of cognition engineering and argue that generative AI has evolved into an era focused on developing deep cognitive capabilities in models. This conceptual framework helps unify various technologies that enhance model cognitive abilities and includes potential future directions like pretraining on latent thought. This feature also distinguishes our work from other survey papers.

## 10   Conclusion

Cognition engineering represents a paradigm shift in AI development, fundamentally transforming our approach from knowledge accumulation to the systematic development of thinking capabilities. This second act of generative AI leverages test-time scaling methodologies alongside specialized training strategies to enable models to engage in deep thinking, complex reasoning, and creative problem-solving. As demonstrated across domains from mathematics to multimodal understanding, these capabilities are already yielding substantial improvements in model performance. The journey into this second act is just beginning, and the continued

Table 12: Comparison to existing surveys. △ denotes limited discussion of the topic. **TTS** denotes Test-time Scaling. **Law** denotes Scaling Laws. **Efficiency** denotes Scaling Efficiency.

| Work | TTS Types | | Law | Efficiency | Comparison | Ensemble | Long-CoT RL | Long-CoT SFT |
|---|---|---|---|---|---|---|---|---|
| | **Long CoT** | **Others** | | | | | | |
| Welleck et al. (2024) | ✗ | ✓ | △ | △ | △ | ✗ | ✗ | ✗ |
| Zeng et al. (2024) | ✓ | ✓ | ✗ | ✗ | ✗ | △ | ✗ | ✗ |
| Ji et al. (2025b) | ✗ | ✓ | △ | △ | ✗ | ✗ | ✗ | ✗ |
| Li et al. (2025i) | ✓ | ✓ | ✗ | ✗ | ✗ | ✗ | △ | ✗ |
| Kumar et al. (2025) | ✓ | ✓ | ✗ | ✗ | ✗ | ✗ | △ | △ |
| Chen et al. (2025e) | ✓ | ✓ | ✗ | ✗ | ✗ | △ | ✗ | △ |
| Zhang et al. (2025b) | ✓ | ✓ | ✗ | ✗ | △ | ✗ | ✗ | △ |
| Ours | ✓ | ✓ | ✓ | ✓ | ✓ | ✓ | ✓ | ✓ |

development of cognition engineering promises a future where AI systems can reason, create, and discover in true partnership with humanity.

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
