# OpenReview forum: "Generative AI Act II: Test Time Scaling Drives Cognition Engineering"
_TMLR — Rejected by TMLR_

### Review · Reviewer_foa3 · 2025-05-16

**Summary Of Contributions:**

I read this work as a position paper/survey that proposes new research directions in test-time scaling. Namely, the paper proposes "cognition engineering" as a generalization of test-time scaling (i.e., teaching LLMs how to "think" at inference time). Then, the main contributions are:
* A definition of cognition engineering, based on Data-Information-Knowledge-Wisdom theory, in which each "layer" imposes additional structure/contextualizaiton of the previous (Section 2),
* A motivational literature review highlighting gaps between status quo LLM capabilities and cognition engineering (Section 3.1), as well as techniques that (are claimed to) provide a foundation cognition engineering (Section 3.2),
* A detailed literature review of test-time scaling approaches (Section 4), training strategies that facilitate test-time scaling (Section 5), and domain-specific progress (Section 6),
* A review of current infrastructure for certain approaches (Section 7),
* and a position on future directions (Section 8).

**Audience:**

Yes

**Claims And Evidence:**

No

**Requested Changes:**

* I would like to see improved precision for the main concepts outlining the work. Is there a more precise definition of "cognitive engineering?" If not mathematical, are there high-level, concrete/operationalizable principles of cognitive engineering that the authors advocate for in LLM development?
* The sections can be significantly reorganized for clarity. I think Sections 2-3 and be combined into a motivating section establishing what cognitive engineering is, and why it is important.
* Sections 4-6 could be either cut down or reorganized. To my above point about "focus:" I think I was missing context as to *why* we were discussed the selected test-time scaling methods in the context of improving cognitive engineering.
    - In particular, something paralleling the organization of 3.2 (Three Pillars) could be easier to follow, and the related works sections should connect back to the theme of cognitive engineering. As-is, many of the related works subsections in Sections 4-6 almost read like self-contained tutorials/overviews of the subject matter.
    - The missing piece is discussion on how each of the discussed approaches lays a foundation for cognitive engineering, and I suspect improving the precision of cognitive engineering in Section 2/outlining high-level, concrete principles of cognitive engineering up front will make these connections easier.
* Section 7 might be removable completely.
* Section 8 is a bit disjointed. Again, the missing piece is connections back to cognitive engineering; i.e., what are the limitations/what is the status quo of current research in each of the areas, and what specific problems should we solve on the way to improving cognitive engineering? I encourage the authors to use Section 8 as an opportunity to create a cohesive roadmap for the LLM research community to follow; that is, what are the ladders/paths to cognitive engineering that the previous work (Sections 4-6) has revealed? What are the specific research questions to solve here, and what progress has been made currently?

**Strengths And Weaknesses:**

**Strengths.**
* The overview of existing test-time scaling methods and potential strategies is mostly clear.
* The general topic is timely; suggesting research directions for more deeply "understanding" knowledge encoded in training data would be welcome in the LLM research community.
* The literature review is very comprehensive.

**Weaknesses.**
* The overarching framing/definition (cognition engineering) is imprecise, leaving room for misinterpretation/subjectivity. For example: "Cognition Engineering is the systematic and constructive development of AI thinking capabilities through test-time scaling paradigms that transcend traditional pretraining approaches. This methodology represents the deliberate cultivation of deep cognitive processes in artificial systems through both human cognitive pattern distillation and AI-driven discovery (e.g., reinforcement learning)." The precision of this definition can be improved in several places, such as:
    - What does it mean to "transcend" traditional pretraining approaches? How does this differ from current test-time scaling paradigms (i.e., why do they not "transcend" traditional approaches)?
    - What is "deliberate cultivation of deep cognitive processes?" By cultivation, do we mean some form of training, test-time scaling, or something else? What is a "deep cognitive process;" namely, how does it differ from the current behaviors or "cognitive processes" exhibited by LLMs?
While I'm aware that some of these questions are answered implicitly or directly later in the work, the lack of precision up-front makes it difficult to follow the survey (i.e., why are the topics discussed relevant).
* To that end, providing more concrete examples for Section 2.3 for characteristics of cognitive engineering would be helpful (e.g., what is a specific task that would exhibit successful cognitive engineering? where do LLMs currently fall short?)
* The writing style skews toward dramatic or rhetorical language. For a scientific paper, even a position paper or survey, a more neutral tone focused on precise claims and clearly stated hypotheses would strengthen the presentation and improve clarity.
* The focus of this work could also be improved. As is, it's a little difficult to evaluate: despite the impressive comprehensiveness of the literature review, and the methodological organization, the overall structure remains unclear. Examples of review papers I have enjoyed generally pose well-defined research directions/questions, and have well-structured, overarching themes that are motivated from the introduction, such as [[1]](https://arxiv.org/pdf/2108.07258), [[2]](https://arxiv.org/abs/2404.09932), or [[3]](https://arxiv.org/abs/2307.15217).

---

> ### Author Response · Authors · 2025-08-02
> **Response to Reviewer foa3 (1/3)**
>
> Thank you for your thoughtful review and constructive feedback. We sincerely appreciate your positive comments on our paper's comprehensive survey, timely topic, and clear overview of the technology. We have provided point-by-point responses to your concerns below.
>
> > **Requested Changes 1:** I would like to see improved precision for the main concepts outlining the work. Is there a more precise definition of "cognitive engineering?" If not mathematical, are there high-level, concrete/operationalizable principles of cognitive engineering that the authors advocate for in LLM development?
>
>
> **Response:** Thank you for this excellent suggestion to improve the precision of our core concepts. You asked for a more concrete, operationalizable definition of "Cognition Engineering," and we agree this is crucial for the paper.
> In response, we have made the following revisions:
> - In Section 1, we revised our definition. The new definition is now more operational by framing "Cognition Engineering" as the deliberate cultivation of deep reasoning processes in LLMs through two primary principles: 1) interventions at inference time (the topic of Section 4), and 2) corresponding training strategies (the topic of Section 5). This change can be found on page 4 of the revised manuscript.
> - In section 2, we have removed subsections 2.2 and 2.3 entirely. The remaining content has been streamlined and reorganized to better support the paper's main focus on developing the deep reasoning capabilities of LLMs. This change can be found on pages 5-6 of the revised manuscript.
>
> Regarding the questions you raised in the "Weaknesses 1" section, we address them directly below:
>
> > **Q1:** What does it mean to "transcend" traditional pretraining approaches?
>
> **A1:** Thank you for the clarifying question. Our use of "transcend" is meant to describe a paradigm shift from traditional pretraining towards the test-time scaling methods we survey.
>
> > **Q2:** How does this differ from current test-time scaling paradigms (i.e., why do they not "transcend" traditional approaches)?
>
> **A2:** In our framework, the test-time scaling paradigms are the new approach. Therefore, the object being transcended is the reliance on a pretraining-only approach, not the test-time scaling methods themselves. We have reviewed the text to ensure this is clear.
>
> > **Q3:** What is a "deep cognitive process;" namely, how does it differ from the current behaviors or "cognitive processes" exhibited by LLMs?
>
> **A3:** In our revised manuscript, we clarify that by "deep cognitive processes," we are referring to the advanced deep reasoning capabilities that we aim to cultivate. A detailed description of these target capabilities is provided in the reorganized Section 2. These processes are further defined by contrasting them with the limitations of first-generation LLMs, which we detail in Section 3.1.
>
> > **Q4:** What is "deliberate cultivation of deep cognitive processes?" By cultivation, do we mean some form of training, test-time scaling, or something else?
>
> **A4:** "Deliberate cultivation" refers to the two primary principles (or methods) we survey for developing these deep reasoning processes. As outlined in our revised definition in Section 1, these are: 1) interventions at inference time (the topic of Section 4), and 2) corresponding training strategies (the topic of Section 5).
>
> We hope these revisions and clarifications successfully address your concerns and make the paper's foundational concepts more precise and well-supported.

---

> ### Author Response · Authors · 2025-08-02
> **Response to Reviewer foa3 (2/3)**
>
> > **Requested Changes 2:** The sections can be significantly reorganized for clarity. I think Sections 2-3 and be combined into a motivating section establishing what cognitive engineering is, and why it is important.
>
> >  **Requested Changes 3:** Sections 4-6 could be either cut down or reorganized. To my above point about "focus:" I think I was missing context as to why we were discussed the selected test-time scaling methods in the context of improving cognitive engineering.
> >- In particular, something paralleling the organization of 3.2 (Three Pillars) could be easier to follow, and the related works sections should connect back to the theme of cognitive engineering. As-is, many of the related works subsections in Sections 4-6 almost read like self-contained tutorials/overviews of the subject matter.
> > - The missing piece is discussion on how each of the discussed approaches lays a foundation for cognitive engineering, and I suspect improving the precision of cognitive engineering in Section 2/outlining high-level, concrete principles of cognitive engineering up front will make these connections easier.
>
> **Response:** Thank you for this very constructive and detailed feedback on the paper's structure and focus. We are addressing your suggestions from Requested Changes 2 and 3 together here, as they are closely related. We agree that the connections between our "Cognition Engineering" framework and the survey sections needed to be much more explicit to guide the reader. Based on your valuable advice, we have made several revisions to the introductory sections to create a stronger and more cohesive narrative thread.
> - In Section 2, we have removed subsections 2.2 and 2.3 and streamlined the remaining content to help establish the concept of cognition engineering more clearly.
> - In Section 3, we made two main revisions to better connect the introduction with the survey content:
>   - We added a new roadmap figure (Figure 1) to visually illustrate the overall structure and the relationship between our framework and the survey topics.
>   - We also revised Section 3.3 to explicitly explain how the following survey sections align with Cognition Engineering.
>
> We believe that with these changes, the introduction now provides the necessary context, and the "Cognition Engineering" framework serves as a much more effective tool for organizing the information that follows. These revisions can be found on pages 5-7.
>
>
> > Requested Changes 4: Section 7 might be removable completely.
>
> **Response:** Thank you for this suggestion regarding Section 7. We appreciate your perspective and agree that the core of the survey lies in Sections 4-6. After careful consideration, we have decided to retain Section 7. Our rationale is that the infrastructure discussed in this section is a critical enabling factor for the large-scale deployment of the technologies previously surveyed. We believe that including this discussion provides a more complete picture and draws necessary attention to this essential aspect of the field. However, we also agree that the section's relevance and purpose could be stated more clearly.
>
> To address this, we have revised the introduction to Section 7 to explicitly establish its goals and better connect it to the preceding discussion. This revision can be found on page 46, highlighted in blue.

---

> ### Author Response · Authors · 2025-08-02
> **Response to Reviewer foa3 (3/3)**
>
> > **Requested Changes 5:** Section 8 is a bit disjointed. Again, the missing piece is connections back to cognitive engineering; i.e., what are the limitations/what is the status quo of current research in each of the areas, and what specific problems should we solve on the way to improving cognitive engineering? I encourage the authors to use Section 8 as an opportunity to create a cohesive roadmap for the LLM research community to follow; that is, what are the ladders/paths to cognitive engineering that the previous work (Sections 4-6) has revealed? What are the specific research questions to solve here, and what progress has been made currently?
>
> **Response:** Thank you for this excellent and very constructive suggestion on how to strengthen Section 8 into a cohesive research roadmap. Building on our revised introduction, we have restructured Section 8 to directly map the future directions to the "Three Pillars" introduced in Section 3.2. For example, Future Direction 1 now corresponds to Pillar 1, Future Direction 2 to Pillar 2, and so on. We hope this structure makes Section 8 a clear and helpful guide for future research in the field. These revisions can be found on pages 47-48, highlighted in blue.
>
>
>
> >**Weakness2:** The writing style skews toward dramatic or rhetorical language. For a scientific paper, even a position paper or survey, a more neutral tone focused on precise claims and clearly stated hypotheses would strengthen the presentation and improve clarity.
>
> **Response:** Thank you for this valuable feedback on the paper's writing style. We agree that a more neutral and precise tone is essential for a scientific survey. To address this, we have carefully revised the introductory sections (1-2). Throughout this revision, we have worked to replace dramatic phrasing with well-supported statements. Our key modifications include:
> - In Section 1, we have removed the particular speculative language and have also reviewed the entire section to ensure a uniform and objective tone.
> - In Section 2, we have restructured the section, removing subsections 2.2 and 2.3 entirely. The remaining content has been refocused and revised to improve its scientific precision.
>
> These revisions can be found on pages 4-6 of the updated manuscript.

---

### Review · Reviewer_9Vyp · 2025-05-31

**Summary Of Contributions:**

This paper provides a comprehensive survey of test-time scaling methods for improving large language-model (LLMs). Beyond the survey aspect, the paper's key claim is the development of LLMs is transitioning from an "Act I" that focuses on prompt engineering where the goal is knowledge retrieval, to an "Act II" that focuses on cognition engineering where the goal is better reasoning.

"Cognition engineering" is defined by the paper as the model's thinking abilities developed via test-time scaling using approaches beyond pre-training. This is based on a conceptual framework of a Data-Information-Knowledge-Wisdom pyramid from cognitive science, with cognition engineering striving to advance model capabilities from Knowledge to Wisdom at the top of the pyramid. Wisdom encompasses cognitive abilities such as judgement, creativity, and meta-cognitive abilities.

The rest of the paper is a detailed and well organized survey of the three broad pillars of cognition engineering that currently exist: knowledge (e.g., large model pre-training and fine-tuning), test-time scaling (e.g., long-COT, best of N), and self-training (e.g., RL for reasoning). The survey covers a large number of relevant papers and is very well organized and structured, judiciously using tables and figures to compare different methods.

**Audience:**

Yes

**Claims And Evidence:**

Yes

**Requested Changes:**

[critical] Review Secs. 1 and 2 for over-claims or under-supported statements and either provide evidence to support the claims, or revise them to be more precise. See examples in "Weaknesses" but please double check other claims carefully.
[Strengthen] Reframe the paper as a straightforward survey of test-time scaling methods or methods for reasoning models. The current framing of Cognitive Engineering does not help much in organizing the information contained in the rest of the paper, nor does it help guide future research to a large degree.

**Strengths And Weaknesses:**

Strengths:
1. Survey is very comprehensive and covers a large number of existing papers (30 of 81 pages of the manuscript are references). Tables and figures and used judiciously to organize the vast amount of data surveyed, especially to compare the relative strengths and weaknesses of different methods.

2. This work is quite valuable as a survey and and "encyclopaedia" of test-time methods, both for researchers new to the field to get a grasp of the current state-of-the-art, and existing researchers looking for resources or related works.

Weaknesses:
1. Beyond the survey, the paper's framing of Cognition Engineering is not quite convincing. The paradigm shift to test-time methods for improving LLMs is perhaps one of the most well-known facts in the field (since the release of models such as o1 and DeepSeek-R1). Hence, the framing of Act I and Act II lack novelty.

2. Claims in the paper are vague and lacks concrete evidence. For example:

(Pages 4-5) "Not only can humans teach AI systems how to approach complex problems, but AI systems can also autonomously
discover novel cognitive patterns and reasoning pathways through techniques like reinforcement learning."

Are there any evidence of such an AI system discovering novel cognitive patterns? Which cognitive patterns?

(Following sentence) "For example, we have already witnessed moments reminiscent of AlphaGo’s famous “Move 37,” where AI demonstrates thinking approaches that transcend human intuition yet ultimately prove effective"

Again, what is the evidence for this? What are the "moments reminiscent"?

(Page 6) "cognition engineering aims to enable AI systems to generate new insights and discoveries through deep thinking" Which new insights or discoveries has AI systems made?

There are more imprecise or under supported claims like these in Sec. 1 and 2 and weakens the quality of this work.

---

> ### Author Response · Authors · 2025-08-02
> **Response to Reviewer 9Vyp**
>
> Thank you for your thoughtful review and constructive feedback. We sincerely appreciate your positive comments on our paper's comprehensive survey, high-quality figures and tables, and its potential value to researchers. We have provided point-by-point responses to your concerns below.
>
> > **Requested Changes 1:** Review Secs. 1 and 2 for over-claims or under-supported statements and either provide evidence to support the claims, or revise them to be more precise. See examples in "Weaknesses" but please double check other claims carefully.
>
> **Response:**  Thank you for this crucial feedback regarding the framing and rigor of our introduction. We agree that some of the initial claims were overly ambitious and required revision. In response, we have revised Sections 1 and 2 to remove under-supported statements and ensure the introduction is more precise and scientifically grounded. Our modifications include:
> - In Section 1, we have removed the specific speculative language you highlighted and have also carefully reviewed the entire section to ensure a consistent and scientific tone.
> - In Section 2, we have revised the section by removing subsections 2.2 and 2.3 entirely. These sections contained some of the broad claims you identified. The remaining content has been streamlined and carefully edited to improve its scientific rigor.
>
> These revisions can be found on pages 4-6 of the updated manuscript. We believe these changes create a stronger foundation for the main survey.
>
> > **Requested Changes 2:** Reframe the paper as a straightforward survey of test-time scaling methods or methods for reasoning models. The current framing of Cognitive Engineering does not help much in organizing the information contained in the rest of the paper, nor does it help guide future research to a large degree.
>
> **Response:** Thank you for this very helpful feedback on the paper's framing. We appreciate your point that our "Cognition Engineering" framework needed to do a better job of organizing the survey's content. In response, we have made structural revisions to the introduction (Sections 1-3) to provide a coherent structure for the survey:
> - In Section 1, we have adjusted the definition of "cognition engineering" to explicitly outline the structure of the survey. The new definition clearly mentions "interventions at inference time" (which maps to Section 4) and "corresponding training strategies" (which maps to Section 5).
> - In Section 2, we have further emphasized the core concept—developing the deep reasoning ability of LLMs—to ensure thematic consistency throughout the introduction.
> - In Section 3, we have made two improvements to guide the reader:
>   - First, we incorporated a new roadmap figure (Figure 1) to illustrate the main structure of our survey and the relationships between its parts.
>   - Second, we amended the text in Section 3.3 to provide explicit guidance for the reader, explaining how the following survey sections fit into our cognition engineering framework.
>
> We believe these revisions now better link the topics within our survey, improving its overall flow and readability. These revisions can be found on pages 4-7 of the updated manuscript.

---

### Review · Reviewer_cTi9 · 2025-07-20

**Summary Of Contributions:**

The authors survey the field of LLMs on how various literature improves their performance in different applications, such as math or programming. They position the work from the lens of cognitive engineering, by first defining this field and drawing connections between various aspects of LLMs and techniques to different requirements for cognitive engineering.

**Audience:**

Yes

**Broader Impact Concerns:**

I am unsure if a broader impact statement is required here, but of course, there are many ethical concerns with LLMs. There are numerous instances of LLMs being trained and deployed that practice hate speech, racism, sexism, etc. This push to make them "think more like a human" should be approached cautiously, as we aren't already necessarily doing that, and the above examples are what we have seen.

Consider my previous text on this:

Human thinking is likely not even logical. Humans lie, cheat, steal, murder, etc., and all of these activities use human thinking. Why would we want AIs that replicate human thinking? This could, and arguably should, be discussed in greater detail. In other words, we likely don't want *all* the properties of human thinking, only the "good" - but what are the good? How do we keep only the good parts of human thinking? How do we know that the AI systems aren't just predicting what we want them to predict to emulate thinking? What does it even mean to be morally good? Can we have an answer to such a question?

**Claims And Evidence:**

Yes

**Requested Changes:**

[Critical] Please clarify if the following is a direct quote or your own words (pg. 4). It is currently in a block quote, so it appears to be a quote without a proper citation or attribution:

Cognition Engineering is the systematic and constructive development of AI thinking capabilities through test-time scaling paradigms that transcend traditional pretraining approaches. This methodology represents the deliberate cultivation of deep cognitive processes in artificial systems through both human cognitive pattern distillation and AI-driven discovery (e.g., reinforcement learning)

[Improvement] Flip the way an acronym is defined; first use is the spelled-out version, then the acronym in parentheses: e.g., (pg. 5) Data-Information-Knowledge-Wisdom (DIKW)

[Improvement] Paragraph at the top of page 6: provide more citations behind some of these claims, definitions of cognition, etc., with more work in the field. Cognitive science must have some impactful, foundational, and historical works before 1995. Please cite those here.

[Critical] Why should we care about the DIKW pyramid theory? Where has it been helpful? Please explain and cite these.

[Improvement] Add a space between "perplexity" and "(Cui et al., 2025b) in paragraph "Finetuning on compressed responses using heuristic methods" on page 20.

[Improvement] pg. 19: "are as follows." Change the period to a colon - so far, and throughout most of the text, you use a colon to precede the list of items

[Improvement] You need to ensure you define all acronyms. "RL" is never defined, though it is obviously referring to reinforcement learning. It is considered best practice to follow. Same with "SFT" - these may be obvious today, but in the future, it may be less clear as the field changes and new terms are introduced

[Improvement] write out C3oT as "Conditioned Compressed Chain-of-Thought," then define the acronym

[Improvement] Be more consistent with when to use acronyms. There are instances where "RL" is used, and others (later on) where "reinforcement learning" is used. Same issue with "best-of-N" (pg. 24).

[Improvement] Table 4: "curriculum sampling strategy" is CSS, but the iterative lengthening strategy is only "IL" - please change IL to ILS.

[Improvement] Very hard to read Table 5 - not enough vertical space for the content. This would perhaps be better as a well-organized paragraph, with a more shortened table to help support the organization (less detail on method and evidence, simply pointing out the problem that is being solved in the related work).

[Improvement] I understand the reasoning behind the use of emojis in some parts of this article, but their frequent use in others feels misplaced and detracts from the article's professional appearance (e.g., Table 6); this is not a GitHub README.md file. For instance, the snowflake icon, to me, in Table 6 suggests that the architecture's parameters are frozen, which is certainly possible, but instead it indicates the model needs to perform inference. This is not very clear, although parameters are usually frozen during inference.

[Critical] Please clarify how you determine, with only a yes or no, whether the training algorithm achieves the RL scaling as reported in Table 6? That seems to be a qualitative, subjective judgment based on how the models demonstrate cognitive behaviors, including self-reflection and self-correction. So, where is your data showing this? Or whose data are you referring to? If the former, please present it. If the latter, please explain and cite it.

[Improvement] "As shown in Table 4, most RL scaling work utilizes Qwen2.5 as the base model." This is only shown as logos. This is not very accessible to reading-impaired individuals. You should at the very least map each logo to the text. Again, these companies may go bankrupt in the future, so their logos are perhaps not future-proof or may not be easily recognized in the future. Perhaps the same issue occurs with the links in Table 7, but it is creative.

[Improvement] Rewrite the sentence ("During the update process, the policy model for next iteration data generation can be fine-tuned from either the initial model or the model in the current iteration") in Selection and Update on pg 35 to: "During the update process, the policy model for the next iteration of data generation can be fine-tuned from either the initial model or the model in the current iteration". The current wording appears incorrect.

[Improvement] It is very difficult to read Table 9. Borderline illegible.

[Improvement] Change 6.7 to "Retrieval-Augmented Generation"

[Improvement] Table 10 has too little vertical space once again. Cannot read.

[Critical] I suggest lessening the position aspect in this paper. I found the best parts of this work to be the survey. In its current form, it is as if different authors wrote the paper: up until the end of Section 3 seems to be nothing more than hypotheticals, exaggerations, and unsupported claims. At least from Section 4 onward, there is more of a focus on the practical means of these technologies, but the first 8 pages or so seem to be more sci-fi than science. The part I struggle with is that scientists today still struggle to understand the human mind - how consciousness develops, why we think the way we do, and why others have conditions that inhibit their thinking. For example, what makes someone a murderer? Is it the way their brain is wired? Is it hormonal? Is it from their upbringing? How do we explain people with "normal upbringing" who go on to commit heinous acts of violence? Thus, how can we even attempt to draw a parallel between a machine and a human when we barely understand what it means to be a human? Otherwise, it appears nothing more than speculation. I think the authors should remove these aspects, or simply make it more explicit and less ambitious in the supposed parallels.

[Critical] The scale and scope of the article are a bit overwhelming. While the survey appears comprehensive, I almost find that it perhaps loses focus as a result of this. It becomes unclear how topics are interconnected as it begins to devolve more into a laundry list of related works and the types of problems they seek to resolve. A greater effort could be made to summarize each section and how it contributes to an overall theme.

**Strengths And Weaknesses:**

Strengths:

A large and comprehensive survey of related works to improving LLMs in their performance across a wide suite of applications.

I believe the audience of TMLR may benefit from such a comprehensive survey, but, to me, it lacks a coherent structure that makes it easy to follow or digest.

Weaknesses:

I dislike the idea of already categorizing developments in LLMs into "acts". This feels premature. Act I is only from 2020 to 2023, and Act II is from 2024 to the present. The distinction between these two, regarding the presence of test-time scaling techniques, seems overexaggerated to me, and perhaps it might be best to reserve declaring a separate Act for a greater technological innovation, something akin to the impact of the Transformer architecture.

I find the writing style to be a bit overly ambitious, and perhaps shares a border with, or is a step away from, science fiction (e.g., "This bidirectional cognitive exchange marks our entry into a new era of
intelligence symbiosis" -pg. 5). For instance, AlphaGo's famous "Move 37" may transcend human intuition as it surprised many, even the world's best Go player, but can we really say that the AI demonstrates "thinking approaches"? The scale of hardware used for self-play training (e.g., 64 GPUs, 19 CPU parameter servers), as well as the proposed algorithms, allowed for 29 million games of self-play to be generated; this is an amount of games that no human is likely to ever physically be able to replicate. However, to be able to still reasonably challenge and play against a machine capable of operating on such a scale demonstrates the remarkable efficiency of human learning and human thinking. In summary, there is still a disconnect between how AIs think and how humans think. Learning for AI is incredibly inefficient by comparison. This is a well-known issue in neural networks regarding their sample inefficiency, so the overall theme that we are achieving cognitive engineering in LLMs or AI, and they are achieving or understanding human thinking, appears to lack scientific rigor.

This feels like a simple rebranding of LLMs, where incorporating reinforcement learning and some inference optimization suddenly yields a supposed new field, "cognition engineering".

Page 5: " Traditional AI systems primarily operated at the data and information levels, whereas first-generation LLMs achieved significant breakthroughs at the knowledge level". Except that is incorrect. A simple, rudimentary artificial neural network (ANN) with a single layer and 4 neurons would already count as operating on the level of knowledge in the DIKW pyramid. An ANN is, by definition, a rule-based system that fundamentally relates patterns and exploits relationships between concepts to conduct inferences. The study of ANNs with their direct one-to-one and mathematical relationship to logical models has been investigated and proven in the field. We do not need to credit LLMs for any success here.

The claim, "Within the DIKW framework, engineering methods can be viewed as the process by which humans consciously guide systems to ascend from the data level to the wisdom level." seems misguided. Engineering is a broad field and a broad word to describe an action. There are many times engineering does not require any form of ascending from the data level to the level of wisdom. For instance, engineering a bridge/building, a heart monitor, a drinking cup, etc., does not seek to guide the systems from data to wisdom. Humans interacting with the systems may eventually obtain data and then derive wisdom from it, but the wisdom is not a property inherent in the inanimate objects. This should be rephrased to reduce the scope of the claim.

The definition of cognition engineering:

"Cognition Engineering is a systematic methodology that constructs and optimizes AI systems’ ability to ascend from the knowledge to wisdom levels of the DIKW pyramid through specific design patterns, training strategies, and computational allocations. It enables AI systems to engage in deep thinking, complex reasoning, and creative problem-solving, exhibiting cognitive characteristics similar to human wisdom-level traits"

The reference to the DIKW pyramid seems unnecessary. I would suggest cutting it and maybe rephrasing it to:

"Cognition Engineering is a systematic methodology that constructs and optimizes AI systems’ ability to ascend from knowledge to wisdom by incorporating specific design patterns, training strategies, and computational allocations. More specifically, the AI system is capable of deep understanding, judgment, creativity, and metacognition."

This is reflected in the DIKW pyramid you included as Figure 1, and the language in the last sentence is similar to the examples listed at the level of wisdom of the Figure, too. My issue with the prior definition was that for an AI system to have "wisdom", it is listed that "deep thinking" or "complex reasoning" must be achieved, but we already have, and for some time, too, AIs that can do this. For instance, the very act of learning to play Atari games (Mnih et al., 2015) with nothing but raw pixel information (from the computer's point of view) is an impressive and complex feat that humans would likely be unable to perform, if under the same conditions.

I am still unsure why 3.2.2 is called "test-time scaling". The examples of work related to this effort, such as Chain-of-Thought, Tree search, self-correction & verification, etc., are then related to a "cognitive workspace". I would almost find this more intuitive than "test-time scaling". Please elaborate on why you chose the former term over the latter. For instance, it appears "test-time scaling" often refers to "...increasing the compute at test time to get better results" [1]. The previous does not sound related to this definition.

3.2 Three Pillars: It seems that these three directions can be reduced to as follows: 3.2.1 more data and better quality, 3.2.2 intermediate reasoning, 3.2.3 reinforcement learning. Of these, it only seems that 3.2.2 might be the most appropriate to attribute to cognition engineering. Again, 3.2.1 (big data) I am unsure if this is what humans use - we certainly gain from having more experiences and observations, but we do not need anywhere near the scale that computers/AIs do to learn a task/activity/subject, even to just recognize a picture as a cat. So, I would argue that the fundamental transformation in how LLMs acquire knowledge is to improve their sample efficiency, not throw more and better data at it. 3.2.3 Self-training with RL: There is some work [2] suggesting some humans exhibit learning similar to RL, but arguing humans learn overall, or that RL is what forms our cognition, is likely an oversimplification analogous to how ANNs are an oversimplification of the neural pathways in our brain.

There appears to be a disconnect in 3.3 - earlier, test-time scaling was a foundation to cognitive engineering, now it is the most immediate and promising avenue for realizing cognitive engineering in practice. This suggests that 3.2.3 efforts are not as substantial toward this goal, and raises the question of why not position cognitive engineering around test-time scaling more directly.

I feel that a work that routinely discusses "thinking" and attempts to implement this in such detail that it replicates human thinking needs to quantify or define this more explicitly. What fundamentally separates thinking from other activities? Why not draw more from psychology on defining what it means to think? Or works from neuroscience? Human thinking is likely not even logical. Humans lie, cheat, steal, murder, etc., and all of these activities use human thinking. Why would we want AIs that replicate human thinking? This could, and arguably should, be discussed in greater detail. In other words, we likely don't want *all* the properties of human thinking, only the "good" - but what are the good? How do we keep only the good parts of human thinking? How do we know that the AI systems aren't just predicting what we want them to predict to emulate thinking?

Section 4 lacks concrete detail. Where does Eq. 1 or Eq. 2 come from? Is there any work that follows this convention? Figure 2: What are s_{1} or r_{1}? I assume for the latter, these are state and reward, respectively, at time step 1, but you should clearly state this. ORM and PRM should be more closely defined near Figure 2, perhaps in the caption.

How is parallel sampling a type of test-time scaling? It seems it would only increase the amount of computational resources during inference, not optimally manage them. Perhaps it is more of a greedy choice here.

pg. 11 for Query-aware sampling: Difficulty-Adaptive Self-Consistency (DSC)

Figure 4 is hard to read. More vertical space should be given to the cells so that the text size is larger.

ToT on pg. 13?

"...integrates data collection into the search process; 3) From ORM to PRM: To avoid the high cost of training a PRM, this method aims to derive a PRM from an ORM" <-- what is "this method"? These look like two separate thoughts

Beam search and A* are not versions of breadth-first search (BFS). A* is an informed search algorithm, and BFS is an uninformed search. Beam search can be applied to various searches.

"This phenomenon has been described as the RL scaling phenomenon or the “Aha moment.” " <-- Can you put a source for where it has been described as such? Also, "aha moment" is not very scientific language for a journal article.

pg. 37: "has transformed software development and boosted productivity." <-- this is still a contested issue. Many actually argue that it can slow down software development because of having to fix code generated by AI due to its errors. It is also reported to have security issues [3]. Overall, this is just an unsupported opinion. At the very least, it is too early to declare it is a fact.

[1] Muennighoff, N., Yang, Z., Shi, W., Li, X. L., Fei-Fei, L., Hajishirzi, H., … Hashimoto, T. (2025). s1: Simple test-time scaling. arXiv [Cs.CL]. Retrieved from http://arxiv.org/abs/2501.19393
[2] Menghai Pan, Weixiao Huang, Yanhua Li, Xun Zhou, Zhenming Liu, Jie Bao, Yu Zheng, and Jun Luo. 2020. Is Reinforcement Learning the Choice of Human Learners? A Case Study of Taxi Drivers. In Proceedings of the 28th International Conference on Advances in Geographic Information Systems (SIGSPATIAL '20). Association for Computing Machinery, New York, NY, USA, 357–366. https://doi.org/10.1145/3397536.3422246
[3] Negri-Ribalta C, Geraud-Stewart R, Sergeeva A, Lenzini G. A systematic literature review on the impact of AI models on the security of code generation. Front Big Data. 2024 May 13;7:1386720. doi: 10.3389/fdata.2024.1386720. PMID: 38803522; PMCID: PMC11128619.

---

> ### Author Response · Authors · 2025-08-02
> **Response to Reviewer cTi9 (1/7)**
>
> Thank you for your thoughtful review and constructive feedback. We appreciate your positive comments on the paper's comprehensive survey. And we have provided point-to-point changes to address your concerns as follows:
> >  **Requested Changes 1:** Please clarify if the following is a direct quote or your own words (pg. 4). It is currently in a block quote, so it appears to be a quote without a proper citation or attribution:
> Cognition Engineering is the systematic and constructive development of AI thinking capabilities through test-time scaling paradigms that transcend traditional pretraining approaches. This methodology represents the deliberate cultivation of deep cognitive processes in artificial systems through both human cognitive pattern distillation and AI-driven discovery (e.g., reinforcement learning)
>
> **Response:** Thank you for your valuable feedback. To clarify that the text in the block quote is our own proposed definition and not a direct quotation, we have added an introductory sentence before it. This change is highlighted in blue on page 4 of the revised manuscript.
>
>
> > **Requested Changes 2:**  Flip the way an acronym is defined; first use is the spelled-out version, then the acronym in parentheses: e.g., (pg. 5) Data-Information-Knowledge-Wisdom (DIKW)
>
> **Response:** Thank you for pointing this out. We have corrected this instance on page 5 and checked the rest of the manuscript for consistency. The change on page 5 has been highlighted in blue in the revised manuscript.
>
>
> > **Requested Changes 3:** Paragraph at the top of page 6: provide more citations behind some of these claims, definitions of cognition, etc., with more work in the field. Cognitive science must have some impactful, foundational, and historical works before 1995. Please cite those here.
>
> **Response:** Thank you for your feedback. As requested, we have added two foundational, pre-1995 citations (Newell et al., 1972; Posner et al., 1993) to the paragraph at the top of page 6 of the revised manuscript.
>
> > **Requested Changes 4:**  Why should we care about the DIKW pyramid theory? Where has it been helpful? Please explain and cite these.
>
> **Response:** Thank you for your insightful question regarding the DIKW pyramid theory. The DIKW pyramid is a fundamental theory in information science, and we find its conceptual leap from 'knowledge' to 'wisdom' particularly relevant. It serves as a valuable framework for our work because we aim to advance LLMs beyond simple knowledge retrieval towards deep reasoning models with wisdom-like cognitive behaviors, such as reflection and self-correction. In the revised version, we have now added a sentence that shows the importance of the DIKW pyramid theory, along with supporting citations (Rowley, 2007; Zins, 2007). The new text is highlighted in blue at the bottom of page 5 of the revised manuscript. Furthermore, Sections 1-3 were also updated to address other comments (see responses to Requested Changes 19 & 20).
>
> > **Requested Changes 5:** Add a space between "perplexity" and "(Cui et al., 2025b) in paragraph "Finetuning on compressed responses using heuristic methods" on page 20.
>
> **Response:** Thank you for pointing this out. We have added the missing space as requested. Due to other revisions, the correction is now on page 19 of the revision.
>
>
> >**Requested Changes 6:** pg. 19: "are as follows." Change the period to a colon - so far, and throughout most of the text, you use a colon to precede the list of items
>
> **Response:** Thank you for pointing out this inconsistency. We have corrected the punctuation on page 19 as suggested. While reviewing the manuscript for consistency, we also identified and corrected other similar instances. These changes are highlighted in blue in the revised manuscript on pages 10, 18, and 47.

---

> > ### Author Response · Authors · 2025-08-02
> > **Response to Reviewer cTi9 (2/7)**
> >
> > > **Requested Changes 7:** You need to ensure you define all acronyms. "RL" is never defined, though it is obviously referring to reinforcement learning. It is considered best practice to follow. Same with "SFT" - these may be obvious today, but in the future, it may be less clear as the field changes and new terms are introduced
> >
> >  > **Requested Changes 9:** Be more consistent with when to use acronyms. There are instances where "RL" is used, and others (later on) where "reinforcement learning" is used. Same issue with "best-of-N" (pg. 24).
> >
> > **Response:** Thank you for these helpful comments on improving the consistency of our acronym usage. We are addressing Requested Changes 7 and 9 together here as they both relate to this topic.
> >
> > Following your suggestions, we have implemented a consistent, two-part rule for all acronyms throughout the manuscript:
> > - Each acronym is now explicitly defined at its first appearance (addressing Requested Change 7).
> > - After its definition, the acronym is used for all subsequent mentions to ensure consistency (addressing Requested Change 9).
> >
> > We have performed a thorough review to apply this rule. The table below summarizes these corrections, showing where each acronym is first defined and on which subsequent pages the terminology was corrected for consistency. All of these changes have been highlighted in blue in the revised manuscript.
> >
> > | Acronym | First Defined on Page | Corrected for Consistent Usage on Pages |
> > | :--- | :--- | :--- |
> > | RL | 4 | 18, 22, 24, 30, 37, 41, 42 |
> > | SFT | 18 | 20, 31, 45 |
> > | BoN | 9 | 23, 43 |
> > | RLHF | 27 | N/A |
> > | DPO | 20 | N/A |
> >
> > >**Requested Changes 8:** write out C3oT as "Conditioned Compressed Chain-of-Thought," then define the acronym
> >
> > **Response:** Thank you for this suggestion. As requested, we have now written out the full name "Conditioned Compressed Chain-of-Thought" before providing the acronym (C3oT). This correction can be found on page 19 of the revised manuscript and is highlighted in blue.
> >
> >
> > > **Requested Changes 10:** Table 4: "curriculum sampling strategy" is CSS, but the iterative lengthening strategy is only "IL" - please change IL to ILS.
> >
> > **Response:** Thank you for pointing out this inconsistency. To ensure our acronyms are consistent, we have changed "IL" to "ILS" in Table 4, as you suggested. This correction can be found on page 25, highlighted in blue.
> >
> > > **Requested Changes 11:** Very hard to read Table 5 - not enough vertical space for the content. This would perhaps be better as a well-organized paragraph, with a more shortened table to help support the organization (less detail on method and evidence, simply pointing out the problem that is being solved in the related work).
> >
> >
> > **Response:** Thank you for your valuable feedback on the readability of Table 5. To address this, we have revised the table to improve its clarity and layout. Specifically, we simplified the content by removing the "Evidence" column and adjusted the formatting to increase vertical space, making it easier to read. The revised table can be found on page 26 of the manuscript.
> >
> >
> > > **Requested Changes 12:** I understand the reasoning behind the use of emojis in some parts of this article, but their frequent use in others feels misplaced and detracts from the article's professional appearance (e.g., Table 6); this is not a GitHub README.md file. For instance, the snowflake icon, to me, in Table 6 suggests that the architecture's parameters are frozen, which is certainly possible, but instead it indicates the model needs to perform inference. This is not very clear, although parameters are usually frozen during inference.
> >
> > **Response:** Thank you for your suggestion. We have removed the emojis from Table 6 and Section 5.1.2 and replaced them with text to improve professionalism and clarity. These changes are highlighted in blue on page 27.

---

> > > ### Author Response · Authors · 2025-08-02
> > > **Response to Reviewer cTi9 (3/7)**
> > >
> > > >**Requested Changes 13:** Please clarify how you determine, with only a yes or no, whether the training algorithm achieves the RL scaling as reported in Table 6? That seems to be a qualitative, subjective judgment based on how the models demonstrate cognitive behaviors, including self-reflection and self-correction. So, where is your data showing this? Or whose data are you referring to? If the former, please present it. If the latter, please explain and cite it.
> > >
> > > **Response:** Thank you for raising this very important point. Our original "yes/no" assertion was a summary based on the existing literature. The RL scaling phenomenon has three important features: performance improves, response length increases, and it demonstrates cognitive behaviors like self-reflection and self-correction. Algorithms like PPO, GRPO, and REINFORCE++ have been reported to exhibit the RL scaling phenomenon, as shown in Table 4, while a similar phenomenon has not been reported for the original REINFORCE algorithm. Our intention was simply to highlight that this phenomenon is not unique to a single algorithm.
> > >
> > > However, you raise a valid concern about the robustness of such a binary claim. Given the rapid pace of development in this field, future work could present counterexamples (e.g., a study demonstrating that REINFORCE can also exhibit the RL scaling phenomenon). Therefore, to ensure the long-term accuracy and rigor of our paper, we have removed this claim. Accordingly, we have removed this column from Table 6 and the corresponding discussion from the text in the "Comparisons with different algorithms" section. These revisions can be found on page 27.
> > >
> > > >**Requested Changes 14:** "As shown in Table 4, most RL scaling work utilizes Qwen2.5 as the base model." This is only shown as logos. This is not very accessible to reading-impaired individuals. You should at the very least map each logo to the text. Again, these companies may go bankrupt in the future, so their logos are perhaps not future-proof or may not be easily recognized in the future. Perhaps the same issue occurs with the links in Table 7, but it is creative.
> > >
> > > **Response:** Thank you for your valuable suggestions. To improve accessibility and ensure our tables are future-proof, we have replaced the logos in Table 4 with plain text and added a caption to Table 7 to define the icon used. These changes are highlighted in blue on page 25 and page 32 of the revised manuscript.
> > >
> > >
> > >  >**Requested Changes 15:** Rewrite the sentence ("During the update process, the policy model for next iteration data generation can be fine-tuned from either the initial model or the model in the current iteration") in Selection and Update on pg 35 to: "During the update process, the policy model for the next iteration of data generation can be fine-tuned from either the initial model or the model in the current iteration". The current wording appears incorrect.
> > >
> > >
> > > **Response:**  Thank you for the correction. We have updated the sentence on page 34 with your suggested wording. The change is highlighted in blue in the revision.
> > >
> > >
> > > > **Requested Changes 16:** It is very difficult to read Table 9. Borderline illegible.
> > >
> > >
> > > **Response:** Thank you for your feedback on Table 9. To improve its legibility, we have split the original table into two (now Table 9 and Table 10), which allows for a clearer layout and larger font. The new tables are on pages 35 and 36 of the revised manuscript.
> > >
> > >
> > >  >**Requested Changes 17:** Change 6.7 to "Retrieval-Augmented Generation"
> > >
> > > **Response:** Thank you for this correction. We have updated the section title on page 44 as requested. The change is highlighted in blue in the revision.
> > >
> > >
> > >
> > >
> > > > **Requested Changes 18:** Table 10 has too little vertical space once again. Cannot read.
> > >
> > > **Response:** Thank you for pointing out the issue with Table 10. We have improved its readability by removing the "Sequence Parallelism" column and increasing the font size and spacing. The updated table can be found on page 47 of the revised manuscript.

---

> > > > ### Author Response · Authors · 2025-08-02
> > > > **Response to Reviewer cTi9 (4/7)**
> > > >
> > > > > **Requested Changes 19:** I suggest lessening the position aspect in this paper. I found the best parts of this work to be the survey. In its current form, it is as if different authors wrote the paper: up until the end of Section 3 seems to be nothing more than hypotheticals, exaggerations, and unsupported claims. At least from Section 4 onward, there is more of a focus on the practical means of these technologies, but the first 8 pages or so seem to be more sci-fi than science. The part I struggle with is that scientists today still struggle to understand the human mind - how consciousness develops, why we think the way we do, and why others have conditions that inhibit their thinking. For example, what makes someone a murderer? Is it the way their brain is wired? Is it hormonal? Is it from their upbringing? How do we explain people with "normal upbringing" who go on to commit heinous acts of violence? Thus, how can we even attempt to draw a parallel between a machine and a human when we barely understand what it means to be a human? Otherwise, it appears nothing more than speculation. I think the authors should remove these aspects, or simply make it more explicit and less ambitious in the supposed parallels.
> > > >
> > > > **Response:**  Thank you for your constructive feedback on the framing of our paper's introduction. We agree with your suggestion to improve the scientific rigor of the introductory sections (1-3).
> > > >
> > > > To achieve this, we have revised these sections. A key part of this revision was to shift our definition of "cognition engineering" away from the direct imitation of human thought processes and toward a more concrete focus on developing the deep reasoning capabilities of LLMs.
> > > >
> > > > The specific changes are as follows:
> > > > - In Section 1: We have removed speculative phrasing, such as the sentence, "The shift implies in Act II, the cognitive exchange becomes bidirectional …This bidirectional cognitive exchange marks our entry into a new era of intelligence symbiosis," to maintain a more scientific tone.
> > > > - In Section 2: We have removed subsections 2.2 and 2.3. The remaining content has been streamlined and reorganized to better support the paper's main focus on developing the deep reasoning capabilities of LLMs.
> > > >
> > > > These revisions can be found from pages 4 to 6 of the updated manuscript. We believe these changes improve the paper's focus and clarity.
> > > >
> > > >
> > > > > **Requested Changes 20:** The scale and scope of the article are a bit overwhelming. While the survey appears comprehensive, I almost find that it perhaps loses focus as a result of this. It becomes unclear how topics are interconnected as it begins to devolve more into a laundry list of related works and the types of problems they seek to resolve. A greater effort could be made to summarize each section and how it contributes to an overall theme.
> > > >
> > > > **Response:** Thank you for this very constructive feedback on the paper's overall structure and focus. We agree with your suggestion to have a clear narrative thread to connect the different topics. To address this, we have made several structural revisions to provide a much clearer roadmap for the reader.
> > > >
> > > > Our main changes are as follows:
> > > > - In Section 1, we have revised the definition of "cognition engineering" to directly foreshadow the structure of the survey. The new definition explicitly mentions "interventions at inference time" (which maps to Section 4) and "corresponding training strategies" (which maps to Section 5).
> > > > - In Section 2, we have sharpened our focus on the core concept—developing the deep reasoning ability of LLMs—to maintain a consistent theme throughout the introduction.
> > > > - In Section 3, we have made two major changes to guide the reader:
> > > >   - First, we introduced a new roadmap figure (Figure 1) to visually represent the main structure of our survey and the relationships between its parts.
> > > >   - Second, we revised the text in Section 3.3 to explicitly guide the reader, explaining how the following survey sections connect to the overarching theme.
> > > >
> > > > We believe these changes to the introductory sections now better connect the topics within our survey, making its flow and scope much easier to follow. These revisions can be found on pages 4-7 of the updated manuscript.

---

> > > > > ### Author Response · Authors · 2025-08-02
> > > > > **Response to Reviewer cTi9 (5/7)**
> > > > >
> > > > > In addition to the point-by-point changes, we have also made the following revisions to address the weaknesses you identified:
> > > > >
> > > > > > **Weakness1:** I find the writing style to be a bit overly ambitious, and perhaps shares a border with, or is a step away from, science fiction (e.g., "This bidirectional cognitive exchange marks our entry into a new era of intelligence symbiosis" -pg. 5). For instance, AlphaGo's famous "Move 37" may transcend human intuition as it surprised many, even the world's best Go player, but can we really say that the AI demonstrates "thinking approaches"? The scale of hardware used for self-play training (e.g., 64 GPUs, 19 CPU parameter servers), as well as the proposed algorithms, allowed for 29 million games of self-play to be generated; this is an amount of games that no human is likely to ever physically be able to replicate. However, to be able to still reasonably challenge and play against a machine capable of operating on such a scale demonstrates the remarkable efficiency of human learning and human thinking. In summary, there is still a disconnect between how AIs think and how humans think. Learning for AI is incredibly inefficient by comparison. This is a well-known issue in neural networks regarding their sample inefficiency, so the overall theme that we are achieving cognitive engineering in LLMs or AI, and they are achieving or understanding human thinking, appears to lack scientific rigor.
> > > > >
> > > > >
> > > > > **Response:** Thank you again for this important feedback. This concern is closely related to the points you raised in Requested Change 19, and we have addressed it as part of a comprehensive revision of our paper's introduction. As detailed in our response to Requested Change 19, we agree that some sentences in Sections 1-2 may indeed appear overly speculative. We have revised Sections 1-2 to remove unsupported parallels to human cognition and provide a more grounded introduction to our survey. We believe these extensive changes also resolve the concerns about the "sci-fi" tone you have raised here.
> > > > >
> > > > >
> > > > > > **Weakness2:**  Page 5: " Traditional AI systems primarily operated at the data and information levels, whereas first-generation LLMs achieved significant breakthroughs at the knowledge level". Except that is incorrect. A simple, rudimentary artificial neural network (ANN) with a single layer and 4 neurons would already count as operating on the level of knowledge in the DIKW pyramid. An ANN is, by definition, a rule-based system that fundamentally relates patterns and exploits relationships between concepts to conduct inferences. The study of ANNs with their direct one-to-one and mathematical relationship to logical models has been investigated and proven in the field. We do not need to credit LLMs for any success here.
> > > > >
> > > > > **Response:** Thank you for this insightful correction regarding the DIKW hierarchy and traditional AI. You make an excellent point that even simple ANNs can be considered to operate at the "knowledge" level, and we appreciate you sharing your expertise. To correct this inaccuracy, we have removed the statement "Traditional AI systems primarily operated at the data and information levels” from our discussion. This revision can be found on page 5 of the manuscript.
> > > > >
> > > > >
> > > > >
> > > > > > **Weakness3:** The claim, "Within the DIKW framework, engineering methods can be viewed as the process by which humans consciously guide systems to ascend from the data level to the wisdom level." seems misguided. Engineering is a broad field and a broad word to describe an action. There are many times engineering does not require any form of ascending from the data level to the level of wisdom. For instance, engineering a bridge/building, a heart monitor, a drinking cup, etc., does not seek to guide the systems from data to wisdom. Humans interacting with the systems may eventually obtain data and then derive wisdom from it, but the wisdom is not a property inherent in the inanimate objects. This should be rephrased to reduce the scope of the claim.
> > > > >
> > > > > **Response:** Thank you for pointing out the flaw in this claim. We agree that our definition of the term "engineering" was too narrow. To resolve the issue, we have removed the relevant section (formerly Section 2.2). This change is reflected on page 6 of the revised manuscript. As this change was part of our revision of the introductory sections, we have also noted this action in our response to Requested Change 19.

---

> ### Author Response · Authors · 2025-08-02
> **Response to Reviewer cTi9 (6/7)**
>
> > **Weakness4:**  I am still unsure why 3.2.2 is called "test-time scaling". The examples of work related to this effort, such as Chain-of-Thought, Tree search, self-correction & verification, etc., are then related to a "cognitive workspace". I would almost find this more intuitive than "test-time scaling". Please elaborate on why you chose the former term over the latter. For instance, it appears "test-time scaling" often refers to "...increasing the compute at test time to get better results" [1]. The previous does not sound related to this definition.
>
> **Response:** Thank you for this excellent question, which gives us an opportunity to clarify our terminology. We agree with the definition you provided: "test-time scaling" refers to increasing compute at test time to get better results. Our rationale for using this term is that methods like Chain-of-Thought, tree search, and self-correction do precisely this: instead of directly outputting a final answer, they increase computation during the inference phase to improve the final output. For instance:
> - In Tree Search, performance can be improved by scaling the tree depth or breadth, thereby increasing compute at test time.
> - In Self-Correction, performance can be improved by scaling the number of revision steps, which also increases compute at test time.
> This classification is also consistent with recent literature [1, 2], which categorizes these techniques as test-time scaling.
>
> > **Weakness5:** There appears to be a disconnect in 3.3 - earlier, test-time scaling was a foundation to cognitive engineering, now it is the most immediate and promising avenue for realizing cognitive engineering in practice. This suggests that 3.2.3 efforts are not as substantial toward this goal, and raises the question of why not position cognitive engineering around test-time scaling more directly.
>
> **Response:** Thank you for pointing out this potential disconnect in our framing of Section 3.3. You are correct that our original wording could have been interpreted as diminishing the importance of other pillars relative to test-time scaling. To address this, we have revised the text in Section 3.3 to clarify that all the pillars we discuss are crucial for advancing cognition engineering. The revised section now provides a clearer roadmap, explicitly stating that Section 4 corresponds to §3.2.2 and Section 5 corresponds to §3.2.3. This change is also part of the revision for Requested Change 20. Please refer to page 7 of the revised manuscript, highlighted in blue.
>
>
> > **Weakness6:**  Section 4 lacks concrete detail. Where does Eq. 1 or Eq. 2 come from? Is there any work that follows this convention? Figure 2: What are s_{1} or r_{1}? I assume for the latter, these are state and reward, respectively, at time step 1, but you should clearly state this. ORM and PRM should be more closely defined near Figure 2, perhaps in the caption.
>
> **Response:** Thank you for pointing out these areas in Section 4 that required more detail and clarification. We have made the following revisions to address each of your points:
> - Sources for Equations: We have now added citations for Equation 1 (from Welleck et al., 2024) and Equation 2 (from Qu et al., 2025) to provide their sources. These can be found on page 7 and page 8, respectively.
> - Clarifications for Figure 2: We have significantly expanded the caption for Figure 2. The new caption now explicitly defines:
>   - The notations used in the figure.
>   - The acronyms ORM and PRM.
>
>   This revision is on page 8 of the updated manuscript.
>
>
> > **Weakness7:**  How is parallel sampling a type of test-time scaling? It seems it would only increase the amount of computational resources during inference, not optimally manage them. Perhaps it is more of a greedy choice here.
>
> **Response:** Thank you for this insightful question, which helps us further clarify our "test-time scaling" classification. Our reasoning for including parallel sampling is as follows:
> - It fits the core definition: Parallel sampling directly increases the computational resources used during inference—by scaling the sampling number—to improve the final result. This aligns with the definition of test-time scaling.
> - Optimality is not a prerequisite: You raised an excellent point regarding optimal management. While some works do optimize parallel sampling to be more compute-optimal (as discussed in Section 4.1.3), we argue that the defining characteristic of "test-time scaling" is the scaling of compute itself, not necessarily its optimality.
> This classification is also supported by recent literature [1, 3].
>
>
> Reference:
>
> [1] Scaling LLM Test-Time Compute Optimally can be More Effective than Scaling Model Parameters, in ICLR 2025.
>
> [2] Can 1B LLM Surpass 405B LLM? Rethinking Compute-Optimal Test-Time Scaling, in arxiv 2025.
>
> [3] Large Language Monkeys: Scaling Inference Compute with Repeated Sampling, in arxiv 2025.

---

> ### Author Response · Authors · 2025-08-02
> **Response to Reviewer cTi9 (7/7)**
>
> > **Weakness8:** pg. 11 for Query-aware sampling: Difficulty-Adaptive Self-Consistency (DSC)
>
> **Response:** Thank you for this correction. We have defined the DSC acronym as requested on page 10 of the revision.
>
>
> >**Weakness9:** Figure 4 is hard to read. More vertical space should be given to the cells so that the text size is larger.
>
> **Response:**  Thank you for your feedback on Figure 4. We have improved its readability by increasing the vertical space and enlarging the font size, as suggested. The revised figure is on page 11 of the manuscript.
>
> >  **Weakness10:** ToT on pg. 13?
>
> **Response:** Thank you for pointing out this undefined acronym. We have now defined "ToT" as "Tree-of-Thought" upon its first use on page 12. This correction has been made in the revised manuscript.
>
> > **Weakness11:** "...integrates data collection into the search process; 3) From ORM to PRM: To avoid the high cost of training a PRM, this method aims to derive a PRM from an ORM" <-- what is "this method"? These look like two separate thoughts
>
> **Response:** Thank you for this suggestion. We have revised this sentence for clarity, replacing the term "this method" with "a line of work". The correction on page 13 of the manuscript is highlighted in blue.
>
>
>
> > **Weakness12:** Beam search and A* are not versions of breadth-first search (BFS). A* is an informed search algorithm, and BFS is an uninformed search. Beam search can be applied to various searches.
>
> **Response:** Thank you for this important correction. We have fixed the misclassification of A* and Beam search on page 14. They are now presented as independent strategies, not as subtypes of BFS. The revision is highlighted in blue.
>
>
>
>
> > **Weakness13:** "This phenomenon has been described as the RL scaling phenomenon or the “Aha moment.” " <-- Can you put a source for where it has been described as such? Also, "aha moment" is not very scientific language for a journal article.
>
> **Response:** Thank you for these helpful suggestions. We have addressed your points as follows:
> - Added Citations: As requested, we have now provided sources [4, 5] on page 24 for the terms "RL scaling phenomenon" and "Aha moment."
> - Regarding the term "aha moment": Thank you for raising this valid concern. We have added citations [4, 5] to show that the term 'Aha moment' has been used in the literature to describe this phenomenon. We have intentionally used it only once to introduce the concept as described in these sources, and thereafter exclusively use the more formal 'RL scaling phenomenon' throughout the manuscript to maintain a scientific tone.
>
>
> > **Weakness14:** pg. 37: "has transformed software development and boosted productivity." <-- this is still a contested issue. Many actually argue that it can slow down software development because of having to fix code generated by AI due to its errors. It is also reported to have security issues [3]. Overall, this is just an unsupported opinion. At the very least, it is too early to declare it is a fact.
>
>
> **Response:** Thank you for raising this valid point. We have revised the sentence to be more neutral by removing the expression 'boosted productivity.' This change can be found on page 37 of the revised manuscript.
>
> Reference:
>
> [4] DeepSeek-R1: Incentivizing Reasoning Capability in LLMs via Reinforcement Learning, in arxiv 2025.
>
> [5] Understanding r1-zero-like training: A critical perspective, in arxiv 2025.

---

> > ### Comment · Reviewer_cTi9 · 2025-08-07
> > **Response to Revisions**
> >
> > Sorry for my delay; I had to set aside enough time to review your revisions.
> >
> > Thank you for listening and following my suggested revisions. Overall, they look great, and I think the submission is much more aligned with the quality of an accepted article. I have just a few quick follow-ups I think may help in final edits, if you could please consider incorporating:
> >
> > 1. DIKW pyramid; include literature that critiques this framework. I think it is only fair to incorporate, however you deem fit, the published research that critiques this DIKW pyramid. For instance, some researchers have proposed alternative frameworks that extend it or modify it in some way, whereas others blatantly are against it (e.g., that it oversimplifies how knowledge is obtained, or that the terms are ill-defined)
> >
> > 2. Thank you for splitting the large table into Tables 9 and 10. Can you also consider allocating more vertical space to them? The text is still "squashed" and inconsistent with other tables. Please apply this same convention to the other tables as well, if needed.
> >
> > 3. Figure 1; thank you so much for including this. Minor nitpicks: could you try and modify the figure so that "Cognition Engineering" is not split in the orange rectangular block? Just move "Engineering:" to its own line. Further, it may look more aesthetically pleasing to restrict each purple cell in Figure 1 to follow the convention that the first line is the text, and the second line is the section reference (e.g., ($\S{6.1}$)). So, "Embodied AI" is on one line, and the next line, you have ($\S{6.5}$). Same for "Code", "Agent", "Safety", and "RAG".
> >
> > 4. In response to the discussion:
> >
> > >Weakness7: How is parallel sampling a type of test-time scaling? It seems it would only increase the amount of computational resources during inference, not optimally manage them. Perhaps it is more of a greedy choice here.
> >
> > >Response: Thank you for this insightful question, which helps us further clarify our "test-time scaling" classification. Our reasoning for including parallel sampling is as follows:
> >
> > >It fits the core definition: Parallel sampling directly increases the computational resources used during inference—by scaling the sampling number—to improve the final result. This aligns with the definition of test-time scaling.
> > >Optimality is not a prerequisite: You raised an excellent point regarding optimal management. While some works do optimize parallel sampling to be more compute-optimal (as discussed in Section 4.1.3), we argue that the defining characteristic of "test-time scaling" is the scaling of compute itself, not necessarily its optimality. This classification is also supported by recent literature [1, 3].
> > >Reference:
> >
> > >[1] Scaling LLM Test-Time Compute Optimally can be More Effective than Scaling Model Parameters, in ICLR 2025.
> >
> > >[2] Can 1B LLM Surpass 405B LLM? Rethinking Compute-Optimal Test-Time Scaling, in arxiv 2025.
> >
> > >[3] Large Language Monkeys: Scaling Inference Compute with Repeated Sampling, in arxiv 2025.
> > \end{quote}
> >
> > I think it is an important clarification. This would be nice to include as a footnote if it has not yet been done so.
> >
> > 5. In response to the discussion:
> >
> > >Weakness11: "...integrates data collection into the search process; 3) From ORM to PRM: To avoid the high cost of training a PRM, this method aims to derive a PRM from an ORM" <-- what is "this method"? These look like two separate thoughts
> >
> > >Response: Thank you for this suggestion. We have revised this sentence for clarity, replacing the term "this method" with "a line of work". The correction on page 13 of the manuscript is highlighted in blue.
> >
> > Thank you; can you also put a citation directly after "a line of work" on page 13? I assume this is Lu et al.'s AutoPSV
> >
> > Summary:
> >
> > I really appreciate your time and energy in revising the submission per our suggestions. The revised article looks very well done, and can serve as a valuable as well as impactful resource for the field.

---

> > > ### Author Response · Authors · 2025-08-09
> > > **Response to the follow-up suggestions from Reviewer cTi9**
> > >
> > > Thank you for your detailed and encouraging feedback on our revised manuscript. We are very grateful for your kind words and for recognizing the effort we put into the revisions. Your guidance has been invaluable in improving the quality of our work.
> > > We have carefully incorporated your latest suggestions into the new version of the manuscript. Please find our point-by-point responses below:
> > >
> > > > **Follow-up suggestion 1:** DIKW pyramid; include literature that critiques this framework. I think it is only fair to incorporate, however you deem fit, the published research that critiques this DIKW pyramid. For instance, some researchers have proposed alternative frameworks that extend it or modify it in some way, whereas others blatantly are against it (e.g., that it oversimplifies how knowledge is obtained, or that the terms are ill-defined)
> > >
> > > **Response:** Thank you for this excellent suggestion. We agree that acknowledging critiques of the DIKW pyramid adds important context and enhances the scientific rigor of our discussion. We have now added a footnote that discusses these limitations and explains our rationale for employing the framework. This can be found on page 5 of the revised manuscript.
> > >
> > > > **Follow-up suggestion 2:** Thank you for splitting the large table into Tables 9 and 10. Can you also consider allocating more vertical space to them? The text is still "squashed" and inconsistent with other tables. Please apply this same convention to the other tables as well, if needed.
> > >
> > > **Response:** Thank you for pointing this out. We have adjusted the formatting of Tables 9 and 10 to increase the vertical spacing, improving their readability. We have also reviewed all other tables to ensure consistent formatting. The changes can be seen on pages 35 and 36.
> > >
> > > > **Follow-up suggestion 3:** Figure 1; thank you so much for including this. Minor nitpicks: could you try and modify the figure so that "Cognition Engineering" is not split in the orange rectangular block? Just move "Engineering:" to its own line. Further, it may look more aesthetically pleasing to restrict each purple cell in Figure 1 to follow the convention that the first line is the text, and the second line is the section reference (e.g., (§ 6.1)). So, "Embodied AI" is on one line, and the next line, you have (§ 6.5). Same for "Code", "Agent", "Safety", and "RAG".
> > >
> > >
> > > **Response:** We appreciate these detailed aesthetic suggestions. We have revised Figure 1 according to your feedback: "Cognition Engineering" is no longer split, and the purple cells now follow the requested two-line convention. The updated figure is on page 5.
> > >
> > > > **Follow-up suggestion 4:**  I think it is an important clarification. This would be nice to include as a footnote if it has not yet been done so.
> > >
> > > **Response:** Thank you for this valuable suggestion. We agree that this clarification is important. We have incorporated this clarification regarding 'Parallel sampling' as a footnote in the manuscript to provide readers with the full context. This has been added on page 8.
> > >
> > > > **Follow-up suggestion 5:** Thank you; can you also put a citation directly after "a line of work" on page 13? I assume this is Lu et al.'s AutoPSV
> > >
> > > **Response:** Thank you for catching this and for the helpful suggestion. We have now added the appropriate citations (Lu et al., 2024; Yuan et al., 2024a) immediately following the phrase "a line of work" on page 13.
> > >
> > > Once again, we sincerely thank you for your time and constructive feedback throughout this revision process. We believe the manuscript is significantly stronger thanks to your input.

---

### Author Response · Authors · 2025-08-02
**General Response**

We sincerely thank all reviewers for their thorough and insightful feedback. We are encouraged by your positive comments on the paper's comprehensive survey (`cTi9`, `9Vyp`, `foa3`), its timeliness (`foa3`), and its potential value to the research community (`cTi9`, `9Vyp`, `foa3`). Your constructive critiques have been invaluable, and we have thoughtfully revised the manuscript to incorporate your suggestions. A summary of the key refinements is as follows:

**Introduction Sections (1-3)**

In response to feedback from all reviewers concerning the paper's framing and tone, we have carefully revised the introductory sections.
- In Section 1, we refined our core definition of "Cognition Engineering" to be more operational and to serve as a clearer framework for the survey. We also adjusted the language to ensure a more objective and scientific tone.
- In Section 2, we have improved the organization to enhance scientific rigor. To create a more direct and focused argument, we have streamlined the content to cohesively establish the core concepts supporting our framework.
- In Section 3, to create a clearer roadmap for the reader, we introduced a new figure (Figure 1) to visually illustrate the survey's structure and the relationships between its parts. We also revised the text, particularly in Section 3.3, to explicitly connect the survey's topics back to our central theme, ensuring a more logical flow into the main body of the paper.

**Survey Sections (4-8)**

Throughout the main survey sections, we have meticulously addressed the detailed feedback provided by the reviewers. This includes improving the readability of tables by adjusting spacing and layout; ensuring consistent definition and usage of all acronyms; adding necessary citations and clarifications for equations and figures; and revising the introductions to Sections 7 and 8 to better ground them within the paper's overall narrative.

We believe these refinements have further strengthened the manuscript, making it clearer, more focused, and more rigorously grounded. For a detailed account of how we addressed each specific point, please see our individual responses.

Thank you once again for your time and valuable contributions to improving our work. The revised manuscript has been uploaded, with changes from the original submission highlighted in blue. Should there be any further questions or suggestions, we warmly welcome continued discussion and are eager to engage further.

---

### Decision · Action_Editor_4uBL · 2025-09-10

**Recommendation:** Reject

**Additional Comments:**

This is a position-type paper that tries to frame the last half decade of large language model development in a framework of "cognition engineering". The initial reviews by the reviewers all felt that paper's definitions were vague and imprecise, which is a serious limitation for a paper that is intended to create a clarifying framework. Although the reviewers were impressed with the author's work to address their concerns, the limitations remain too severe to recommend acceptance.

**Audience:**

Yes

**Audience Explanation:**

Many of the reviewers felt that the survey in the paper was of great value to the community.

**Claims And Evidence:**

No

**Claims Explanation:**

In their initial reviews, all reviewers agreed that the paper was overly ambitious in tone, imprecise in its definitions, and had vague claims that were not fully backed up. After the rebuttal process, a couple reviewers felt more comfortable with the paper, but some serious concerns remained, namely lack of precision in the framing and ambiguity. The remaining concerns are too severe to recommend acceptance at TMLR. Although the reviewers appreciated the revision by the authors, I believe the overall thrust of the concerns in the initial reviews is still an issue in the current draft.